# Ubiquilin-2 liquid droplets catalyze α-synuclein fibril formation

Tomoki Takei [1,2], Yukiko Sasazawa [1,2,3,4], Daisuke Noshiro [5], Mitsuhiro Kitagawa[1,2], Tetsushi Kataura[2,6], Hiroko Hirawake-Mogi[1], Emi Kawauchi[2], Yuya Nakano[2], Etsu Tashiro [2,11], Tsuyoshi Saitoh[7,12], Shigeru Nishiyama[7], Seiichiro Ogawa[2], Soichiro Kakuta [8], Saiko Kazuno [8], Yoshiki Miura [8], Daisuke Taniguchi[1], Viktor I Korolchuk[9], Nobuo N Noda [5], Shinji Saiki [1,4,6], Masaya Imoto [1,2,4✉] & Nobutaka Hattori [1,3,4,10✉]

## Abstract

**Liquid–liquid phase separation (LLPS) and subsequent liquid–gel/solid transition are considered common aggregation mechanisms of proteins linked to neurodegenerative diseases. α-synuclein (α-syn), the main factor in Parkinson's disease pathology, has been reported to undergo LLPS, thereby accelerating aggregate formation. However, the precise molecular events involved in the early stages of α-syn aggregation remain controversial. In this study, we show that α-syn aggregation is promoted by droplets formed by ubiquilin-2 (UBQLN2), rather than by α-syn LLPS itself. During the liquid–gel/solid transition of UBQLN2 droplets, α-syn within the droplets transforms into pathogenic fibrils both in vitro and in cells. Immunohistochemistry of brain sections from sporadic Parkinson's disease patients revealed UBQLN2 in *substantia nigra* Lewy bodies, implicating UBQLN2 in α-syn aggregation in vivo. Furthermore, the small compound SO286 inhibited both UBQLN2 self-association and its interaction with α-syn by binding to the STI1 domain, thereby suppressing α-syn aggregation. These findings demonstrate that UBQLN2 droplets catalyze α-syn fibrillization and suggest that small molecules targeting fibril-catalyzing proteins such as UBQLN2 may represent a promising therapeutic approach for neurodegenerative diseases.**

**Keywords** Liquid–liquid Phase Separation; Ubiquilin-2; α-synuclein; Parkinson's Disease
**Subject Categories** Molecular Biology of Disease; Post-translational Modifications & Proteolysis

## Introduction

Some neurodegenerative disease-associated proteins form liquid droplets via liquid–liquid phase separation (LLPS) (Gruijs da Silva et al, 2022; Patel et al, 2015; Wegmann et al, 2018). Over time, these droplets transition from a highly labile liquid state to a hydrogel state, and eventually to a solid-like condensate, via self-interaction and oligomerization of the proteins within, thereby leading to the formation of amyloid fibrils (Alberti et al, 2019; Banani et al, 2017). Recently, α-synuclein (α-syn) has been reported to be one such protein (Hardenberg et al, 2021; Mukherjee et al, 2024; Mukherjee et al, 2023; Piroska et al, 2023; Poudyal et al, 2022; Ray et al, 2020; Sawner et al, 2021; Xu et al, 2022); however, the formation of α-syn liquid droplets in vitro reportedly requires extremely high protein concentrations (200 μM at physiological pH), and their liquid–gel/solid transition requires a relatively long incubation period (up to 20 days) (Ray et al, 2020). The mechanisms underlying the LLPS and liquid−gel/solid transitions of α-syn under physiological conditions therefore remain controversial.

Ubiquilins (UBQLNs) are shuttle proteins that are involved in protein quality control for the maintenance of homeostasis (Kleijnen et al, 2000; Ko et al, 2004; Lee et al, 2013; Stieren et al, 2011). Among the five human UBQLNs, UBQLN1, UBQLN2, and UBQLN4 share highly conserved domains (Marin, 2014). These three UBQLNs are multidomain proteins composed of an intrinsically disordered region, which includes two STI1 (STI1-1 and STI1-2) regions flanked by the N-terminal ubiquitin-like (UBL) domain and the C-terminal ubiquitin-associated domain (Zheng et al, 2020). Only UBQLN2 also contains a proline-rich region between the STI1-2 region and the ubiquitin-associated domain; this region contributes to the self-interactions of UBQLN2 (Dao et al, 2018). Recent studies have revealed that these UBQLNs undergo LLPS into liquid droplets under physiological conditions (Dao et al, 2018, 2022; Gerson et al, 2021).

[1]Department of Neurology, Juntendo University Faculty of Medicine, Tokyo, Japan. [2]Department of Biosciences and Informatics, Keio University, Kanagawa, Japan. [3]Research Institute for Diseases of Old Age, Juntendo University Graduate School of Medicine, Tokyo, Japan. [4]Division for Development of Autophagy Modulating Drugs, Juntendo University Faculty of Medicine, Tokyo, Japan. [5]Institute for Genetic Medicine, Hokkaido University, Sapporo, Japan. [6]Department of Neurology, Institute of Medicine, University of Tsukuba, Ibaraki, Japan. [7]Department of Chemistry, Keio University, Kanagawa, Japan. [8]Biomedical Research Core Facilities, Juntendo University Graduate School of Medicine, Tokyo, Japan. [9]Biosciences Institute, Faculty of Medical Sciences, Campus for Ageing and Vitality, Newcastle University, Newcastle upon Tyne, UK. [10]Neurodegenerative Disorders Collaborative Laboratory, RIKEN Center for Brain Science, Saitama, Japan. [11]Present address: Laboratory of Biochemistry, Showa Pharmaceutical University, Tokyo, Japan. [12]Present address: Institute of Medicine, University of Tsukuba, Tennodai, Tsukuba, Ibaraki, Japan. ✉E-mail: m.imoto.xz@juntendo.ac.jp; nhattori@juntendo.ac.jp

Although the functions of UBQLN LLPS remain unclear, it may be that UBQLN2 undergoes LLPS into membraneless organelles or condensates, such as stress granules, to transfer ubiquitinated proteins to proteasome or autophagy systems because UBQLN2 droplets are disassembled by interactions between UBQLN2 and ubiquitin (Dao et al, 2018; Kleijnen et al, 2000; Ma et al, 2023; Sandoval-Pistorius et al, 2023). Furthermore, UBQLN2 LLPS may contribute to the formation of intracytoplasmic inclusion bodies in neurodegenerative disorders (Alexander et al, 2018; Dao et al, 2019; Safren et al, 2024; Sharkey et al, 2018). However, the relationship between UBQLN2 LLPS and the development of neurodegenerative diseases, such as Parkinson's disease (PD), remains unclear.

In the present study, we established the role of UBQLN2 LLPS in α-syn aggregation under physiological conditions and propose a strategy for suppressing α-syn aggregation using a UBQLN2-binding compound.

# Results

## α-Syn is incorporated into UBQLN2 liquid droplets and transformed into aggregates in vitro

UBQLN2 has been widely implicated in several neurodegenerative diseases (Deng et al, 2011; Fahed et al, 2014; Gerson et al, 2021; Lin et al, 2022; Safren et al, 2024; Synofzik et al, 2012; Williams et al, 2012), including PD (Mori et al, 2012; Sandoval-Pistorius et al, 2023). We therefore examined the possible involvement of UBQLN2 LLPS in α-syn aggregation. Recombinant UBQLN2 at 10 μM (1% DyLight488-labeling) immediately underwent LLPS to form liquid droplets in the presence of 3% polyethylene glycol (PEG), which is often used to mimic an intracellular crowding environment in vitro (Fig. 1A). Conversely, recombinant α-syn at 10 μM (1% DyLight633-labeling) did not undergo LLPS in the presence of 3% PEG for up to 20 days (Appendix Fig. S1A). When α-syn was mixed with UBQLN2 in the presence of 3% PEG, α-syn was immediately incorporated into UBQLN2 droplets (Figs. 1A and EV1A). However, a deletion mutant lacking the UBL domain of UBQLN2 formed liquid droplets that failed to incorporate α-syn at 24 h (Fig. EV1B). Recently, it was reported that tau, a protein involved in Alzheimer's disease, can also form dynamic liquid droplets in vitro under physiologically relevant conditions (Kanaan et al, 2020). In our study, we found that α-syn was also incorporated into tau droplets under comparable conditions (Fig. 1B).

Intriguingly, smaller UBQLN2/α-syn droplets gradually disappeared, giving rise to larger droplets that progressively lost their spherical morphology (Fig. 1A). This time-dependent transition of liquid droplets into more stable states is referred to as the liquid–gel/solid transition (Kaganovich, 2017; Noda et al, 2020). To explore whether the loss of spherical shape is caused by the liquid–gel/solid transition of UBQLN2/α-syn droplets, we used 10% 1,6-hexanediol (1,6-HD), which is a hydrophobic reagent that inhibits liquid droplet formation (Babinchak et al, 2019). At 24 h, UBQLN2/α-syn droplets dissolved following the addition of 1,6-HD. By contrast, after 96 h, these droplets were resistant to 1,6-HD, suggesting that the UBQLN2/α-syn droplets had transitioned from a liquid to a gel/solid state (Fig. EV1C). Furthermore, although α-syn was also incorporated into UBQLN1 and UBQLN4 droplets

(Fig. 1A), these droplets maintained their spherical shape for up to 96 h (Fig. 1A). UBQLN1/α-syn droplets were dissolved in 1,6-HD at both 24 and 96 h, whereas UBQLN4/α-syn droplets were resistant to 1,6-HD at both 24 and 96 h (Fig. EV1C).

## α-Syn aggregation is synchronized with the liquid−gel/ solid transition of UBQLN2 droplets

Fluorescence recovery after photobleaching (FRAP), which characterizes the fluidity of proteins, revealed that the fluorescence recovery rate of UBQLN2 in the droplets was much lower at 96 h than at 24 h (Fig. 1C(a); Appendix Fig. S1B). Similarly, the fluorescence recovery rate of α-syn inside UBQLN2 droplets was much lower at 96 h than at 24 h (Fig. 1C(a); Appendix Fig. S1B). These data suggest that α-syn dynamics decrease alongside the solidification of UBQLN2 liquid droplets. The fluorescence recovery rate of UBQLN1 was slightly greater than that of UBQLN2 at 24 h, and this fast recovery rate was maintained even at 96 h (Fig. 1C(b); Appendix Fig. S1B). By contrast, the fluorescence recovery rate of UBQLN4 was significantly lower than that of UBQLN2 at both 24 and 96 h (Fig. 1C(c); Appendix Fig. S1B). These results are consistent with previous findings indicating that among the three UBQLNs, UBQLN1 is the most liquid-like and UBQLN4 is the most aggregation-prone (Gerson et al, 2021). Interestingly, incorporation of α-syn into any UBQLN droplets did not alter the intrinsic phase behavior of these UBQLNs (Fig. EV1D). Although the fluorescence recovery rate of α-syn was similar among UBQLN1, UBQLN2, and UBQLN4 droplets at 24 h, it was much lower in the UBQLN2 droplets only at 96 h (Fig. 1C; Appendix Fig. S1B), suggesting that α-syn undergoes aggregation and solidification within UBQLN2 droplets over time.

Fibrillar species reportedly emerge from condensates upon aberrant amyloidogenic solidification (Babinchak et al, 2019; Patel et al, 2015; Ray et al, 2020). Similarly, we revealed that fibrillar-like species of α-syn and UBQLN2 emerged from subsets of gel/solid-state UBQLN2/α-syn droplets after 96 h of LLPS induction (Fig. 1D). Fibrillar α-syn was also detected using transmission electron microscopy. After 48 h of LLPS induction, small amyloid-like fibrils emerged from the subsets of UBQLN2/α-syn droplets and transformed into larger filaments by 72 and 96 h (Fig. 1E). Immunoelectron microscopy revealed that the amyloid-like fibrils that had formed at 96 h were positive for α-syn (Fig. 1E). To confirm that the gold colloid signals detected during α-syn immunostaining were not due to nonspecific binding, a UBQLN2-only sample was included as a negative control. No gold colloid signal was observed in this control. Taken together, these results suggest that α-syn undergoes aggregation (the formation of oligomers and fibrils) specifically inside UBQLN2 droplets, and indicate that the liquid−gel/solid transition of droplets occurs at concentrations similar to those estimated in vivo (Dao et al, 2018; Piroska et al, 2023).

## α-Syn directly interacts with UBQLN2

Given that α-syn was incorporated into the UBQLN2 droplets in vitro, we next examined whether α-syn can directly interact with UBQLN2. Recombinant α-syn-FLAG and hemagglutinin (HA)-UBQLN2 were incubated together, and HA-UBQLN2 bound to α-syn-FLAG was pulled down using anti-FLAG beads and detected by

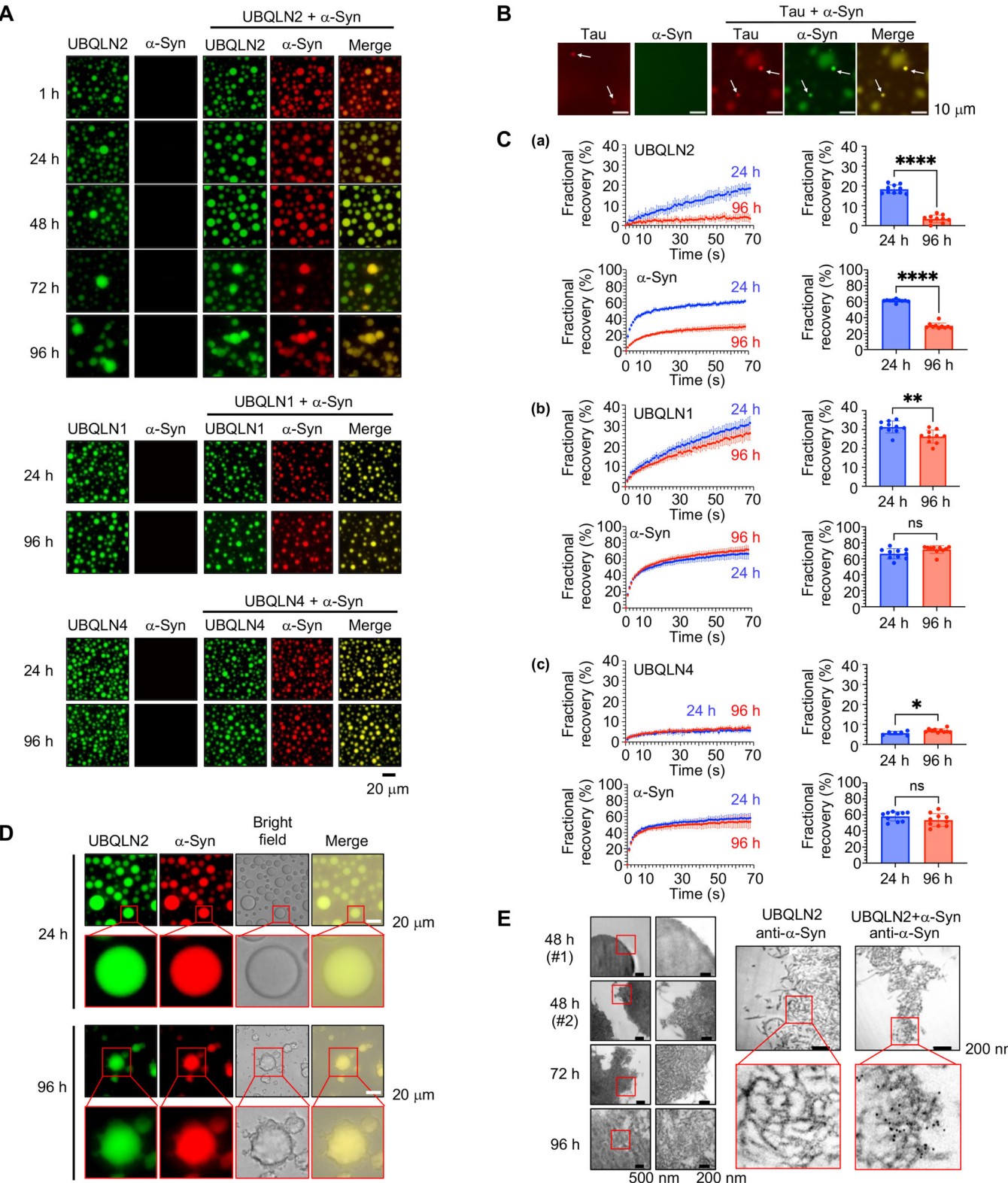

**Figure 1.** α-Synuclein (α-syn) aggregates within ubiquilin (UBQLN)2 droplets in vitro.

(A) Fluorescence microscopy images of individual solutions of 10 µM of each UBQLN construct (1% DyLight488-labeling) and 10 µM α-syn construct (1% DyLight633-labeling), as well as mixed solutions of each UBQLN and α-syn. At least three experiments were replicated. (B) Fluorescence microscopy images of individual solutions of 10 µM of tau (5% DyLight633-labeling) and 10 µM α-syn (1% DyLight488-labeling), as well as mixed solutions of tau and α-syn after incubation at 37 °C for 5 h. White arrows indicate liquid droplets. (C) Fluorescence recovery after photobleaching (FRAP) analysis of droplets of UBQLN2 (a), UBQLN1 (b), and UBQLN4 (c) incorporating α-syn. Plots show the average FRAP recovery curves from ten separate droplets. Values are the means ± standard deviations (SDs) (left). Quantification of fractional recovery at 70 s after photobleaching from ten separate droplets. *$P < 0.05$, **$P < 0.01$, ****$P < 0.0001$, N.S., nonsignificant (Welch's $t$ test). Values are the means ± SDs (Right). (a) UBQLN2: $P < 0.0001$, α-syn: $P < 0.0001$. (b) UBQLN1: $P = 0.0032$, α-syn: $P = 0.0516$. (c) UBQLN4: $P = 0.0244$, α-syn: $P = 0.1544$. (D) Fluorescence microscopy images of mixed solutions of 10 µM UBQLN2 construct (1% DyLight488-labeling) and 10 µM α-syn construct (1% DyLight633-labeling) after incubation at 37 °C for 24 and 96 h. At least three experiments were replicated. (E) Transmission electron microscopy images of UBQLN2 droplets incorporating α-syn at 48, 72, and 96 h after UBQLN2 liquid–liquid phase separation (LLPS) induction. Immunoelectron microscopy images of UBQLN2 droplets incorporating α-syn at 96 h after UBQLN2 LLPS induction using immuno-gold-labeled α-syn (6-nm gold particles). Source data are available online for this figure.

western blotting using an anti-UBQLN2 antibody. Consistent with a previous finding indicating that UBQLN2 interacts with α-syn in vivo, we confirmed that α-syn directly binds to UBQLN2 in vitro (Fig. EV2A; Appendix Fig. S2). Similarly, α-syn also directly binds to UBQLN1 and UBQLN4 (Fig. EV2A; Appendix Fig. S2).

To determine the region of UBQLN2 that is required for α-syn binding, the binding abilities of a series of UBQLN2 mutants with successive N- or C-terminal deletions to α-syn were evaluated using far-western blotting. UBQLN2 mutants lacking the STI1-2 region failed to bind to α-syn (Fig. EV2B–D), and UBQLN2 mutants with both the UBL domain and STI1-1 bound α-syn, while those with only the UBL domain did not (Fig. EV2B–D). These results indicate that α-syn binds to the STI1 regions of UBQLN2. Furthermore, UBQLN2 mutants lacking the STI1-2 region showed a greater decrease in α-syn binding than those lacking the STI1-1 region (Fig. 2A), indicating that STI1-2 is the primary binding site for α-syn.

We next studied the interaction between UBQLN2 and α-syn via high-speed atomic force microscopy (HS-AFM), which can visualize intrinsically disordered proteins and even liquid droplets with much higher resolution than fluorescence microscopy (Fujioka et al, 2020; Yamasaki et al, 2020). Unexpectedly, HS-AFM imaging of the α-syn–UBQLN2 mixture at 1 µM revealed a complex maze of higher-order condensates covering approximately half of the mica surface (Fig. 2B). These condensates appeared to follow a pattern of spinodal decomposition—a type of LLPS caused by concentration fluctuations rather than nucleation—which may be promoted by an increase in the local protein concentration on the negatively charged mica; the physiological relevance of this change is discussed in the Discussion. This pattern was not observed for either UBQLN2 or α-syn alone, indicating that it is formed via a direct UBQLN2–α-syn interaction (Fig. 2B).

## α-Syn aggregation is facilitated by UBQLN2 droplets in cell culture

We next examined whether the UBQLN2 LLPS-mediated α-syn aggregation observed in vitro also occurs in cultured cells. UBQLN2 was reported to undergo LLPS and form droplets when cells are treated with stressors such as arsenite (i.e., AsNaO$_2$ oxidative stress) (Dao et al, 2018). Indeed, we observed that UBQLN2 formed small droplets at 3 h following AsNaO$_2$ (50 µM) treatment in SH-SY5Y cells stably expressing wild type (WT) α-syn-enhanced green fluorescent protein (EGFP) (α-syn(WT)-EGFP/SH-SY5Y cells); over time, these droplets grew in size (Fig. 3A). In synchrony with

the formation of UBQLN2 droplets, α-syn-EGFP condensates appeared 3 h after AsNaO$_2$ treatment and were almost entirely co-localized with UBQLN2 droplets (Fig. 3A), indicating that α-syn is also incorporated into UBQLN2 droplets in cultured cells. Following 50 µM AsNaO$_2$ treatment, stress granules (SGs) were formed after 3 h, as indicated by the presence of puncta positive for the SG markers G3BP (Fig. 3B). Most of these SGs were co-localized with UBQLN2, consistent with previous reports (Alexander et al, 2018; Dao et al, 2018). Although the G3BP-positive puncta disappeared 6 h after AsNaO$_2$ treatment, droplets of UBQLN2 containing α-syn remained and continued to grow over time (Fig. 3B). To directly observe the structure of the large aggregates containing α-syn, we performed immunoelectron microscopy analysis. α-syn(WT)-EGFP/SH-SY5Y cells were fixed after 12 h of AsNaO$_2$ treatment and immunolabeled with anti-α-syn antibody and colloidal golds. We confirmed that gold colloids labeled the surface of the aggregate structure corresponding to α-syn(WT)-EGFP puncta (Fig. 3C).

In patients with sporadic PD and in some familial PD patients, α-syn is extensively phosphorylated at Ser129 in synucleinopathic lesions; in vitro, this phosphorylation promotes α-syn fibril formation, which is a hallmark of aggregated α-syn (Anderson et al, 2006; Fujiwara et al, 2002). Moreover, α-syn aggregates reportedly exhibit sarkosyl insolubility (Ikeda et al, 2019; Ko et al, 2008). Therefore, we examined the α-syn phosphorylation levels in the sarkosyl-insoluble fraction prepared from α-syn(WT)-EGFP/SH-SY5Y cells at 0, 3, 6, and 12 h after AsNaO$_2$ treatment. As shown in Fig. 3D, the amount of Ser129-phosphorylated α-syn in the sarkosyl-insoluble fraction was increased in a time-dependent manner, indicating that α-syn aggregates inside UBQLN2 droplets over time (Fig. 3D; Appendix Fig. S3A).

Together, our findings suggest that α-syn is incorporated into UBQLN2 droplets formed in response to oxidative stress and undergoes aggregation during the liquid–gel/solid transition of UBQLN2 droplets in cultured cells.

Furthermore, when FLAG-UBQLN2 was transiently overexpressed in α-syn(WT)-EGFP/SH-SY5Y cells, UBQLN2 formed puncta, as detected by anti-FLAG staining. Notably, α-syn also formed puncta that co-localized with those of UBQLN2 (Fig. EV3A(a)). In addition, FRAP analysis revealed that the fluorescence recovery rate of these α-syn puncta at 24 h after transfection was significantly lower than that of non-punctate α-syn (Fig. EV3A(b,c)), suggesting that overexpressed UBQLN2 sequesters α-syn and promotes α-syn condensate formation even in the absence of AsNaO$_2$-induced stress.

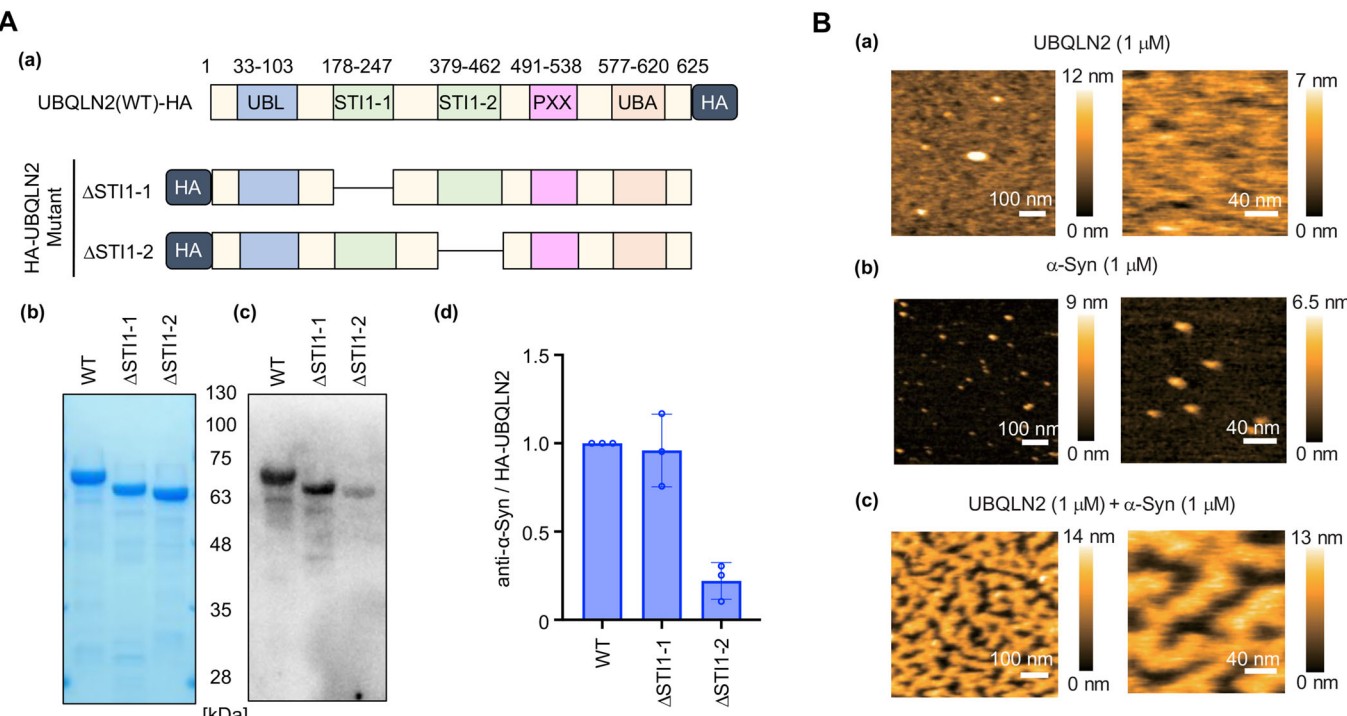

**Figure 2. α-Syn directly interacts with UBQLN.**

(A) (a) Schematic representations of the ΔSTI1-1/UBQLN2 and ΔSTI1-2/UBQLN2 mutants. (b) Representative SDS-PAGE images of each HA-tagged UBQLN2 mutant stained with Coomassie Brilliant Blue (CBB). (c) Representative far-western blotting results showing α-syn binding to each UBQLN2 mutant. After SDS-PAGE and membrane transfer, the membranes were incubated with recombinant His-tagged α-syn and probed with an anti-α-syn antibody. (d) Quantification of α-syn binding to each UBQLN2 mutant. The signal from far-western blotting was normalized to the amount of HA-UBQLN2 protein (CBB staining). Graphs show the mean ± SD from three independent experiments, corresponding to the representative results shown in (b, c). (B) (a) High-speed atomic force microscopy (HS-AFM) images of UBQLN2 covering a mica surface. UBQLN2 (1 μM) was applied to mica. (b) HS-AFM images of α-syn proteins. α-Syn (1 μM) was applied to mica. (c) HS-AFM images of spinodal decomposition-like patterns caused by UBQLN2 and α-syn. A mixture of UBQLN2 (1 μM) and α-syn (1 μM) was applied to mica. Source data are available online for this figure.

To confirm whether UBQLN2 is indeed responsible for AsNaO₂-induced α-syn aggregation in cultured cells, we constructed UBQLN-knockout (KO) SH-SY5Y cells expressing pathogenic α-syn(A53T)-EGFP (Control-KO, UBQLN2-KO, and UBQLN1-KO cells) and monitored α-syn condensate formation. Larger α-syn condensates that grew over time were observed upon AsNaO₂ treatment in Control-KO and UBQLN1-KO cells, but not in UBQLN2-KO cells (Fig. EV3B; Appendix Fig. S3B). Because we failed to establish UBQLN4-KO cells, the involvement of UBQLN4 in AsNaO₂-induced α-syn condensation was studied via small interfering RNA (siRNA)-mediated knockdown experiments. Although AsNaO₂ triggered the formation of larger α-syn condensates in cells with UBQLN1 or UNQLN4 knockdown, only small α-syn condensates were observed when UBQLN2 was silenced (Fig. 3E). However, UBQLN2 knockout did not affect the formation of G3BP1- and eIF4G1-positive SG puncta after 3 h of AsNaO₂ treatment in our cell system (Fig. EV3C; Appendix Fig. S3C). We next examined α-syn phosphorylation levels in the sarkosyl-insoluble fraction prepared from each UBQLN-KO or -knockdown cell line 12 h after AsNaO₂ treatment. The amount of Ser129-phosphorylated α-syn in the sarkosyl-insoluble fraction was increased upon AsNaO₂ treatment in the Control-KO, UBQLN1-

KO, and UBQLN4-knockdown cells, but not in the UBQLN2-KO cells (Fig. 3F; Appendix Fig. S3D). Collectively, these results suggest that UBQLN2 is indispensable for AsNaO₂-induced α-syn aggregation in a cultured cell system.

When α-syn(WT)-EGFP/SH-SY5Y cells were treated with puromycin, which induces the LLPS of UBQLN2 (Dao et al, 2018), the formation of larger α-syn condensates was observed over time (Fig. EV3D). FRAP analysis revealed that the fluorescence recovery rate of these larger α-syn condensates at 8 h after puromycin treatment was significantly lower than that of non-condensates of α-syn (Fig. EV3D). In contrast, puromycin did not induce SG formation for up to 8 h, even under conditions in which UBQLN2 formed large puncta (Fig. EV3E). UBQLN1 and UBQLN4 were also detected in larger α-syn condensates (Fig. EV3D). However, these puromycin-induced larger α-syn condensates were still observed in cells with UBQLN1 or UBQLN4 knockdown, but not in cells with UBQLN2 knockdown (Fig. EV3F). These results indicate that UBQLN2, but not UBQLN1 or UBQLN4, is responsible for the formation of large α-syn condensates in cultured cells in response to various cellular stresses.

We next examined whether UBQLN2 inclusions could be detected in Lewy bodies (LBs) identified in brain autopsies of PD patients

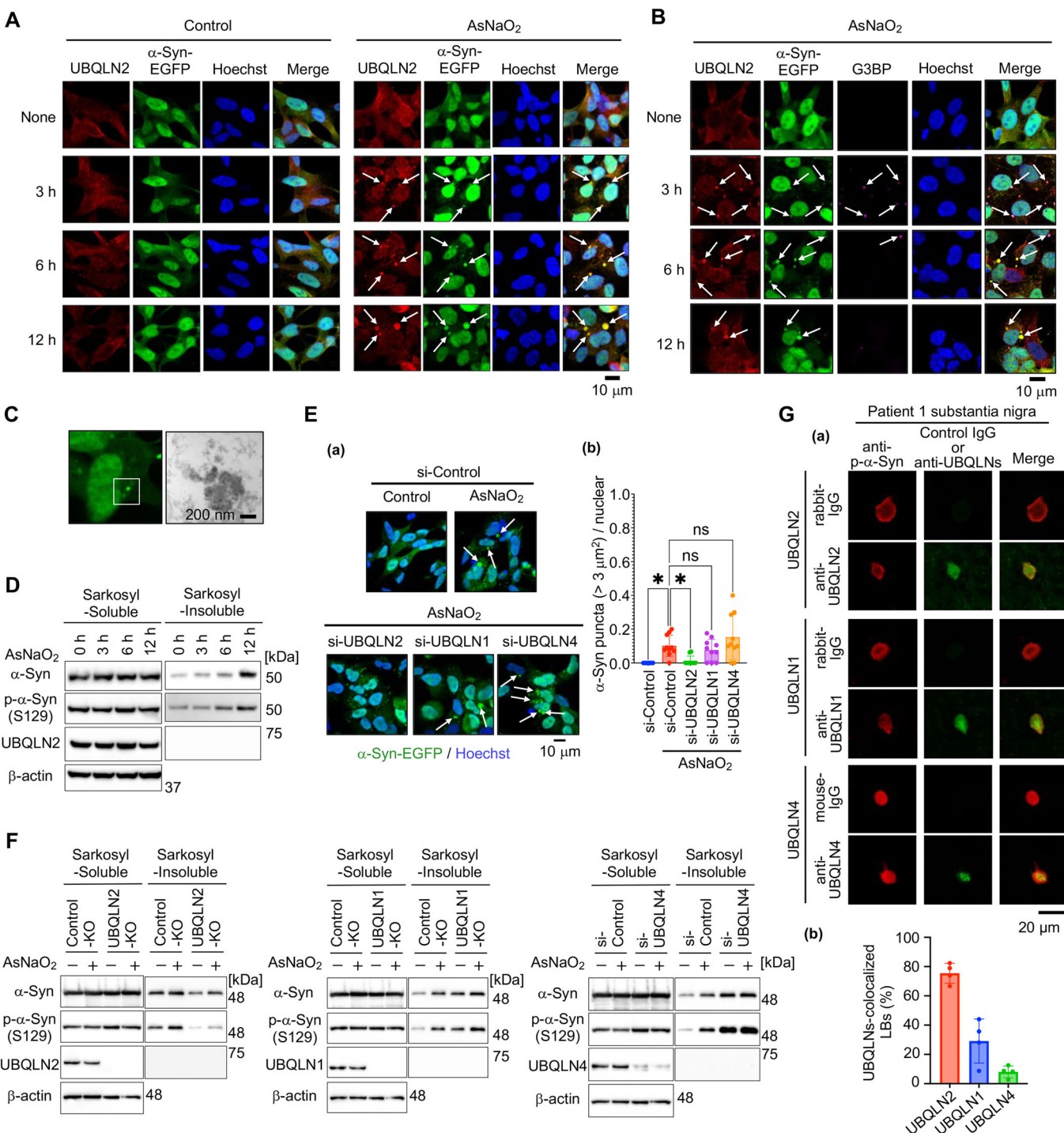

(Appendix Table S1). As shown in Figs. 3G and EV3G, UBQLN2 immunoreactivity was clearly observed in LBs in the substantia nigra of four sporadic PD patients, whereas background signal was barely detectable in the control IgG. These results suggest that UBQLN2 may be pathologically involved in α-syn aggregation. UBQLN4, albeit very rarely, and UBQLN1 were detected in some α-syn-positive inclusions, though less frequently than UBQLN2, possibly owing to the ability of UBQLN2 to form heterodimers with UBQLN1 and UBQLN4 (Lee et al, 2013; Zheng et al, 2020).

## Liquid−gel/solid transition of UBQLN2 droplets is inhibited by the UBQLN2-binding compound SO286

The compound 1,2,3,6-tetra-O-benzoyl-*muco*-inositol (SO286) (Ogawa et al, 1979) (Fig. 4A) was originally identified through a neuroprotective compound screen of our in-house chemical library. Using biotinylated SO286 and its inactive analog SO82, we found that SO286—but not SO82—directly bound to UBQLN1, UBQLN2, and UBQLN4 (Fig. 4A,B).

◀ **Figure 3. α-Syn aggregates within UBQLN2 droplets in cultured cells.**

(A) α-Syn(WT)-EGFP/SH-SY5Y cells were treated with 50 μM AsNaO₂ for 3, 6, and 12 h and immunostained with anti-UBQLN2 and Hoechst. White arrows indicate α-syn-incorporated UBQLN2 droplets. (B) α-Syn(WT)-EGFP/SH-SY5Y cells were treated with 50 μM AsNaO₂ for 3, 6, and 12 h and immunostained with anti-UBQLN2, anti-G3BP, and Hoechst. At least three experiments were replicated. White arrows indicate α-syn-incorporated UBQLN2 droplets. (C) α-Syn(WT)-EGFP/SH-SY5Y cells were treated with 50 μM AsNaO₂ for 12 h. Immunoelectron microscopy image of large α-syn puncta using immuno-gold-labeled α-syn (6-nm gold particles). (D) α-Syn(WT)-EGFP/SH-SY5Y cells were treated with 50 μM AsNaO₂ for 3, 6, and 12 h and sarkosyl-soluble and -insoluble fractions were analyzed by SDS-PAGE and immunoblotting. Representative blot shown, n = 2. (E) (a) Fluorescence microscopy of α-syn(WT)-EGFP/SH-SY5Y cells with UBQLN knockdown. Cells were transfected with si-Control, si-UBQLN1/2/4, treated with 50 μM AsNaO₂ for 12 h post 48 h transfection, and stained with Hoechst. White arrows indicate larger α-syn-EGFP condensates. (b) Quantification of the number of α-syn-EGFP condensates (> 3 μm²) per cell. *P < 0.05, N.S., nonsignificant (Dunnett's test, versus si-Control/AsNaO₂). Ten images were quantified per condition. Values are the means ± SDs. si-Control (AsNaO₂) vs si-Control: P = 0.0113, si-Control(AsNaO₂) vs si-UBQLN2(AsNaO₂): P = 0.0319, si-Control(AsNaO₂) vs si-UBQLN1(AsNaO₂): P = 0.8626, si-Control(AsNaO₂) vs si-UBQLN4(AsNaO₂): P = 0.3245. (F) UBQLN2-KO, UBQLN1-KO, or si-UBQLN4-transfected α-syn(A53T)-EGFP/SH-SY5Y cells were treated with 50 μM AsNaO₂ for 12 h. Sarkosyl-soluble and -insoluble fractions were analyzed by SDS-PAGE and immunoblotting. Representative blot shown, n = 2. (G) (a) Accumulation of UBQLN2 in LBs of sporadic PD patients. (a) Sections from the midbrain of pathologically diagnosed PD cases were stained with each anti-UBQLN antibodies or anti-p-α-syn antibody. As negative controls, the corresponding control IgG (rabbit or mouse, at the same concentrations) was used together with the anti-p-α-syn antibody. (b) The percentage of phospho-α-syn-positive LBs that co-localized with each UBQLN was quantified by confocal microscopic observation. At least 15 LBs from the autopsy brains of four patients were evaluated. Source data are available online for this figure.

Furthermore, surface plasmon resonance analysis revealed that SO286 selectively bound to UBQLN2 with an affinity ($K_d$) of 0.351 μM, whereas its affinities for UBQLN1 and UBQLN4 were 9.49 μM and 17.0 μM, respectively (Figs. 4C and EV4A). An interaction analysis between biotin–SO286 and UBQLN2 truncation mutants using surface plasmon resonance identified STI1-1 and STI1-2 as the binding sites for SO286 (Fig. 4D). The deletion of either STI1-1 or STI1-2 did not reduce the affinity of UBQLN2 for SO286, indicating that these regions are redundantly involved in SO286 binding (Fig. 4E). We then investigated whether the binding of SO286 to UBQLN2 affects the formation of liquid droplets of UBQLN2. Recombinant UBQLN2 protein (10 μM) was incubated with SO286 in the presence of PEG. After 24 h, the numbers and sizes of the UBQLN2 liquid droplets were not affected by SO286 treatment at concentrations up to 20 μM. Similar results were obtained when UBQLN1 and UBQLN4 were used instead of UBQLN2, indicating that SO286 did not affect the formation of liquid droplets of UBQLN1, UBQLN2, or UBQLN4 (Fig. EV4B).

Next, we examined the effects of SO286 on the liquid−gel/solid transition of UBQLN2. Treatment with SO286 suppressed the time-dependent spherical collapse of UBQLN2 droplets in a dose-dependent manner (Fig. 4F(a)). In contrast, SO82, an inactive analog of SO286, failed to inhibit the spherical collapse of UBQLN2 droplets (Fig. 4F(b)). Furthermore, FRAP experiments revealed that the time-dependent reduction in the fluorescence recovery rate of UBQLN2 droplets was inhibited by 20 μM SO286 treatment (Fig. 4G; Appendix Fig. S4). These data suggest that SO286 inhibits the liquid−gel/solid transition of UBQLN2 droplets.

Because SO286 binds to STI1 regions that mediate the liquid−gel/solid transition of UBQLN2 droplets by promoting self-interaction and oligomerization (Dao et al, 2018), we speculated that SO286 may inhibit the liquid−gel/solid transition of UBQLN2 droplets by competitively inhibiting the STI1-mediated self-interaction of UBQLN2. We therefore examined the effects of SO286 on the interaction between immunoprecipitated FLAG-UBQLN2 and recombinant HA-UBQLN2. This interaction was inhibited by SO286 in a dose-dependent manner, but not by SO82 (Fig. 4H). Together, these findings suggest that SO286 inhibits the liquid−gel/solid transition of UBQLN2 droplets by inhibiting the STI1-mediated self-interaction of UBQLN2.

## UBQLN2-mediated α-syn aggregation is inhibited by SO286

In addition to inhibiting UBQLN2 self-interaction, SO286—but not SO82—also inhibited the interaction between α-syn and UBQLN2 in a dose-dependent manner (Fig. 5A). We therefore investigated the effects of SO286 on the formation of UBQLN2/α-syn droplets. When SO286 was added to UBQLN2/α-syn droplets at concentrations of up to 20 μM, no changes in the formation of UBQLN2/α-syn droplets were detected after 1 h using fluorescence microscopy (Fig. 5B; Appendix Fig. S5A). Intriguingly, SO286 treatment specifically inhibited the spinodal decomposition of the UBQLN2/α-syn mixture, but not that formed by α-syn and UBQLN1 or UBQLN4 (Figs. 5C and EV5A). These data suggest that the α-syn−UBQLN2 interaction is crucial for LLPS via the spinodal decomposition mechanism and indicate that SO286 inhibits this mode of LLPS, presumably by inhibiting the α-syn−UBQLN2 interaction.

Because SO286 inhibited UBQLN2–UBQLN2 and UBQLN2–α-syn interactions as well as the time-dependent solidification of UBQLN2 liquid droplets (Figs. 4F–H and 5A), we examined whether SO286 inhibits the aggregation of α-syn inside UBQLN2 liquid droplets. SO286 (20 μM) did not inhibit the incorporation of α-syn into UBQLN2 liquid droplets; however, it suppressed the time-dependent spherical collapse of α-syn-incorporated UBQLN2 droplets (Fig. 5B). In contrast, SO82 failed to inhibit the spherical collapse of α-syn-incorporated UBQLN2 droplets (Fig. EV5B). Moreover, FRAP analysis revealed that SO286 (20 μM) suppressed the decrease in the fluorescence recovery rate of both α-syn and UBQLN2 at 96 h in UBQLN2/α-syn droplets (Fig. 5D; Appendix Fig. S5B), whereas SO82 (20 μM) failed to suppress this decrease (Fig. EV5C). Furthermore, UBQLN2/α-syn fibrillar species emerging from the droplets were not observed upon SO286 treatment for up to 96 h (Fig. 5E). Together, these results suggest that SO286 suppresses α-syn aggregation inside UBQLN2 droplets.

Similarly, tau protein at 8 μM (1% DyLight633-labeled) did not undergo LLPS under the conditions in which UBQLN2 formed droplets (3% PEG); however, it was immediately incorporated into UBQLN2 droplets (Fig. 5F(a)). Furthermore, tau aggregated and solidified within UBQLN2 droplets after 96 h, as assessed by FRAP analysis. This tau aggregation in UBQLN2 droplet was inhibited by SO286 (Fig. 5F(b,c)).

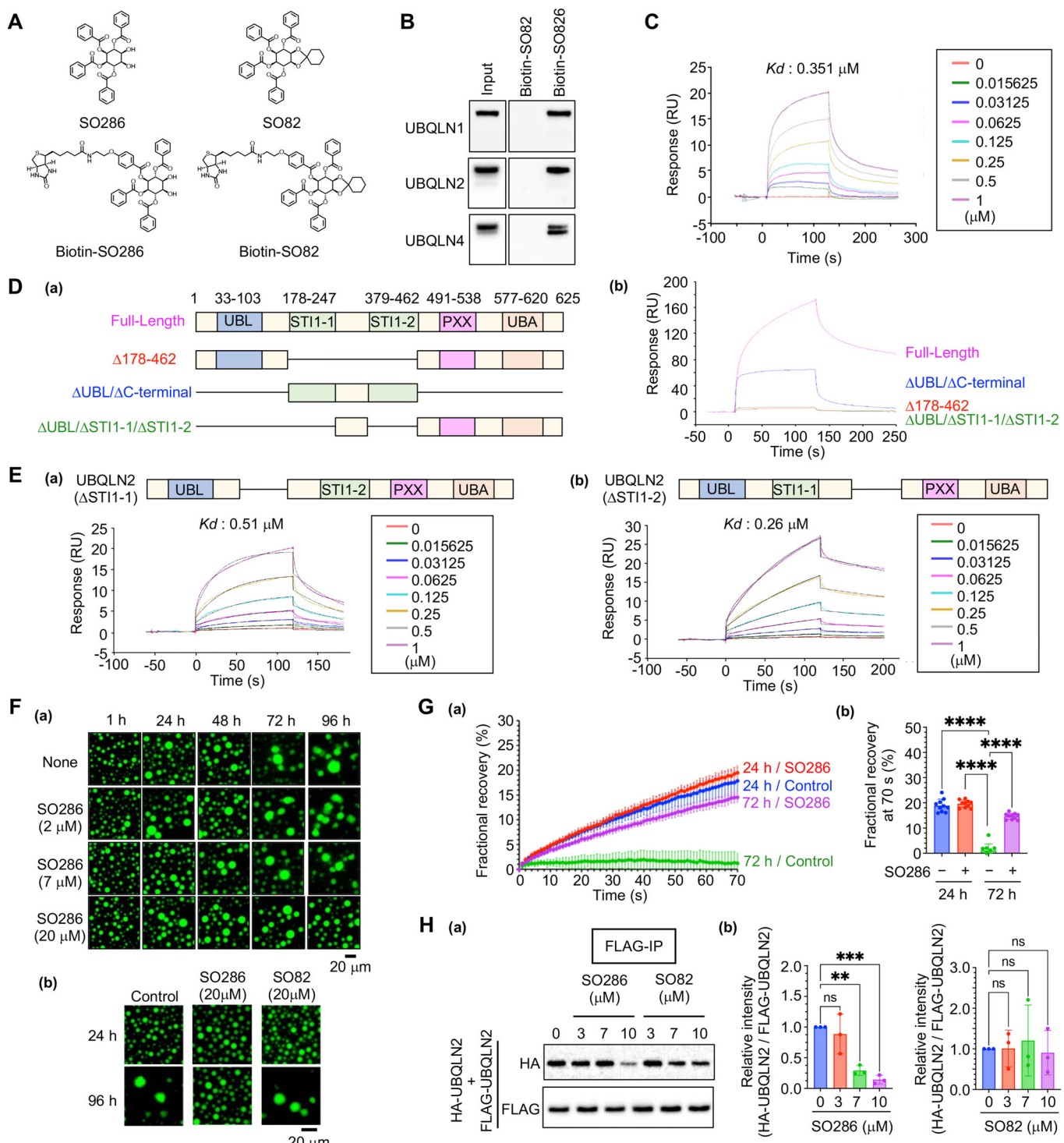

## SO286 inhibits UBQLN2-mediated α-syn aggregation in cultured cells

Next, we examined whether SO286 could inhibit UBQLN2 LLPS-mediated α-syn aggregation in cultured cells. SO286 (20 μM) did not inhibit AsNaO$_2$-induced UBQLN2 droplet formation or the incorporation of α-syn into UBQLN2 droplets. However, it

suppressed the time-dependent growth of UBQLN2/α-syn droplets in AsNaO$_2$-treated α-syn(WT)-EGFP/SH-SY5Y cells (Fig. 6A). In contrast, SO82, an inactive analog, did not suppress this growth (Fig. EV6A). Moreover, Thioflavin S (ThS) staining revealed that SO286 inhibited protein aggregation within UBQLN2 droplets (Fig. 6B). SO286 also reduced the amount of α-syn and Ser129-phosphorylated α-syn in sarkosyl-insoluble fractions prepared from

◄

**Figure 4.  SO286 binds to the STI1 regions of UBQLN2.**

(A) Structures of SO286 (upper-left), SO82 (upper-right), biotin–SO286 (bottom-left) and biotin-SO82 (bottom-right). (B) Biotin–SO286 and biotin-SO82 were mixed with whole cell lysate of SH-SY5Y cells. These compounds were pull-downed by streptavidin beads and eluted by an excess of biotin. Elute solutions were immunoblotted with the indicated antibodies. (C) Surface plasmon resonance (SPR) analysis of biotin–SO286 reactivity with UBQLN2. The vertical axis represents the amount of binding in the resonance unit (RU). (D) (a) Illustration of the domain deletion mutants of UBQLN2. (b) SPR analysis of biotin–SO286 reactivity with the UBQLN2 deletion mutants. The vertical axis represents the amount of binding in the resonance unit (RU). (E) SPR analysis of biotin–SO286 reactivity with UBQLN2 (ΔSTI1-1) (a) and UBQLN2 (ΔSTI1-2) (b). The vertical axis represents the amount of binding in the RU. (F) (a, b) Fluorescence microscopy images show solutions of the 10 μM UBQLN2 construct (1% DyLight488-labeling) after incubation at 37 °C for the indicated hours in the presence or absence of SO286 or SO82. At least three experiments were replicated. (G) FRAP of UBQLN2 droplets in the presence of SO286. (a) Plots display average FRAP recovery curves from ten separate droplets. Values are the means ± SDs. (b) Quantification of fractional recovery at 70 s after photobleaching from ten separate droplets. ****$P < 0.0001$ (Dunnett's test, versus 72 h/Control). Values are the means ± SDs. 72 h/Control vs 24 h/Control: $P < 0.0001$, 72 h/Control vs 24 h/SO286: $P < 0.0001$, 72 h/Control vs 72 h/SO286: $P < 0.0001$. (H) (a) Immunoprecipitated FLAG-UBQLN2 and HA-UBQLN2 were incubated with SO286 or SO82. FLAG-UBQLN2 was pulled down with anti-FLAG beads, eluted with FLAG peptide, and immunoblotted. At least three experiments were replicated. (b) Quantification of the ratio of the intensity of HA-UBQLN2 to that of FLAG-UBQLN2. **$P < 0.01$, N.S., nonsignificant (Dunnett's test, versus SO286 [0 μM] or SO82 [0 μM]). Values are the means ± SDs. SO286 [0 μM] vs SO286 [3 μM]: $P = 0.7573$, SO286 [0 μM] vs SO286 [7 μM]: $P = 0.0024$, SO286 [0 μM] vs SO286 [20 μM]: $P = 0.0007$, SO82 [0 μM] vs SO82 [3 μM]: $P > 0.9999$, SO82 [0 μM] vs SO82 [7 μM]: $P = 0.9433$, SO82 [0 μM] vs SO82 [7 μM]: $P = 0.9934$. Source data are available online for this figure.

AsNaO$_2$-treated α-syn(WT)-EGFP/SH-SY5Y cells at 12 h (Fig. 6C; Appendix Fig. S6), and similar results were obtained when α-syn(A53T)-EGFP was expressed in SH-SY5Y cells (Fig. EV6B; Appendix Fig. S6). Collectively, our findings indicate that SO286 inhibited α-syn aggregation by inhibiting both the α-syn–UBQLN2 interaction and the liquid–gel/solid phase transition of UBQLN2 droplets.

## Discussion

To date, neurodegenerative disease-associated proteins, including α-syn, are believed to form fibers on their own via their ability to undergo LLPS and the subsequent liquid−gel/solid transition (Hardenberg et al, 2021; Mukherjee et al, 2024; Mukherjee et al, 2023; Piroska et al, 2023; Poudyal et al, 2022; Ray et al, 2020; Sawner et al, 2021; Xu et al, 2022). The present findings indicate that UBQLN2 LLPS catalyzes the fibril formation of α-syn under physiological conditions that do not cause LLPS and self-fibrillation of α-syn alone. This role of UBQLN2 is conceptually distinct from its previously reported cytoprotective functions involving the proteasome or autophagy systems for the degradation of ubiquitinated structures (Kleijnen et al, 2000; Ko et al, 2004; Lee et al, 2013; Sandoval-Pistorius et al, 2023; Stieren et al, 2011; Valentino et al, 2024).

SGs are transient, membraneless organelles that form through LLPS in response to stresses such as oxidative stress or heat shock (Mathieu et al, 2020; Wolozin and Ivanov, 2019). In our system, SGs appeared and co-localized with UBQLN2/α-syn droplets 3 h after AsNaO$_2$ treatment but dissolved by 6 h, whereas UBQLN2/α-syn droplets persisted and matured into gel/solid assemblies that promoted α-syn aggregation. Puromycin, which triggered UBQLN2/α-syn droplet formation without inducing SGs, produced a similar outcome; namely, UBQLN2-catalyzed α-syn aggregation proceeds independently of SG biogenesis. Consistently, UBQLN2 knockout did not impair arsenite-induced SG formation, underscoring SG-independent pathways and highlighting that previous reports of UBQLN2 as a negative regulator of SGs under heat shock (Alexander et al, 2018) may reflect stress- or cell-type-specific differences.

Our findings indicate that the early step in UBQLN2 LLPS-catalyzed α-syn fibril formation is the incorporation of α-syn into UBQLN2 droplets, followed by the interaction between UBQLN2 and incorporated α-syn. Furthermore, our results suggest that although the UBL domain of UBQLN2 is not required for its direct interaction with α-syn, it is indispensable for initiating α-syn incorporation into condensates (Figs. EV1B and EV2A). A recent study suggests that client proteins partitioning –the process where certain proteins are selectively concentrated inside liquid droplets instead of staying in the surrounding solution–can be governed solely by intrinsic "stickiness" (Villegas and Levy, 2022); our findings imply that the UBL domain modulates the condensate's internal physicochemical environment, such as stickiness, to facilitate α-syn entry. Once recruited, α-syn binds primarily to the STI1-2, and to a lesser extent to the STI1-1, regions of UBQLN2, regions that also engage heat-shock protein 70 (Hsp70) and other quality control partners (Hjerpe et al, 2016; Ma et al, 2023; Zheng et al, 2021). Detailed structural elucidation of this interface remains an important objective.

For the interaction between α-syn and UBQLN2, the two STI1 regions of UBQLN2, particularly the STI1-2 region, serve as binding sites for α-syn. These STI1 regions of UBQLN2 also reportedly bind to other components, such as Hsp70, that are crucial for the protein quality control function of UBQLN2 (Hjerpe et al, 2016; Ma et al, 2023; Zheng et al, 2021). Nevertheless, the detailed binding mechanism between UBQLN2 and α-syn remains to be elucidated in future studies.

The self-association of α-syn through the hydrophobic face of the NAC domain, which is autoinhibited by the interaction between the N- and C-terminal domains, is important for LLPS and the subsequent liquid−gel/solid transition at high concentrations of α-syn alone in vitro (Sawner et al, 2021). The binding of α-syn to UBQLN2 may change the autoinhibitory conformation of α-syn within the UBQLN2 droplets, thereby promoting NAC domain-mediated self-association and the liquid−gel/solid transition into fibrils. Notably, the observation of α-syn–UBQLN2 LLPS via the spinodal decomposition mechanism on mica at physiologically low concentrations (1 μM) may mimic the initial LLPS process of α-syn–UBQLN2 droplets in specific cellular environments, such as in negatively charged membranes that affect α-syn LLPS. However, this mechanism is likely insufficient for α-syn fibrillation because UBQLN1 and UBQLN4, which undergo LLPS and interact with α-syn similarly to UBQLN2, cannot catalyze this reaction. This may be because of differences in the droplet dynamics of each UBQLN;

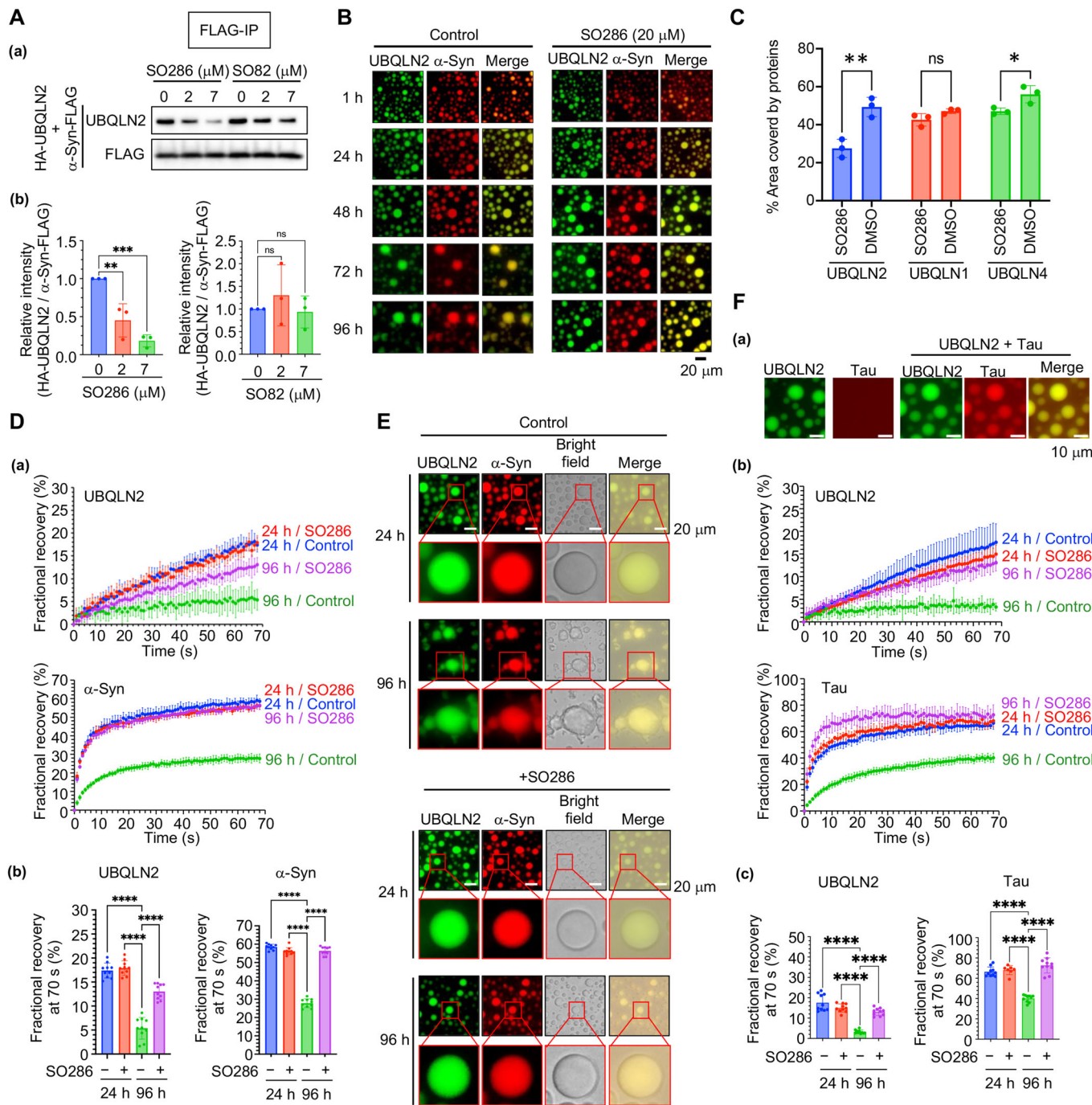

UBQLN1 and UBQLN4 droplets are constantly liquid-like and gel/solid-like, respectively, whereas UBQLN2 droplets can undergo a liquid−gel/solid transition. The ability of UBQLN2 to both undergo the liquid−gel/solid transition and bind to α-syn may be essential for catalyzing α-syn fibrillation.

To determine whether our cell-based findings are pathologically reflected in PD patient brains, we conducted immunohistochemical studies on the brain sections from patients with sporadic PD. Consistent with our findings in cells, UBQLN2 was detected in LBs

in the substantia nigra of four sporadic PD patients. There have been conflicting reports regarding the presence of UBQLN2 in α-syn inclusions in postmortem brains of patients with PD, with some studies detecting UBQLN2 and others reporting it to be extremely rare (Mori et al, 2012; Rutherford et al, 2013). However, in the present study, we detected UBQLN2 in LBs in the substantia nigra of four cases of sporadic PD, suggesting that our findings that UBQLN2-catalyzed formation of α-syn inclusion may be relevant to a clinically meaningful mechanism in PD pathogenesis, at least in some cases.

**Figure 5.    SO286 inhibits α-syn aggregation within UBQLN2 droplets in vitro.**

(A) (a) Immunoprecipitated α-syn-FLAG and recombinant HA-UBQLN2 were incubated with SO286 or SO82. α-Syn-FLAG was pulled down with anti-FLAG beads, eluted with the FLAG peptide, and immunoblotted. (b) Quantification of HA-UBQLN2 intensity. **$P < 0.01$, ***$P < 0.001$, N.S., nonsignificant (Dunnett's test, versus SO286 [0 μM] or SO82 [0 μM]). Values are the means ± SDs. At least three experiments were replicated. SO286 [0 μM] vs SO286 [2 μM]: $P = 0.0045$, SO286 [0 μM] vs SO286 [7 μM]: $P = 0.0006$, SO82 [0 μM] vs SO82 [2 μM]: $P = 0.6332$, SO82 [0 μM] vs SO82 [7 μM]: $P = 0.9761$. (B) Fluorescence microscopy images showing mixed solutions of 10 μM UBQLN2 construct (1% DyLight488-labeling) and 10 μM α-syn construct (1% DyLight633-labeling) in the presence of 20 μM SO286. (C) Each UBQLN (1 μM) was premixed with SO286 (0.25 mM) before being mixed with α-syn (1 μM). After high-speed AFM imaging, the resulting images (Fig. EV5A) were used to quantify the proportion of surface area occupied by the proteins (α-syn together with UBQLN1, UBQLN2, or UBQLN4). Values are calculated from HS-AFM images from three independent experiments. *$P < 0.05$, **$P < 0.01$, N.S., nonsignificant (Welch's $t$ test). UBQLN2: p = 0.0057, UBQLN1: $P = 0.1114$, UBQLN4: $P = 0.0500$. (D) FRAP analysis of UBQLN2 droplets with α-syn and SO286: (a) Plots display average FRAP recovery curves from ten separate droplets of UBQLN2 and α-syn. Values are the means ± SDs. (b) Quantification of fractional recovery at 70 s after photobleaching from ten separate droplets. ****$P < 0.0001$ (Dunnett's test, versus 96 h/Control). Values are the means ± SDs. UBQLN2; 96 h/Control vs 24 h/Control: $P < 0.0001$, 96 h/Control vs 24 h/SO286: $P < 0.0001$, 96 h/Control vs 96 h/SO286: $P < 0.0001$. α-syn; 96 h/Control vs 24 h/Control: $P < 0.0001$, 96 h/Control vs 24 h/SO286: $P < 0.0001$, 96 h/Control vs 96 h/SO286: $P < 0.0001$. (E) Fluorescence microscopy of UBQLN2/α-syn fibrils from droplets in LLPS buffer with 3% PEG and SO286 (10 μM each, 1% DyLight-labeled). (F) (a) Fluorescence microscopy images of individual solutions of 10 μM of UBQLN2 (1% DyLight488-labeling) and 8 μM tau (5% DyLight633-labeling), as well as mixed solutions of UBQLN2 and tau after incubation at 37 °C for 24 h. (b) FRAP analysis of droplets of UBQLN2 incorporating tau in the presence or absence of 20 μM SO286. (c) Quantification of fractional recovery at 70 s after photobleaching from ten separate droplets. ****$P < 0.0001$ (Dunnett's test, versus 96 h/Control). Values are the means ± SDs. UBQLN2; 96 h/Control vs 24 h/Control: $P < 0.0001$, 96 h/Control vs 24 h/SO286: $P < 0.0001$, 96 h/Control vs 96 h/SO286: $P < 0.0001$. Tau; 96 h/Control vs 24 h/Control: $P < 0.0001$, 96 h/Control vs 24 h/SO286: $P < 0.0001$, 96 h/Control vs 96 h/SO286: $P < 0.0001$. Source data are available online for this figure.

SO286—a compound that binds to the UBQLN2 STI1 regions, which contain the binding sites of α-syn to UBQLN2—inhibited α-syn fibrillation by disrupting both the UBQLN2 self-interaction (required for the liquid−gel/solid transition of droplets) and the α-syn–UBQLN2 interaction. Intriguingly, although SO286 inhibited the formation of large UBQLN2 droplets in cells, many factors likely affect UBQLN2 and α-syn LLPS in cells. Therefore, we are currently establishing cells derived from PD patients and an animal model system to evaluate the clinical potential of SO286.

Beyond α-syn, we showed that tau, whose fibrils characterize Alzheimer's disease, partitioned into UBQLN2 droplets and aggregated during their maturation, an effect that was also suppressed by SO286. These observations raise the possibility that UBQLN2 is a common catalyst for multiple aggregation-prone proteins, providing a mechanistic link to the frequent co-occurrence of α-syn and tau pathologies in PD (Cisbani et al, 2017; Smith et al, 2019) and to evidence that tau accelerates α-syn aggregation (Pan et al, 2022). The incorporation of α-syn into tau droplets (Fig. 1B) further supports cross-talk between these two pathologies.

Although further studies are required to elucidate the detailed mechanisms of α-syn fibrillation in vivo, our findings provide the concept of the UBQLN2 LLPS-catalyzed fibrillation of α-syn and demonstrate the effects of UBQLN2 LLPS modulation by an STI1 region-targeted compound in the context of neurodegenerative disease development.

It has been reported that α-syn interacts with vesicle-associated membrane protein 2 to form droplets with synaptic vesicles (Agarwal et al, 2024; Wang et al, 2024), and that β-syn delays the liquid−gel/solid transition of α-syn (Li et al, 2024). Both of these findings may also prevent α-syn pathological aggregation. By contrast, $Ca^{2+}$ influx-induced RNA G-quadruplex assembly has been reported to accelerate α-syn phase transition and aggregation (Matsuo et al, 2024). Together with our findings, these results suggest that α-syn aggregation is regulated by phase separation with different interacting proteins and RNAs and is dependent on stress and intracellular conditions. These findings suggest a new approach for the development of therapeutic interventions for PD.

# Methods

**Reagents and tools table**

| Reagent/resource | Reference or source | Identifier or catalog number |
|---|---|---|
| **Experimental models** | | |
| SH-SY5Y cells (H. sapiens) | ATCC | CRL-2266 |
| HEK293T cells (H. sapiens) | ATCC | CRL-3216 |
| SH-SY5Y cells stably expressing α-syn(WT)-EGFP | This study | N/A |
| SH-SY5Y cells stably expressing α-syn(A53T)-EGFP | This study | N/A |
| Control-KO SH-SY5Y cells stably expressing α-syn(A53T)-EGFP | This study | N/A |
| UBQLN1-KO SH-SY5Y cells stably expressing α-syn(A53T)-EGFP | This study | N/A |
| UBQLN2-KO SH-SY5Y cells stably expressing α-syn(A53T)-EGFP | This study | N/A |
| UBQLN1, UBQLN2 and UBQLN4 triple-KO T-REx293 cells | Itakura et al (2016) | N/A |
| **Recombinant DNA** | | |
| pGEX-6p-1 | Cytiva | 28954648 |
| pGEX-6p-1-UBQLN2 | This study | N/A |
| pGEX-6p-1-UBQLN2(Δ178-462) | This study | N/A |
| pGEX-6p-1-UBQLN2(ΔUBL/ΔC-Terminal) | This study | N/A |
| pGEX-6p-1-UBQLN2(ΔUBL/ΔSTI1-1/ΔSTI1-2) | This study | N/A |

| Reagent/resource | Reference or source | Identifier or catalog number |
|---|---|---|
| pGEX-6p-1-UBQLN2(ΔSTI1-1) | This study | N/A |
| pGEX-6p-1-UBQLN2(ΔSTI1-2) | This study | N/A |
| pGEX-6p-1-UBQLN2(ΔUBL) | This study | N/A |
| pGEX-6p-1-UBQLN1 | This study | N/A |
| pGEX-6p-1-UBQLN4 | This study | N/A |
| pGEX-6p-1-UBQLN2-HA | This study | N/A |
| pGEX-6p-1-HA-UBQLN2(ΔSTI1-1) | This study | N/A |
| pGEX-6p-1-HA-UBQLN2(ΔSTI1-2) | This study | N/A |
| pGEX-6p-1-ΔN-UBQLN2(#1)-HA | This study | N/A |
| pGEX-6p-1-ΔN-UBQLN2(#2)-HA | This study | N/A |
| pGEX-6p-1-ΔN-UBQLN2(#3)-HA | This study | N/A |
| pGEX-6p-1-ΔN-UBQLN2(#4)-HA | This study | N/A |
| pGEX-6p-1-ΔN-UBQLN2(#5)-HA | This study | N/A |
| pGEX-6p-1-ΔN-UBQLN2(#6)-HA | This study | N/A |
| pGEX-6p-1-ΔC-UBQLN2(#7)-HA | This study | N/A |
| pGEX-6p-1-ΔC-UBQLN2(#8)-HA | This study | N/A |
| pGEX-6p-1-ΔC-UBQLN2(#9)-HA | This study | N/A |
| pGEX-6p-1-ΔC-UBQLN2(#10)-HA | This study | N/A |
| pGEX-6p-1-ΔC-UBQLN2(#11)-HA | This study | N/A |
| pGEX-6p-1-α-syn | This study | N/A |
| pRSET-C-His-α-syn | This study | N/A |
| pcDNA3.1( + )-FLAG-UBQLN2 | This study | N/A |
| pcDNA3.1(+)-α-syn(WT)-FLAG | This study | N/A |
| pEGFP-N1-α-syn(WT) | This study | N/A |
| pEGFP-N1-α-syn(A53T) | This study | N/A |
| lentiCRISPRv2-blast plasmid | Addgene | 83480 |
| **Antibodies** | | |
| Anti-UBQLN1 | Cell Signaling Technology Inc. | #14526 |
| Anti-UBQLN1 | Sigma-Aldrich | U7383 |
| Anti-UBQLN1 | Proteintech Group Inc. | 23516-1-AP |
| Anti-UBQLN2 | Novus Biologicals | NBP2-25164 |
| Anti-UBQLN2 | Proteintech Group Inc. | 23449-1-AP |
| Anti-UBQLN4 | Santa Cruz Biotechnology | sc-136145 |

| Reagent/resource | Reference or source | Identifier or catalog number |
|---|---|---|
| Anti-FLAG | Wako Pure Chemical Industries, Ltd. | 018-22381 |
| Anti-HA | Proteintech Group Inc. | 51064-2-AP |
| Anti-α-syn | Abcam | ab138501 |
| 6 nm colloidal gold anti-rabbit IgG | Jackson ImmunoResearch Laboratories | 111-195-144 |
| Anti-G3BP | Abcam | ab56574 |
| Anti-eIF4G1 | Proteintech Group Inc. | 15704-1-AP |
| Anti-α-syn (phospho S129) | Abcam | ab51253 [EP1536Y] |
| Anti-α-Syn (phospho S129) | Wako Pure Chemical Industries, Ltd. | pSyn#64 |
| Anti-β-actin | Merck Millipore | MAB1501[clone C4] |
| Rabbit (DA1E) mAb IgG XP® Isotype Control | Cell Signaling Technology Inc. | #3900 |
| Mouse IgG1 Isotype Control | Agilent DAKO | x0931 |
| **Oligonucleotides and other sequence-based reagents** | | |
| Control-sgRNA | Shalem et al (2014) | N/A |
| UBQLN2-sgRNA | This study | N/A |
| UBQLN1-sgRNA | This study | N/A |
| UBQLN1-siRNA | Kind gift from the activities of the Molecular Profiling Committee, Grant-in-Aid for Scientific Research on Innovative Areas "Advanced Animal Model Support." | N/A |
| UBQLN2-siRNA | Kind gift from the activities of the Molecular Profiling Committee, Grant-in-Aid for Scientific Research on Innovative Areas "Advanced Animal Model Support." | N/A |
| UBQLN4-siRNA | Kind gift from the activities of the Molecular Profiling Committee, Grant-in-Aid for Scientific Research on Innovative Areas "Advanced Animal Model Support." | N/A |
| siRNA negative control | Thermo Fisher Scientific | 12935300 |
| **Chemicals, enzymes, and other reagents** | | |
| SO286 | Kind gift from Professor Emeritus Seiichiro Ogawa of Keio University | N/A |
| SO82 | Kind gift from Professor Emeritus Seiichiro Ogawa of Keio University | N/A |
| Biotin–SO286 | NARD Institute, Ltd. | N/A |
| Biotin-SO82 | NARD Institute, Ltd. | N/A |
| Sodium arsenite | Sigma-Aldrich | S7400 |
| 1,6-hexanediol | Tokyo Chemical Industry | 629-11-8 |
| DMEM | Sigma-Aldrich | D5796 |
| 100 U/mL penicillin/ streptomycin | Nacalai Tesque | 26253-84 |

| Reagent/resource | Reference or source | Identifier or catalog number |
|---|---|---|
| MEM nonessential amino acid solution | Nacalai Tesque | 06344-56 |
| Sodium pyruvate | Nacalai Tesque | 06977-34 |
| L-glutamine | Nacalai Tesque | 16948-04 |
| Lipofectamine 3000 | Thermo Fisher Scientific | L3000001 |
| G418 | Roche Diagnostics | G418-RO |
| PEI MAX | Polysciences | 49553-93-7 |
| Blasticidin S | Wako Pure Chemical Industries, Ltd. | 029-18701 |
| *Escherichia coli* | TaKaRa Bio Inc. | 9126 |
| Luria–Bertani broth | Nacalai Tesque | 20068-75 |
| Isopropyl β-D-1-thiogalactopyranoside | Wako Pure Chemical Industries, Ltd. | 094-05144 |
| PBS | TaKaRa Bio Inc. | T9181 |
| Protease inhibitor cocktail | Merck Millipore | 11697498001 |
| Triton X-100 | Wako Pure Chemical Industries, Ltd. | 581-81705 |
| Glutathione Sepharose 4 Fast Flow | Cytiva | 17513202 |
| PreScission Protease | Cytiva | 27-0843-01 |
| Amicon Ultra Centrifugal Filter | Merck Millipore | UFC901024 |
| DyLight488 NHS Ester | Thermo Fisher Scientific | 46402 |
| DyLight633 NHS Ester | Thermo Fisher Scientific | 46414 |
| polyethylene glycol(PEG)-8000 | Sigma-Aldrich | P2139 |
| 96-well glass-bottom plate | Mattek | P96G-1.5-5-F |
| Bovine serum albumin | Sigma-Aldrich | A8022 |
| Full-length tau (2N4R, Tau441) | MarqStress | SPR-479B |
| anti-FLAG beads | Wako Pure Chemical Industries, Ltd. | 018-22783 |
| FLAG peptide | Wako Pure Chemical Industries, Ltd. | 043-34583 |
| West Dura Extended Duration Substrate | Thermo Fisher Scientific | 34076 |
| Paraformaldehyde | Wako Pure Chemical Industries, Ltd. | 163-20145 |
| Glutaraldehyde | Nacalai Tesque | 17003-92 |
| Tween 20 | Wako Pure Chemical Industries, Ltd. | 167-11515 |
| Glutaraldehyde | TAAB | G004 |
| Epok812 | Oken Shoji | #02-1003 |
| Gridded glass coverslip | Matsunami Glass | GC1310 |
| Goat serum | Sigma-Aldrich | G9023 |
| Prolong Gold | Thermo Fisher Scientific | P36930 |
| ThS | Sigma-Aldrich | T1892 |
| Lipofectamine RNAiMax | Thermo Fisher Scientific | 13778075 |

| Reagent/resource | Reference or source | Identifier or catalog number |
|---|---|---|
| 26-G needle | TERUMO Corporation | SS-01T2613S |
| Blocking One Histo | Nacalai Tesque | 0634964 |
| TrueBlack Lipofuscin Autofluorescence Quencher | Biotium, Inc. | 23007 |
| VECTASHIELD mounting medium | Vector laboratories | H1800 |
| **Software** | | |
| CHOPCHOP | Labun et al (2019) | |
| ImageJ Fiji (version: 1.54 f) | National Institutes of Health | |
| BIAcore T200 evaluation software (ver. 2.0) | Cytiva | |
| Kodec4.4.7.39 software | Ngo et al (2015) | |
| GraphPad Prism (ver. 9.4.1) | GraphPad Software | |
| **Other** | | |
| BZ-X810 | Keyence Corporation | |
| LSM880 | Carl Zeiss | |
| JEM-1400 Flash | JEOL Ltd. | |
| BIAcore T200 | Cytiva | |
| MS-NEX | Research Institute of Biomolecule Metrology Co., Ltd. | |
| Electron beam-deposited tips | NanoWorld | |

## Materials

The SO286 and SO82 were a kind gift from Professor Emeritus Seiichiro Ogawa of Keio University, Tokyo, Japan. Biotin–SO286 and Biotin-SO82 were purchased from NARD Institute, Ltd, Tokyo, Japan. Sodium arsenite ($AsNaO_2$) was purchased from Sigma-Aldrich (St. Louis, MO, USA). The 1,6-hexanediol was purchased from Tokyo Chemical Industry, Tokyo, Japan.

## Cell culture

SH-SY5Y cells (ATCC, Manassas, VA, USA #CRL-2266) and HEK293T cells (ATCC, #CRL-3216) were cultured in Dulbecco's Modified Eagle Medium (DMEM) supplemented with 10% fetal bovine serum (Sigma-Aldrich, St. Louis, MO, USA), 100 U/mL penicillin/streptomycin (Nacalai Tesque, Kyoto, Japan), MEM nonessential amino acid solution (Nacalai Tesque), 1 mM sodium pyruvate (Nacalai Tesque), and 2 mM L-glutamine (Nacalai Tesque) at 37 °C with 5% $CO_2$. SH-SY5Y cells stably expressing α-syn(WT)-EGFP and α-syn(A53T)-EGFP were established via the transfection of the respective vectors into SH-SY5Y cells using Lipofectamine 3000 (Thermo Fisher Scientific, Waltham, MA, USA), followed by selection with G418 (Roche Diagnostics, Basel, Switzerland). The KO cell lines were generated using a lentiviral expression system. Single guide RNA (sgRNA) sequences targeting the *UBQLN2* and *UBQLN1* genes were designed using the

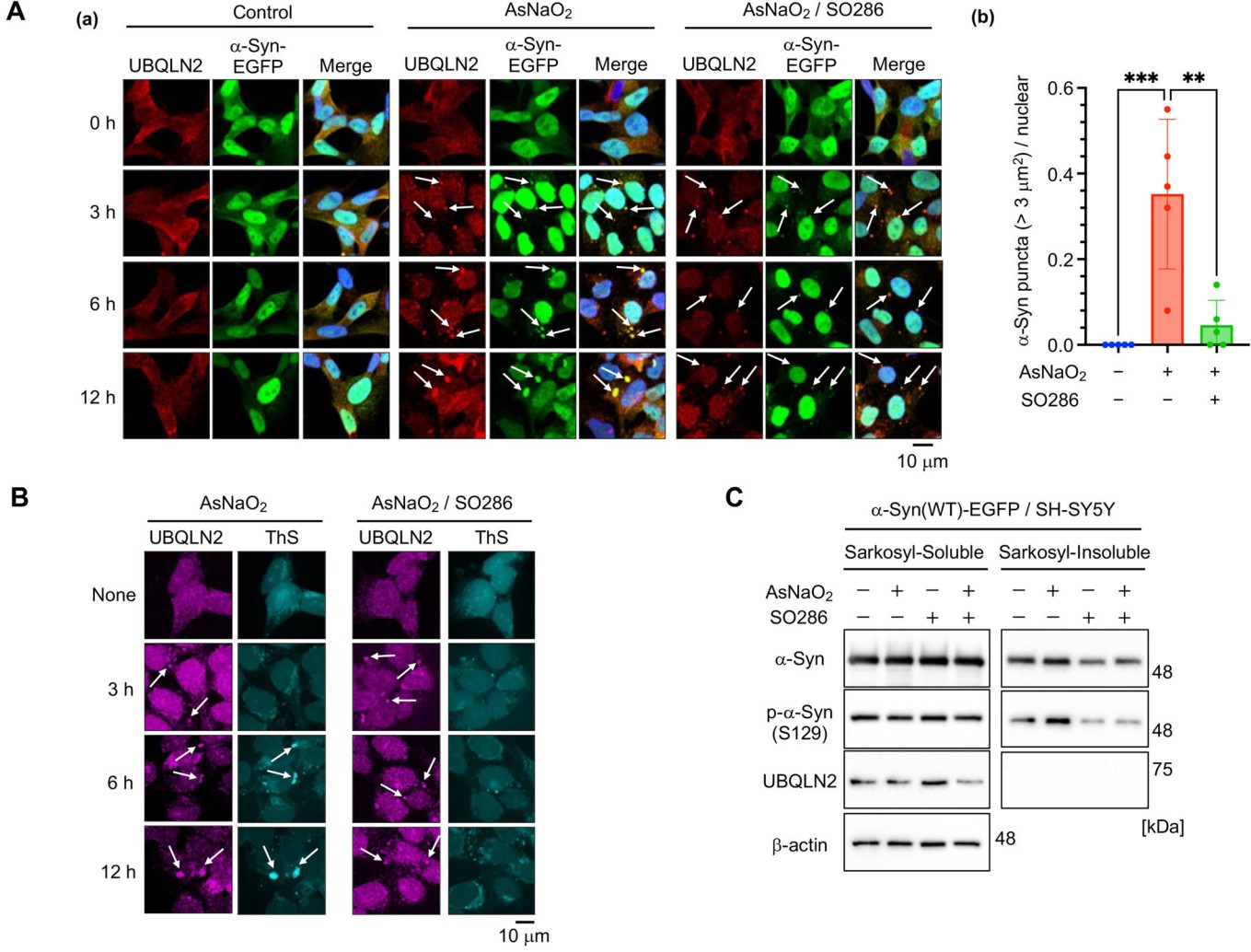

**Figure 6.  SO286 inhibits α-syn aggregation within UBQLN droplets in cultured cells.**

(A) (a) α-Syn(WT)-EGFP/SH-SY5Y cells were treated with 50 μM AsNaO₂ for the indicated time points in the presence of 20 μM SO286 and immunostained with anti-UBQLN2 and Hoechst. UBQLN2 droplets and α-syn-EGFP condensates are indicated by white arrows. (b) Quantification of the number of α-syn-EGFP puncta (> 3 μm²) per cell. **P < 0.01, ***P < 0.001 (Dunnett's test, versus AsNaO₂). Five images were quantified per condition. Values are the means ± SDs. AsNaO₂ vs Control: P = 0.0004, AsNaO₂ vs AsNaO₂/SO286: P = 0.0013. (B) α-Syn(WT)-EGFP/SH-SY5Y cells were treated with 50 μM AsNaO₂ for the indicated time points in the presence of 20 μM SO286 before being immunostained with anti-UBQLN2 and ThS. UBQLN2 droplets are indicated by white arrows. (C) α-Syn(WT)-EGFP/SH-SY5Y cells were treated with 50 μM AsNaO₂ for 12 h in the presence of 20 μM SO286. Cell lysates of the sarkosyl-soluble and -insoluble fractions were subjected to SDS-PAGE and immunoblotted with the indicated antibodies. Representative blot shown, n = 2. Source data are available online for this figure.

CHOPCHOP tool (Labun et al, 2019). These nontargeting sgRNA sequences for generating Control-KO cells (5′-CGCAGTCATTCGA-TAGGAAT-3′) (Shalem et al, 2014), UBQLN2 (5′-CGCGGTGCCCGA-GAACAGCT-3′), and UBQLN1 (5′-GACATTAGCAACGGAAGCCC-3′) were inserted into the BsmbI sites of a lentiCRISPRv2-blast plasmid (Addgene, Watertown, MA, USA plasmid 83480) using standard techniques. HEK293T cells were transiently co-transfected with lentiviral vectors using PEI MAX reagent (Polysciences, Warrington, PA, USA). Four hours after transfection, the medium was replaced with fresh culture medium. After culturing for 72 h, the growth medium containing the lentivirus was collected. SH-SY5Y cells stably expressing α-syn (WT)-EGFP and α-syn(A53T)-EGFP were incubated with the collected virus-containing medium for 48 h. Uninfected cells were removed using 5 μg/mL blasticidin S (Wako Pure Chemical Industries, Ltd., Osaka, Japan).

UBQLN1, UBQLN2 and UBQLN4 triple-KO T-Rex293 cells (kindly provided by Professor Eisuke Itakura, Chiba University, Chiba, Japan) (Itakura et al, 2016) were cultured in Dulbecco's MEM supplemented with 10% fetal bovine serum, 100 U/mL penicillin/streptomycin, MEM nonessential amino acid mixture, 1 mM sodium pyruvate, and 2 mM L-glutamine at 37 °C with 5% CO₂. All SH-SY5Y cells used in this study were authenticated by short tandem repeat profiling performed by ATCC. All the cell lines were tested negative for mycoplasma contamination (MycoAlert Detection Kit, Lonza, Basel, Switzerland).

## Expression and purification of recombinant proteins

UBQLN1, UBQLN2, UBQLN4, HA-UBQLN2, and α-syn were amplified, subcloned, and inserted into the pGEX-6p-1 vector

(Cytiva). Glutathione S-transferase (GST)-tagged UBQLN constructs were expressed in the *Escherichia coli* (*E. coli*) strain (TaKaRa Bio Inc., Kusatsu, Shiga, Japan) in Luria–Bertani broth (Nacalai Tesque) at 37 °C overnight. Bacteria were induced with isopropyl β-D-1-thiogalactopyranoside (Wako Pure Chemical Industries, Ltd.) and harvested after 4 h at 37 °C. The bacterial pellets were frozen, lysed in phosphate-buffered saline (PBS; TaKaRa Bio Inc.) containing 0.1% Triton X-100 (Wako Pure Chemical Industries, Ltd.) and protease inhibitor cocktail (Merck Millipore, Burlington, MA, USA), and cleared by centrifugation at 13,000×*g* for 15 min at 4 °C. GST-tagged proteins were purified using a Glutathione Sepharose 4 Fast Flow (Cytiva, Marlborough, MA, USA). To cleave the GST tags, the beads were incubated with PreScission Protease (Cytiva) for 16 h at 4 °C. Flow-through fractions containing unlabeled recombinant proteins were collected and concentrated, and the buffer was exchanged with LLPS buffer (200 mM NaCl, 20 mM sodium phosphate, and 0.5 mM ethylenediaminetetraacetic acid; pH 6.8) using an Amicon Ultra Centrifugal Filter (Merck Millipore). GST-tagged α-syn was expressed in *E. coli.* and purified using the same method as for UBQLN constructs, which involves affinity chromatography with Glutathione Sepharose 4 Fast Flow followed by cleavage of the GST tag using PreScission Protease.

The 260/280 absorbance ratios of the prepared UBQLN1, UBQLN2, UBQLN4, and α-syn proteins are 0.90, 0.85, 0.93, and 0.85, respectively.

## Fluorescent labeling

UBQLN1, UBQLN2, and UBQLN4 were fluorescently labeled with DyLight488 NHS Ester (Thermo Fisher Scientific), and α-syn was fluorescently labeled with DyLight633 NHS Ester (Thermo Fisher Scientific) according to the manufacturer's instructions.

## Formation and imaging of condensates

The UBQLN1, UBQLN2, and UBQLN4 constructs were prepared to contain 10 µM protein (spiked with DyLight488-labeled UBQLN1, UBQLN2, or UBQLN4 at a 1:100 molar ratio) in LLPS buffer containing 3% PEG (for UBQLN2 and 4) or 5% PEG (for UBQLN1). These samples were mixed with 10 µM α-syn (spiked with DyLight633-labeled α-syn, 1:100 molar ratio), and SO286 or SO82 was added to a 96-well glass-bottom plate (Mattek) and incubated with 3% bovine serum albumin (BSA; Sigma-Aldrich) to reduce the rapid coating of the protein droplets onto the glass surface. Condensates were imaged at 1, 24, 48, 72, and 96 h after phase separation using a BZ-X810 fluorescence microscope (Keyence Corporation, Osaka, Japan) at room temperature.

Full-length tau (2N4R, Tau441) was obtained from MarqStress. DyLight633 labeling was performed according to the manufacturer's protocol. Tau droplet formation was conducted following the method described in reference (Kanaan et al, 2020). Tau at 10 µM (spiked with DyLight633-labeled tau, 1:20 molar ratio) in tau-LLPS buffer (10 µM HEPES-NaOH pH 7.4, 150 mM NaCl, 0.1 mM EDTA, 2 mM DTT), and LLPS was induced by adding 10% PEG. To examine α-syn uptake, 10 µM α-syn (spiked with DyLight488-labeled α-syn, 1:100 molar ratio) were mixed with tau, and LLPS was induced by adding 10% PEG. Tau droplets were then imaged after incubation at 37 °C for 5 h to confirm uptake.

## Interaction assays

To detect the interaction between UBQLN2 and α-syn, FLAG-tagged α-syn was purified from HEK293T cells, while HA-tagged UBQLN constructs were expressed and purified from *E. coli*. HEK293T cells were transfected 24 h after seeding using Lipofectamine 3000 with α-syn-FLAG. Forty-eight hours post-transfection, the cells were lysed in BA100 buffer (20 mM Tris-HCl [pH 7.5], 100 mM NaCl, 0.1% Triton X-100, and proteinase inhibitor). The cell lysates were then incubated with anti-FLAG beads (Wako Pure Chemical Industries, Ltd.) for 3 h at 4 °C with constant rotation. The α-syn-FLAG/anti-FLAG beads were washed twice with BA100 buffer and three times with LLPS buffer before being incubated with either UBQLN1, UBQLN2, or UBQLN4 recombinant proteins for 2 h at 4 °C with constant rotation. The beads were then washed three times with LLPS buffer and eluted with FLAG peptide (Wako Pure Chemical Industries, Ltd.). Next, the samples were loaded onto a 10%–20% Bis-tris gel, and the proteins were separated via sodium dodecyl sulfate polyacrylamide gel electrophoresis (SDS-PAGE). Proteins were then transferred onto a polyvinylidene fluoride (PVDF) membrane and probed with specific antibodies, followed by detection using West Dura Extended Duration Substrate (Thermo Fisher Scientific).

To detect the self-interaction of UBQLN2, FLAG-tagged UBQLN2 was purified from UBQLN1/2/4 triple-knockout T-REx 293 cells to eliminate interference from endogenous UBQLN proteins and to enable specific analysis of interactions with HA-tagged UBQLN2 purified from *E. coli*. The cells were transfected 24 h after seeding using Lipofectamine 3000 with FLAG-UBQLN2. Forty-eight hours post-transfection, the cells were lysed in BA100 buffer. The cell lysates were then incubated with anti-FLAG beads for 3 h at 4 °C with constant rotation. Next, the FLAG-UBQLN2/anti-FLAG beads were washed twice with BA100 buffer and three times with LLPS buffer before being incubated with HA-UBQLN2 recombinant protein for 2 h at 4 °C with constant rotation. The beads were then washed three times with LLPS buffer and eluted with FLAG peptide. The samples then underwent SDS-PAGE and probing with antibodies using the same protocol as for the detection of interactions between UBQLN2 and α-syn-FLAG.

The primary antibodies were as follows: anti-UBQLN1 (Cell Signaling Technology Inc, Danvers, MA, USA, #14526), anti-UBQLN2 (Novus Biologicals, Centennial, CO, USA, NBP2-25164), anti-UBQLN4 (Santa Cruz Biotechnology, Dallas, TX, USA, sc-136145), anti-HA (Proteintech Group Inc., Rosemont, IL, USA, 51064-2-AP), and anti-FLAG (Wako Pure Chemical Industries, Ltd., 018-22381).

## Far-western blotting

Far-western blotting was conducted using a method modified from Wu et al (Wu et al, 2007). Each UBQLN2-HA mutant was resolved using SDS-PAGE and transferred onto a PVDF membrane. The membrane was then blocked with blocking buffer (1% BSA, 20 mM Tris-HCl [pH 7.5], and 1 mM dithiothreitol) containing 0.05% Tween 20 for 1 h at room temperature, incubated with 20 µM His-α-syn recombinant protein in blocking buffer overnight at 4 °C, and washed with washing buffer (10 mM Tris-HCl and 100 mM NaCl). Next, the membrane was fixed with 4% paraformaldehyde (Wako Pure Chemical Industries, Ltd.) and 0.02% glutaraldehyde (Nacalai Tesque) at room temperature for 30 min and washed with washing

buffer. The His-α-syn protein that remained bound to the UBQLN2-HA mutants on the membrane was then probed using an anti-α-syn monoclonal antibody (Abcam, Cambridge, UK, ab138501) for 3 h at room temperature. Subsequently, the membrane was washed twice using washing buffer containing 0.02% Tween 20 (Wako Pure Chemical Industries, Ltd.) to remove unbound antibodies. Bound antibodies were then detected using West Dura Extended Duration Substrate.

## FRAP

FRAP experiments were performed using an LSM880 confocal laser scanning microscope (Carl Zeiss, Oberkochen, Germany). Images were acquired via a 488 nm or 633 nm laser at a 100 ms exposure. Images were taken every 1 s over 70 s. Images from each dataset were analyzed in ImageJ Fiji (NIH, version: 1.54 f) (Schneider et al, 2012) to measure the fluorescent intensity over time, thus generating FRAP curves for the images. The data points were fitted to a single-exponential equation to determine the recovery time.

## Transmission electron microscopy

For conventional electron microscopy, samples were fixed with 2.5% glutaraldehyde (TAAB, Aldermaston, UK) in 0.1 M phosphate buffer (pH 7.4). Fixed samples were then dehydrated through a graded ethanol series and embedded in Epok812 (Oken Shoji, Tokyo, Japan). Ultrathin sections were cut before being stained with uranyl acetate and lead citrate, and were then examined using a transmission electron microscope (JEM-1400 Flash, JEOL Ltd., Tokyo, Japan). For immunoelectron microscopy, the samples were blocked with 1% BSA in PBS for 10 min, incubated with anti-α-syn (Abcam, ab138501) for 4 h, and then incubated with a secondary antibody (6 nm colloidal gold anti-rabbit IgG, Jackson ImmunoResearch Laboratories, West Grove, PA, USA) for 4 h at room temperature. After being washed with PBS, the samples were fixed and embedded as for conventional electron microscopy.

For immunoelectron microscopy of cells, α-syn(WT)-EGFP/SH-SY5Y cells were grown on a gridded glass coverslip (Matsunami Glass, Osaka, Japan). Following treatment with AsNaO$_2$ for 12 h, the cells were fixed and permeabilized with fixation and permeabilization (FP) solution (2% paraformaldehyde, 0.25% Triton X-100. α-syn-EGFP puncta were imaged using an LSM880 confocal laser scanning microscope. The cells were immunolabeled with anti-α-syn antibody and colloidal golds, after which they were embedded as described above. Ultrathin sections were cut before being stained with uranyl acetate and lead citrate, and were then examined using a transmission electron microscope (JEM-1400 Flash, JEOL Ltd.).

## Immunofluorescence

Cells grown on glass coverslips were fixed and permeabilized with FP solution at room temperature for 10 min before being further fixed with 2% paraformaldehyde at room temperature for 20 min. Coverslips were washed with PBS prior to blocking with 5% goat serum (Sigma-Aldrich) in 0.25% Triton X-100 for 1 h at room temperature. The cells were incubated with primary antibody for 3 h at room temperature, and then with secondary antibody for 2 h at room temperature in the dark. Primary and secondary antibodies were diluted in 3% BSA and 0.25% Triton X-100 solution. After the primary and secondary antibody incubations, three five-minute washes with PBS containing 0.25% Triton X-100 were performed at room temperature on an orbital shaker. Coverslips were mounted with Prolong Gold (Thermo Fisher Scientific) mounting media, and the immunostaining was observed and imaged using an LSM880 confocal laser scanning microscope. The primary antibodies were as follows: anti-UBQLN1 (Sigma-Aldrich, U7383), anti-UBQLN2 (Novus Biologicals, NBP2-25164), anti-UBQLN4 (Santa Cruz Biotechnology, sc-136145), anti-FLAG (Wako Pure Chemical Industries, Ltd., 018-22381), anti-G3BP (Abcam, ab56574), and anti-eIF4G1 (Proteintech Group Inc., 15704-1-AP).

## ThS staining

Cells grown on glass coverslips were fixed and permeabilized with a solution containing FP solution at room temperature for 10 min, and then further fixed with 2% paraformaldehyde at room temperature for 20 min. Coverslips were washed with PBS and incubated with 0.1% ThS (Sigma-Aldrich) in PBS, and were then immunostained with anti-UBQLN2 (Novus Biologicals, NBP2-25164) using the same protocol as for immunofluorescence. Cells on the coverslips were observed and imaged using an LSM880 confocal laser scanning microscope.

## siRNA transfection

The siRNAs targeting UBQLN1, UBQLN2, and UBQLN4 were kindly provided by the activities of the Molecular Profiling Committee, Grant-in-Aid for Scientific Research on Innovative Areas "Advanced Animal Model Support." The siRNA negative control was purchased from Thermo Fisher Scientific. The SH-SY5Y cells were transfected with siRNAs using Lipofectamine RNAiMax (Thermo Fisher Scientific) in accordance with the manufacturer's instructions.

## Subcellular fractionation using sarkosyl

Cells were resuspended in A68 buffer (10 mM Tris-HCl [pH 7.4], 800 mM NaCl, 1 mM ethylenediaminetetraacetic acid, 0.5 mM egtazic acid, 10% sucrose, and proteinase inhibitor) and lysed by passing the samples ten times through a 26-G needle (TERUMO Corporation, Tokyo, Japan). The cell lysates were then incubated with 2% sarkosyl in A68 buffer on ice for 30 min with constant rotation before being centrifuged at 100,000×g for 20 min at 4 °C. The resulting pellet and supernatant were designated the sarkosyl-soluble and -insoluble fractions, respectively. After a bicinchoninic acid assay was performed to adjust the protein concentrations, SDS-PAGE and immunoblotting analyses were performed for each fraction. The primary antibodies were as follows: anti-UBQLN2 (Novus Biologicals, NBP2-25164), anti-α-syn (phospho S129) (Abcam, ab51253 [EP1536Y]), anti-α-syn (Abcam, ab138501), and anti-β-actin (Merck Millipore, MAB1501[clone C4]) antibodies.

## Western blot quantification

The band intensities of Western blot images were quantified using ImageJ Fiji. For each blot, the intensity of target protein bands was measured by defining regions of interest (ROIs) around the bands, and background signals were subtracted. The resulting values were

normalized to the corresponding loading control. The number of biological replicates analyzed and the corresponding statistical analyses are detailed in the Figure legends.

## Binding analysis via surface plasmon resonance assay

A surface plasmon resonance assay was performed using a BIAcore T200 instrument (Cytiva). The sensor chip streptavidin was obtained from Cytiva. First, 1 µM biotin–SO286 was fixed onto the sensor chip according to the manufacturer's instructions. Next, UBQLN2 or HA-UBQLN2 constructs were injected at 30 µL/min over a 2 min period in PBS containing 0.05% Tween 20 and 1 mg/mL BSA as the running buffer. The regeneration of the sensor chip was then performed using 10 mM glycine (pH 1.5) and 0.5% Tween 20 buffer. A reference flow cell (i.e., flow cell with biotin) was used for each binding analysis of biotin–SO286. Specific binding was quantified by subtracting the ligand response from the reference response. Kinetic constants were calculated using a 1:1 binding model with BIAcore T200 evaluation software (ver. 2.0; Cytiva).

## HS-AFM

The HS-AFM observations were performed in tapping mode using a sample scanning HS-AFM instrument (MS-NEX, Research Institute of Biomolecule Metrology Co., Ltd., Tokyo, Japan) and cantilevers (length: ~7 µm, width: approximately 2 µm, thickness: approximately 0.08 µm, resonant frequency: 1.2 MHz in air, and spring constant: 0.15 N/m) with electron beam-deposited tips (USC-F1.2-k0.15, Nano-World, Neuchâtel, Switzerland) (Uchihashi et al, 2012).

For the imaging of UBQLN2 and α-syn, one drop (~2 µL) of UBQLN2 (1 µM) or α-syn solution (1 µM) was deposited onto freshly cleaved mica that was glued to the top of a glass stage (diameter, 1.5 mm; height, 2 mm). For the imaging of the α-syn and UBQLN2 mixture, α-syn solution (1 µM) was mixed with UBQLN2 solution (1 µM), and one drop (~2 µL) of the mixture was deposited onto freshly cleaved mica. For the imaging of the α-syn and UBQLN1, UBQLN2, or UBQLN4 mixture in the presence or absence of SO286, SO286 (0.25 mM, dissolved in dimethyl sulfoxide) or dimethyl sulfoxide was premixed with UBQLN1, UBQLN2, or UBQLN4 solution (1 µM) and incubated for 5 min. The α-syn solution (1 µM) was then added, and one drop (~2 µL) of the mixture was deposited onto freshly cleaved mica. For each observation, the mica was rinsed after incubation for 3 min and immersed in a liquid cell containing approximately 90 µL of imaging buffer (150 mM NaCl and 20 mM 4-[2-hydroxyethyl]-1-piperazineethanesulfonic acid-NaOH [pH 7.5]). The imaging conditions were as follows: scan size (pixel size, imaging rate), $600 \times 600 \, nm^2$ ($150 \times 150$ pixels, 2 frames/s) or $200 \times 200 \, nm^2$ ($120 \times 120$ pixels, 4 frames/s). Imaging was performed at 23 °C. The HS-AFM data were viewed and analyzed using Kodec4.4.7.39 software (Ngo et al, 2015).

The calculation of the percentage area covered by proteins was conducted using the "Auto Threshold" and "Analyze Particles" functions of ImageJ software. The values were calculated from images ($600 \times 600 \, nm^2$) obtained from three independent experiments.

## Clinical samples

Detailed clinical information was collected from the neurodegenerative disease database maintained by the Department of Neurology at Juntendo University, Tokyo, Japan. Details on these patients are given in Appendix Table S1. All procedures involving brain autopsy were conducted in accordance with the ethical standards of the Ethics Committee of Juntendo University School of Medicine, Tokyo, Japan (approval number: 2019012), as well as the principles outlined in the 1964 Declaration of Helsinki and its subsequent amendments or equivalent ethical guidelines. Neuropathological evaluations for PD were performed by neuropathologists affiliated with the Department of Neurology at Juntendo University, Tokyo, Japan. Written informed consent for autopsy and subsequent analysis of tissue samples was obtained from all patients or their legal representatives.

## Histochemical analysis of human brain

Double immunofluorescence staining was performed using 6-µm-thick, 15% neutral buffered formalin-fixed, paraffin-embedded sections from the midbrain of pathologically diagnosed PD cases collected by the Department of Neurology, Juntendo University Hospital, Tokyo, Japan. Deparaffinized sections microwaved in Tris EDTA buffer, pH 9.0 (Agilent DAKO, Santa Clara, CA, USA, S2367) for 10 min for antigen retrieval, were neutralized with PBS. Sections were treated with blocking buffer (Nacalai Tesque, cat. no. 0634964, Blocking One Histo) and incubated with anti-UBQLN1 (1:200; Proteintech Group Inc., 23516-1-AP), anti-UBQLN2 (1:200; Proteintech Group Inc., 23449-1-AP), or anti-UBQLN4 (1:200; Santa Cruz Biotechnology, sc-136145; A333) antibodies along with anti-α-Syn (phospho S129) (1:250; Wako Pure Chemical Industries, Ltd., pSyn#64) or anti-α-syn (phospho S129) (1:1000, ab51253; Abcam, EP1536Y) overnight at 4 °C. As negative controls, the corresponding isotype control IgGs were used at the same concentrations as the primary antibodies, together with the anti-p-α-syn antibody, as follows: Rabbit (DA1E) mAb IgG XP® Isotype Control (Cell Signaling Technology Inc.); and Mouse IgG1 Isotype Control (Agilent DAKO, Santa Clara, CA, USA). Primary antibodies were visualized with secondary antibodies (1:1000; Thermo Fisher Scientific, Alexa Fluor 488plus and 594plus) for 1 h at room temperature. To reduce lipofuscin autofluorescence, sections were further treated with TrueBlack Lipofuscin Auto-fluorescence Quencher (Biotium, Inc., Fremont, CA, USA), diluted with 70% ethanol for 30 s, and briefly washed with PBS before mounting with VECTASHIELD mounting medium (Vector laboratories, Newark, CA, USA) and assessed using LSM880 with the Airyscan laser scanning microscope system (Carl Zeiss).

## Statistical analysis

Statistical analysis was performed using GraphPad Prism (ver. 9.4.1) software (GraphPad Software, San Diego, CA, USA). Details of each statistical method are shown in the respective figure legends. Sample sizes were not determined by statistical power calculations. However, the chosen sample sizes are consistent with those used in previous studies and were considered sufficient to detect biologically meaningful effects.

# Data availability

Source data of microscopy images in Figs. 3, 6 and EV3 have been deposited in the BioImage Archive under accession number S-BIAD2253.

The source data of this paper are collected in the following database record: biostudies:S-SCDT-10_1038-S44318-025-00591-1.

## Peer review information

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

## Acknowledgements

We thank K. Okuma for technical assistance with electron microscopy, A Sumii for technical assistance with histochemical analysis of human brains, S Soma for technical assistance with cell-based assays, and A Matsumoto and K Hoshino for assistance with target identification of SO286. We also thank Prof. Eisuke Itakura at Chiba University for kindly providing T-REx293 cells. We thank the Research Institute for Diseases of Old Age, the Laboratory of Proteomics and Biomolecular Science, the Laboratory of Molecular and Biochemical Research, and the Laboratory of Cell Biology, Biomedical Research Core Facilities and Juntendo University Graduate School of Medicine for technical assistance. We thank Bronwen Gardner, PhD, from Edanz (https://jp.edanz.com/ac) for editing a draft of this manuscript. This work was supported by JSPS KAKENHI (Grant No. 18H02099 and 21H02072 to MI, Grant No. 24K10651 and 23K20044 to YS, Grant No. 21H04820 and 24H00068 to NH) from The Ministry of Education, Culture, Sports, Science and Technology, Japan, and by JSPS KAKENHI Grant Number JP 22H04922 (AdAMS) to MI, and AMED (Grant No. JP 16nk0101346 to MI, and Grant No. JP 25wm0625503 to NH).

## Author contributions

**Tomoki Takei**: Investigation; Writing—original draft. **Yukiko Sasazawa**: Funding acquisition; Investigation; Writing—original draft; Writing—review and editing. **Daisuke Noshiro**: Formal analysis; Investigation. **Mitsuhiro Kitagawa**: Investigation. **Tetsushi Kataura**: Investigation. **Hiroko Hirawake-Mogi**: Investigation. **Emi Kawauchi**: Investigation. **Yuya Nakano**: Investigation. **Etsu Tashiro**: Investigation. **Tsuyoshi Saitoh**: Investigation. **Shigeru Nishiyama**: Project administration. **Seiichiro Ogawa**: Project administration. **Soichiro Kakuta**: Investigation; Project administration. **Saiko Kazuno**: Investigation. **Yoshiki Miura**: Project administration. **Daisuke Taniguchi**: Project administration. **Viktor I Korolchuk**: Validation; Writing—review and editing. **Nobuo N Noda**: Conceptualization; Writing—original draft; Project administration; Writing—review and editing. **Shinji Saiki**: Project administration; Writing—review and editing. **Masaya Imoto**: Conceptualization; Supervision; Funding acquisition; Writing—original draft; Project administration; Writing—review and editing. **Nobutaka Hattori**: Supervision; Funding acquisition.

Source data underlying figure panels in this paper may have individual authorship assigned. Where available, figure panel/source data authorship is listed in the following database record: biostudies:S-SCDT-10_1038-S44318-025-00591-1.

## Disclosure and competing interests statement

The authors declare no competing interests.

# Expanded View Figures

**Figure EV1.  α-Syn aggregates within UBQLN2 droplets in vitro.**

(**A**) Fluorescence microscopy images of mixed solutions of UBQLN2 (1% DyLight488-labeling) with α-syn (1% DyLight633-labeling) for 24 h at the indicated concentrations. (**B**) Fluorescence microscopy images of mixed solutions of 10 μM UBQLN2, ΔSTI1-2/UBQLN2 or ΔUBL/UBQLN2 (1% DyLight488-labeling) and 10 μM α-syn (1% DyLight633-labeling) after incubation at 37 °C for 24 h. (**C**) Fluorescence microscopy images showing the disassembly of UBQLN droplets incorporating α-syn following the addition of 10% 1,6-HD. (**D**) (a) Fluorescence microscopy images of each 10 μM UBQLN (1% DyLight488-labeling) in the presence or absence of 10 μM α-syn (1% DyLight633-labeling) for 24 h and 96 h. (b) FRAP analysis of droplets of each UBQLN in the presence or absence of 10 μM α-syn for 24 h and 96 h. (c) Quantification of fractional recovery at 70 s after photobleaching from 10 separate droplets. *$P < 0.05$, N.S., nonsignificant (Welch's *t* test). Values are the means ± SDs. UBQLN2 (24 h): $P = 0.0136$, UBQLN2 (96 h): $P = 0.5731$, UBQLN1 (24 h): $P = 0.3975$, UBQLN1 (96 h): $P = 0.0894$, UBQLN4 (24 h): $P = 0.3556$, UBQLN4 (96 h): $P = 0.8157$. Source data are available online for this figure.

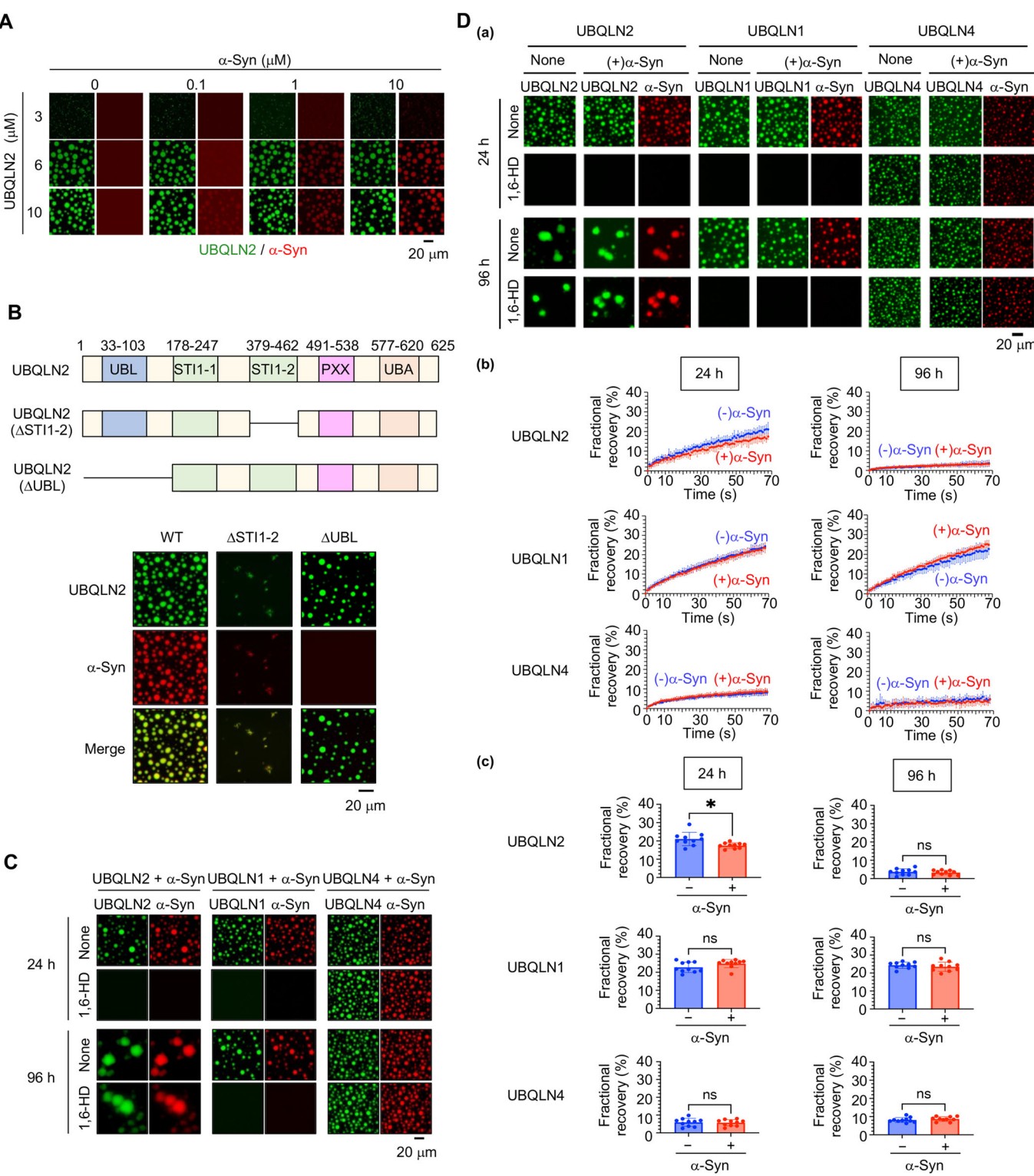

**A**

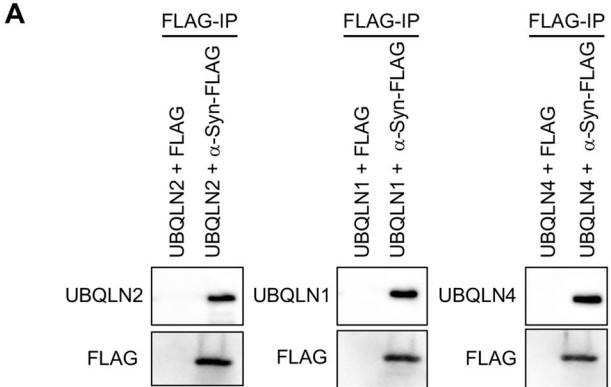

**B** (a)

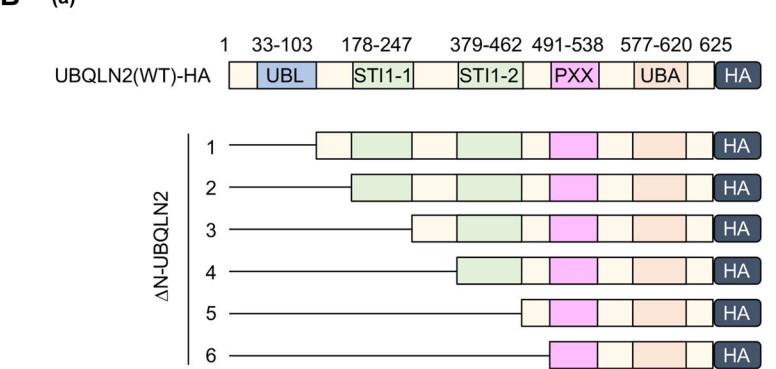

(b)

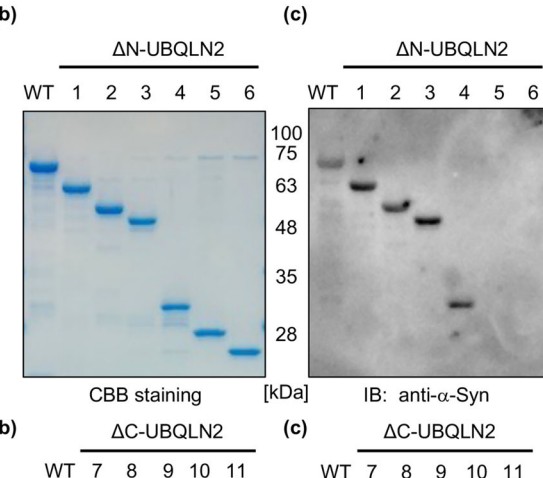

**C** (a)

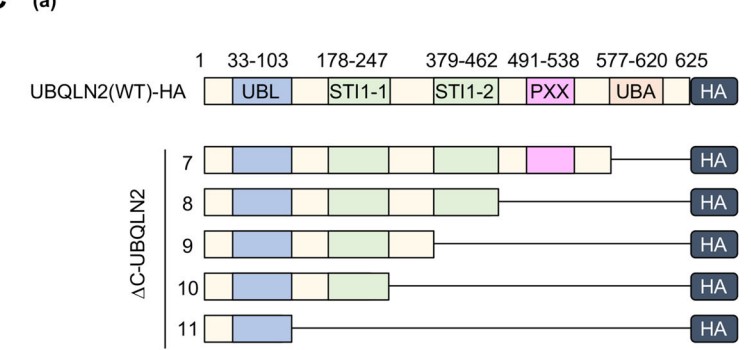

(b) (c)

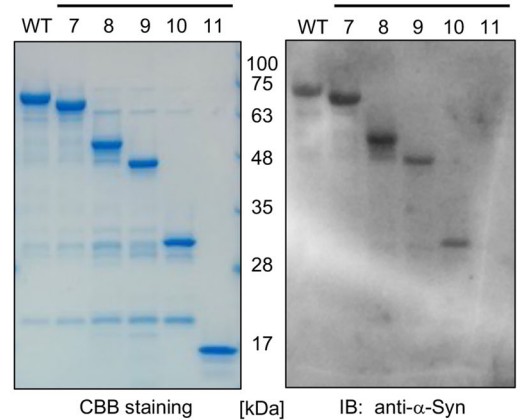

**D**

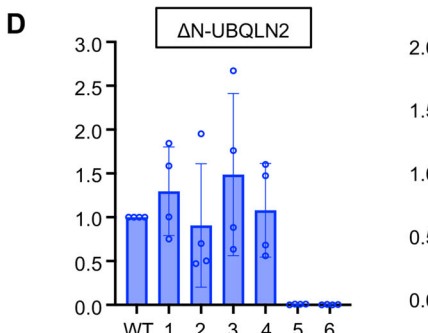

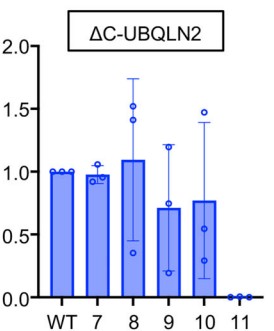

◀ **Figure EV2.  α-Syn directly interacts with UBQLN2.**

(**A**) Recombinant α-syn-FLAG and each hemagglutinin (HA)-UBQLN were incubated before α-syn-FLAG was pulled down with anti-FLAG beads and α-syn-FLAG was eluted with the FLAG peptide. The eluates were then immunoblotted with the indicated antibodies. Representative blot shown, $n = 2$. (**B, C**) (a) Illustrations of the UBQLN2 mutant with successive N-terminal deletions (**B**) or successive C-terminal deletions (**A**). (b) Representative SDS-PAGE images of each HA-tagged UBQLN2 mutant stained with Coomassie Brilliant Blue (CBB). (c) Representative far-western blotting results showing α-syn binding to each UBQLN2 mutant. After SDS-PAGE and membrane transfer, the membranes were incubated with recombinant His-tagged α-syn, followed by detection with an anti-α-syn antibody. (**D**) Quantification of α-syn binding to each UBQLN2 mutant. The binding signal (western blot) was normalized to the amount of HA-UBQLN2 protein (CBB staining). Graphs show the mean ± SD from four independent experiments for ΔN-UBQLN2 mutants and three independent experiments for ΔC-UBQLN2 mutants, corresponding to the representative results shown in (**B, C**). Source data are available online for this figure.

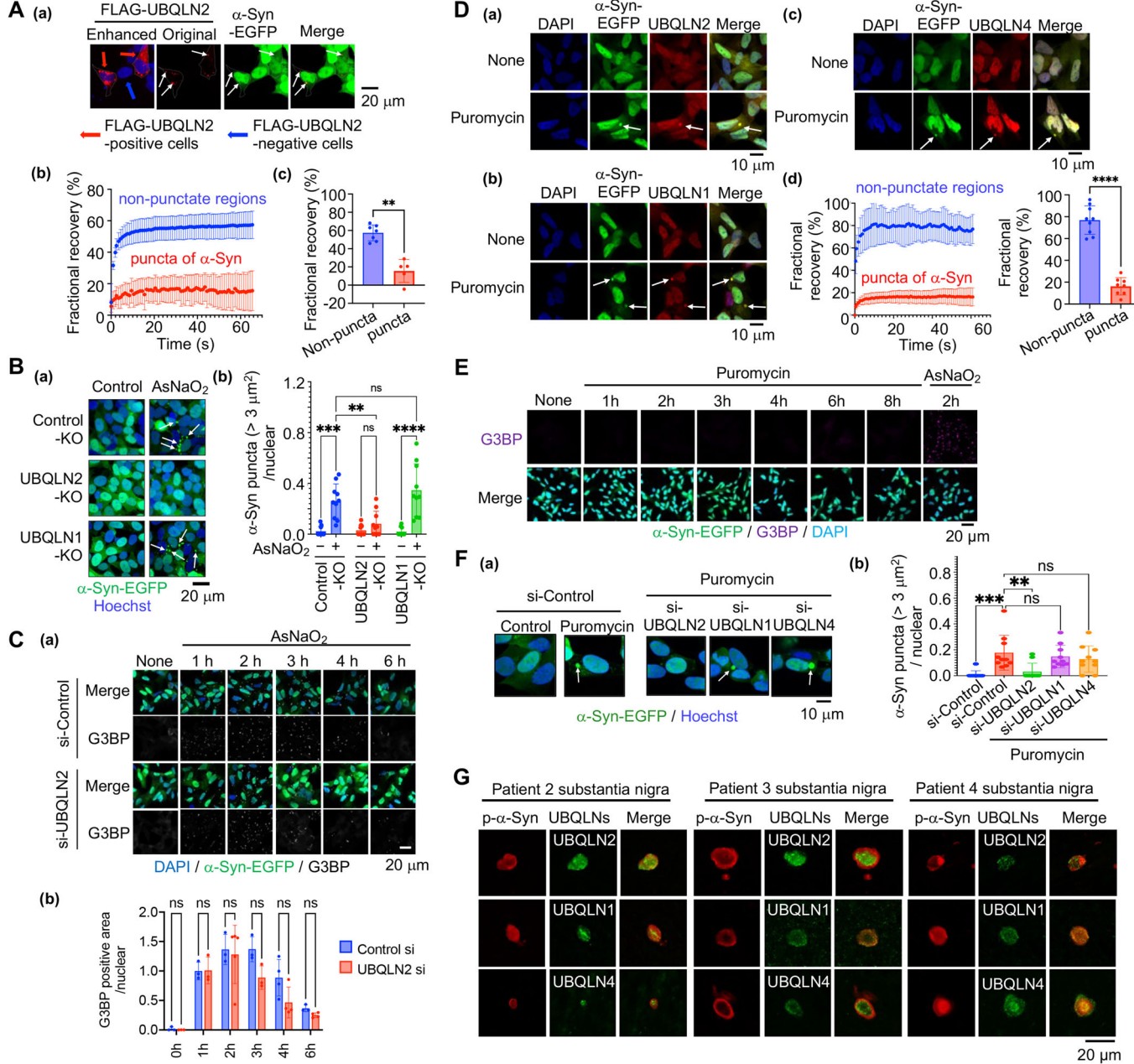

**Figure EV3.** α-Syn aggregates within UBQLN2 droplets in cultured cells.

(**A**) (a) α-Syn(WT)-EGFP/SH-SY5Y cells were transiently transfected with FLAG-UBQLN2 for 24 h and stained with anti-FLAG antibody and DAPI both FLAG-UBQLN2-expressing and non-expressing cells were observed under a microscope. Images in the left column were linearly contrast-enhanced using ImageJ for visualization purposes. At least three experiments were replicated. (b) FRAP analysis of droplets of α-syn puncta at 24 h following transfection. Plots show the average FRAP recovery curves from at least 5 separate droplets. Non-puncta regions were photobleached as a control in the FRAP analysis. **$P < 0.01$ (Mann–Whitney test). Values are the means ± SDs. Non-puncta vs puncta: $P = 0.0025$. (**B**) (a) Fluorescence microscopy of these cells after 12 h $AsNaO_2$ (50 μM) treatment, stained with Hoechst. White arrows indicate large α-syn-EGFP condensates. At least three experiments were replicated. (b) Quantification of condensates ( $> 3\,\mu m^2$) per cell. **$P < 0.01$, ***$P < 0.001$, ****$P < 0.0001$, N.S. (two-way ANOVA, Tukey's test). Ten images per condition. Values of mean ± SD. Control-KO/Control vs Control-KO/$AsNaO_2$: $P = 0.0002$, UBQLN2-KO/Control vs UBQLN2-KO/$AsNaO_2$: $P = 0.9047$, UBQLN1-KO/Control vs UBQLN1-KO/$AsNaO_2$: $P < 0.0001$, Control-KO/$AsNaO_2$ vs UBQLN2-KO/$AsNaO_2$: $P = 0.0092$, Control-KO/$AsNaO_2$ vs UBQLN1-KO/$AsNaO_2$: $P = 0.5394$. (**C**) Fluorescence microscopy of α-syn(WT)-EGFP/SH-SY5Y cells with UBQLN2 knockdown. (a) Cells were transfected with si-Control or si-UBQLN2, treated with 50 μM $AsNaO_2$ for the indicated time points post 48 h transfection, and immunostained with anti-G3BP and DAPI. (b) Quantification of the number of G3BP-positive condensates per cell. The values are normalized to the mean of the Ctrl si group at 1 h. N.S., nonsignificant. Statistical comparisons between control si and UBQLN2 si were performed at each time point using the Mann–Whitney test ($n = 3$–5, technical replicates). 0 h: $P > 0.9999$, 1 h: $P > 0.9999$, 2 h: $P = 0.9881$, 3 h: $P = 0.2596$, 4 h: $P = 0.2488$, 6 h: $P = 0.9704$. (**D**) α-Syn (WT)-EGFP/SH-SY5Y cells were treated with 3 μg/ml puromycin for 8 h and subsequently immunostained with anti-UBQLN2 (a), anti-UBQLN1 (b), or anti-UBQLN4 and DAPI. (c) UBQLN2 droplets and large α-syn-EGFP condensates are indicated by white arrows. At least three experiments were replicated. (d) FRAP analysis was performed on α-syn droplets following treatment with 3 μg/ml puromycin for 8 h. Plots show the average FRAP recovery curves from at least 8 separate droplets. Values are the mean ± SD. Quantification of fractional recovery at 70 s after photobleaching from at least 8 separate droplets. Values are the means ± SDs. Non-puncta regions were photobleached as a control in the FRAP analysis. ****$P < 0.001$ (Mann–Whitney test). Non-puncta vs puncta: $P < 0.0001$. (**E**) α-Syn(WT)-EGFP/SH-SY5Y cells were treated with 3 μg/ml puromycin for the indicated time points or 50 μM $AsNaO_2$ for 2 h, and subsequently immunostained with anti-G3BP and DAPI. (**F**) (a) Fluorescence microscopy of α-syn(WT)-EGFP/SH-SY5Y cells with UBQLN knockdown. Cells transfected with si-Control, si-UBQLN2, si-UBQLN1, or si-UBQLN4 were treated with puromycin (10 μg/mL, 6 h) and stained with Hoechst. (b) Quantification of condensates ( $> 3\,\mu m^2$) per cell. **$P < 0.01$, ***$P < 0.001$, N.S. (Dunnett's test, versus si-Control/puromycin). Ten images per condition; mean ± SD. si-Control (Puromycin) vs si-Control: $P = 0.0006$, si-Control(Puromycin) vs si-UBQLN2(Puromycin): $P = 0.0028$, si-Control(Puromycin) vs si-UBQLN1(Puromycin): $P = 0.8805$, si-Control(Puromycin) vs si-UBQLN4(Puromycin): $P = 0.5718$. (**G**) Accumulation of UBQLN2 in LBs of sporadic PD patients (patient 2 ~ 4). Sections from the midbrain of pathologically diagnosed PD cases were stained with each anti-UBQLN antibodies or anti-p-α-Syn antibody.

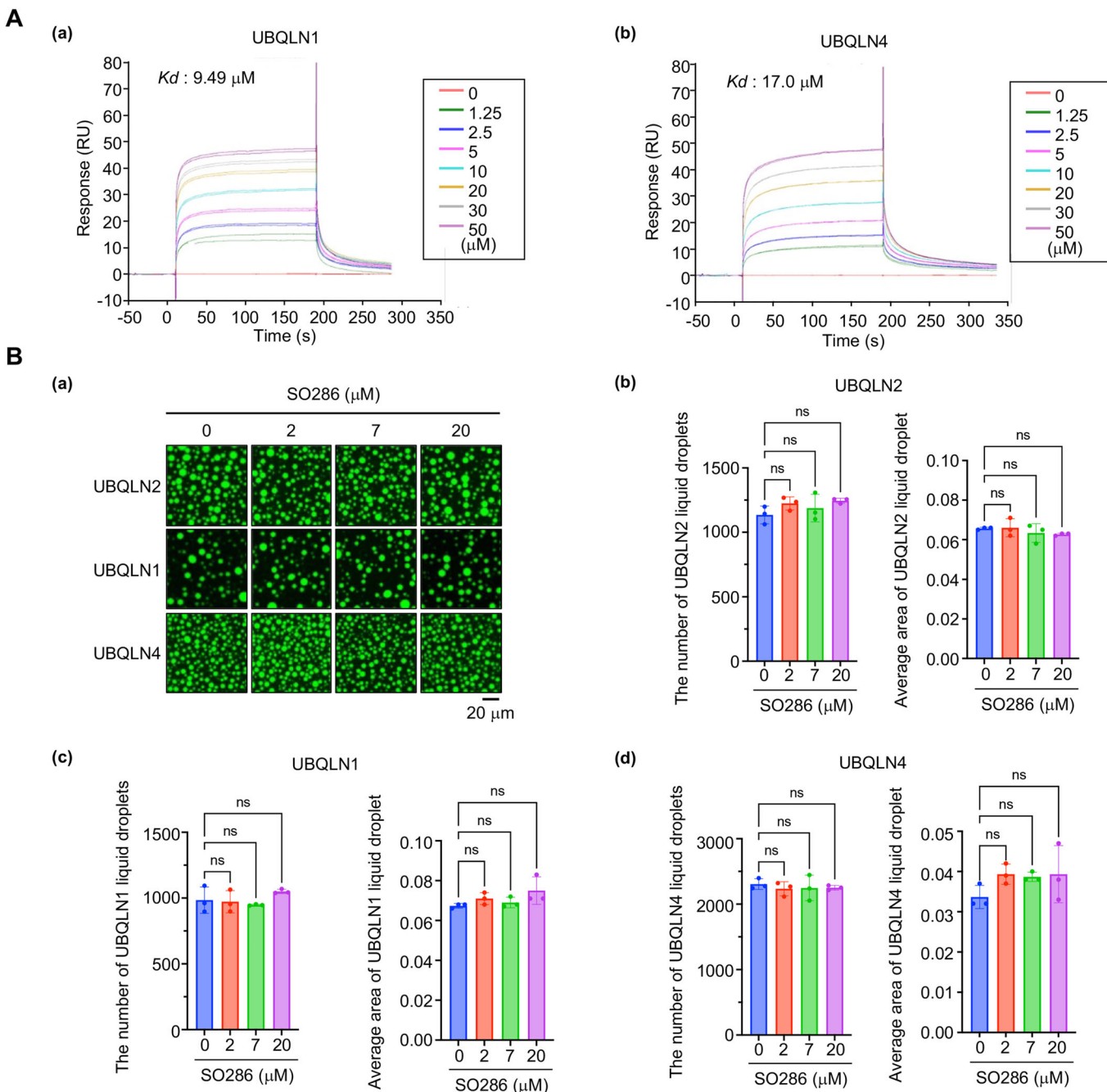

**Figure EV4. SO286 binds to the STI1 regions of UBQLN2.**

(**A**) Surface plasmon resonance analysis of biotin–SO286 reactivity with UBQLN1 (a) and UBQLN4 (b). The vertical axis represents the amount of binding in the resonance unit (RU). (**B**) (a) Fluorescence microscopy images showing 10 μM solutions of each UBQLN construct (1% DyLight488-labeling) after incubation at 37 °C for 24 h in the presence of SO286. The number and size of the droplets of UBQLN2 (b), UBQLN1 (c), and UBQLN4 (d) were quantified. N.S., nonsignificant (Dunnett's test, versus SO286 [0 μM]). Three images were quantified per condition. Values are the means ± SDs. (b) The number of UBQLN2 liquid droplets; 0 μM vs 2 μM: $P = 0.3283$, 0 μM vs 7 μM: $P = 0.6847$, 0 μM vs 20 μM: $P = 0.1971$. Average area of UBQLN2 liquid droplet; 0 μM vs 2 μM: $P = 0.9986$, 0 μM vs 7 μM: $P = 0.7304$, 0 μM vs 20 μM: $P = 0.5769$. (c) The number of UBQLN1 liquid droplets; 0 μM vs 2 μM: $P = 0.9918$, 0 μM vs 7 μM: $P = 0.8381$, 0 μM vs 20 μM: $P = 0.5056$. Average area of UBQLN1 liquid droplet; 0 μM vs 2 μM: $P = 0.5748$, 0 μM vs 7 μM: $P = 0.9208$, 0 μM vs 20 μM: $P = 0.1141$. (d) The number of UBQLN4 liquid droplets; 0 μM vs 2 μM: $P = 0.8147$, 0 μM vs 7 μM: $P = 0.8762$, 0 μM vs 20 μM: $P = 0.8955$. Average area of UBQLN4 liquid droplet; 0 μM vs 2 μM: $P = 0.2767$, 0 μM vs 7 μM: $P = 0.3611$, 0 μM vs 20 μM: $P = 0.2767$. Source data are available online for this figure.

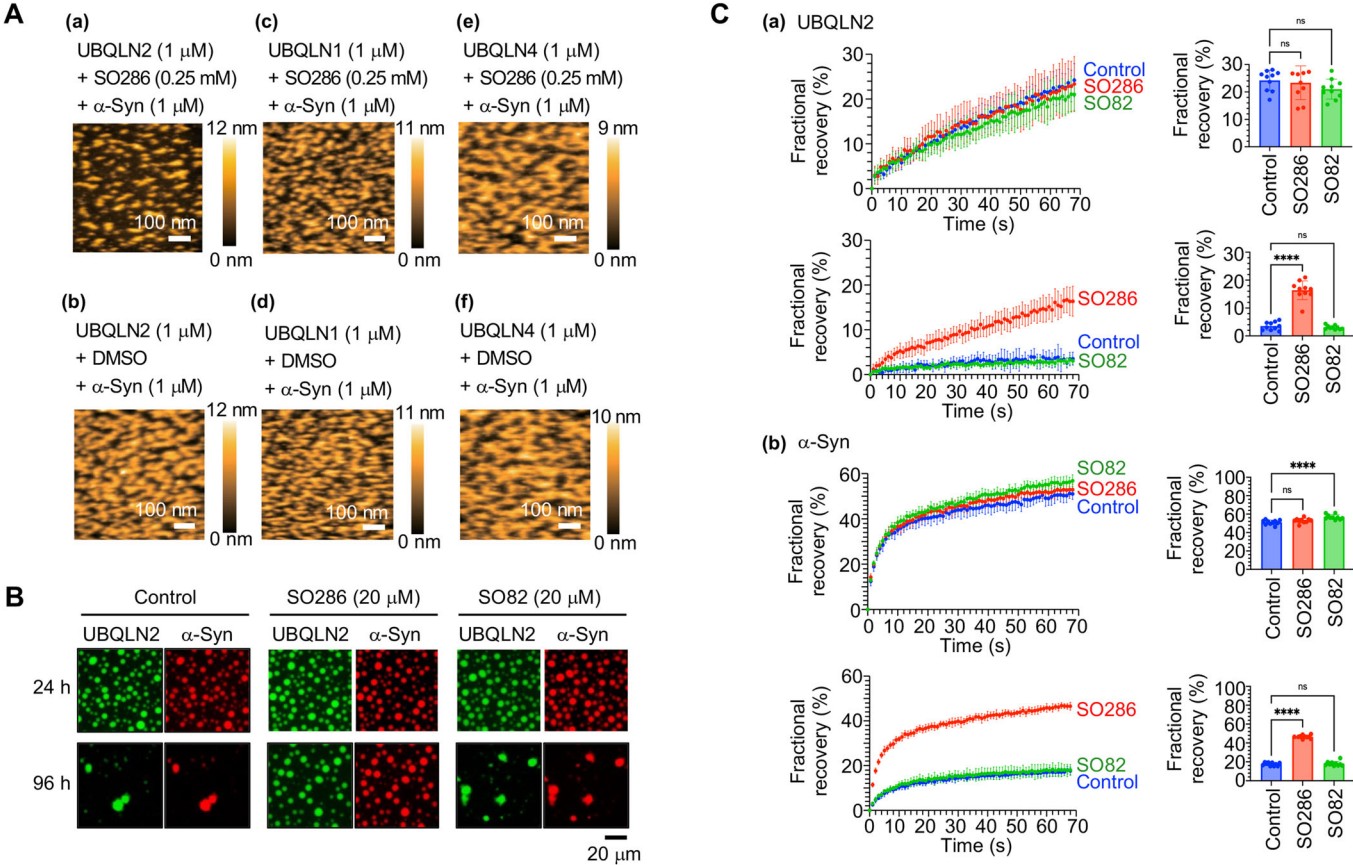

**Figure EV5. SO286 inhibits α-syn aggregation within UBQLN2 droplets in vitro.**

(**A**) Effects of SO286 on the spinodal decomposition-like patterns of α-syn and UBQLN2 (a, b), UBQLN1 (c, d), or UBQLN4 (e, f). Each UBQLN (1 μM) was premixed with (a, c, e) or without (b, d, f) SO286 (0.25 mM) before being mixed with α-syn (1 μM). (**B**) Fluorescence microscopy images of mixed solutions of 10 μM UBQLN2 construct (1% DyLight488-labeling) and 10 μM α-syn construct (1% DyLight633-labeling) after incubation at 37 °C for 24 and 96 h in the presence or absence of 20 μM SO286 or SO82. (**C**) FRAP analysis of droplets of UBQLN2 incorporating α-syn in the presence or absence of 20 μM SO286 or SO82 and quantification of fractional recovery at 70 s after photobleaching from 10 separate droplets. ****$P < 0.0001$, N.S., nonsignificant (Dunnett's test, versus Control). Values are the means ± SDs. (a) 24 h/Control vs 24 h/SO286: $P = 0.8863$, 24 h/Control vs 24 h/SO82: $P = 0.2308$, 96 h/Control vs 96 h/SO286: $P < 0.0001$, 96 h/Control vs 96 h/SO82: $P = 0.8253$. (b) 24 h/Control vs 24 h/SO286: $P = 0.1874$, 24 h/Control vs 24 h/SO82: $P < 0.0001$, 96 h/Control vs 96 h/SO286: $P < 0.0001$, 96 h/Control vs 96 h/SO82: $P = 0.8330$. Source data are available online for this figure.

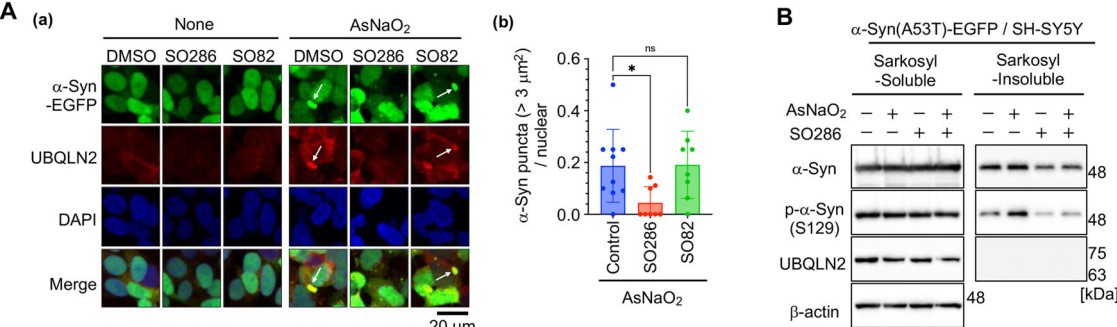

**Figure EV6.  SO286 inhibits α-syn aggregation within UBQLN2 droplets in cultured cells.**

(A) (a) a-Syn(WT)-EGFP/SH-SY5Y cells were treated with 50 μM AsNaO$_2$ for the indicated time points in the presence of 20 μM SO286 or SO82 and immunostained with anti-UBQLN2 and DAPI. UBQLN2 droplets and large α-syn-EGFP condensates are indicated by white arrows. Control vs SO286: $P = 0.0341$, Control vs SO82: $P = 0.9969$. (b) Quantification of the number of α-syn-EGFP puncta ( $> 3\,\mu m^2$) per cell. *$P < 0.05$, N.S., nonsignificant (Dunnett's test, versus AsNaO$_2$/DMSO). Nine images were quantified per condition. Values are the means ± SDs. (B) α-Syn(A53T)-EGFP/SH-SY5Y cells were treated with 50 μM AsNaO$_2$ for 12 h in the presence of 20 μM SO286. Sarkosyl-soluble and -insoluble fractions were subjected to SDS-PAGE and immunoblotted with the indicated antibodies. Representative blot shown, $n = 2$. Source data are available online for this figure.

