## [Peer Review File · The EMBO Journal]

Ubiquilin-2 liquid droplets catalyze α -synuclein fibril formation

Tomoki Takei, Yukiko Sasazawa, Daisuke Noshiro, Mitsuhiro Kitagawa, Tetsushi Kataura, Hiroko Hirawake-Mogi, Emi Kawauchi, Yuya Nakano, Etsu Tashiro, Tsuyoshi Saitoh, Shigeru Nishiyama, Seiichiro Ogawa, Soichiro Kakuta, Saiko Kazuno, Yoshiki Miura, Daisuke Taniguchi, Viktor Korolchuk, Nobuo Noda, Shinji Saiki, Masaya Imoto, and Nobutaka Hattori

Corresponding author(s): Masaya Imoto (m.imoto.xz@juntendo.ac.jp) , Nobutaka Hattori (nhattori@juntendo.ac.jp)

Review Timeline:

Transfer from Review Commons:	18th Jul 25
Editorial Decision:	8th Aug 25
Revision Received:	1st Sep 25
Editorial Decision:	11th Sep 25
Revision Received:	16th Sep 25
Accepted:	26th Sep 25

Editor: Hartmut Vodermaier

Transaction Report: This manuscript was transferred to The EMBO JOURNAL following peer review at Review Commons.

Review #1

1. Evidence, reproducibility and clarity:

Evidence, reproducibility and clarity (Required)

This manuscript hypothesizes that UBQLN2 uses its phase separation propensity to recruit and promote a-synuclein aggregation within UBQLN2 droplets. The authors use a combination of in vitro and cell culture-based experiments to examine the interactions between UBQLNs and a-synuclein. Using several deletion constructs of UBQLNs, the authors attempt to determine the domains responsible for the interaction between UBQLN2 and a-synuclein. They narrow down the UBQLN2 interaction region to the middle STI1 domains, although some of their data suggest complications in their interpretations. In any case, they suggest that UBQLN2 may promote a-synuclein aggregation. This latter point complements other phase separation studies as it has been proposed that several other aggregation-prone proteins can be driven towards aggregation via phase separation of the aggregation-prone protein itself or recruitment to condensates made by other proteins that phase separate. The authors show that a-synuclein is recruited to stress-induced UBQLN2 puncta in cells. The novelty of this study is the suggestion that a compound, specifically SO286, binds to the STI1 domain, interferes with UBQLN2 association with itself, and further affects the interaction between UBQLN2 and a-synuclein (reducing the propensity for a-synuclein to aggregate in UBQLN2 droplets in vitro and in cells). UBQLN2 droplets in cells appear smaller in the presence of SO286. While the study presents SO286 as a potential tool to disrupt UBQLN2 functions and disrupt UBQLN2-a-synuclein interactions, many questions remain unanswered. As will be mentioned in more detail below, several controls are necessary to investigate how much a-synuclein affects UBQLN droplets both in vitro and in cells. Additionally, the molecular mechanisms underlying how a-synuclein interacts with UBQLN2 remain speculative. Some of the data presented within the manuscript raise questions as to which domains control interactions between UBQLN2 and a-synuclein - is it just STI1-2 or a combination of domains? Lastly, a big question still remains regarding the physiological connection between UBQLN2 and a-synuclein interactions. While the work highlights the potential for this interaction, extension into other cell models and/or primary cell types (e.g., neurons) would further raise the relevance and profile of this work. As it currently stands, the work is preliminary.

****Major comments:****

1) The data highlighting dUBL UBQLN2 phase separation without a-synuclein recruitment

contrasts with data later in the manuscript where the authors suggest that a-synuclein interacts with UBQLN2 via the STI1 domains. These results are inconsistent and need to be explained. The lack of a-synuclein recruitment to dUBL droplets suggests that a-synuclein may need the UBL region to interact, unless I am misunderstanding?

2) The experiments in Figure S1B focus on different UBQLNs and the effect of a-syn on their dissolution by hexanediol. However, the important controls of what happens without a-syn are not shown. How much of the observed effects are due to a-syn vs. the intrinsic ability of UBQLNs to respond to hexanediol over time? On a similar note, FRAP experiments should be repeated without added Syn to assess whether a-syn is changing UBQLN properties or if these are intrinsic properties of UBQLNs?

3) While the focus of this work is not on different UBQLNs, the different properties with a-syn could stem from differences on how a-syn influences the phase diagram of the different UBQLNs. Have the authors considered experiments where they add a-syn to UBQLN conditions where they do not phase separate to determine how/if a-syn influences the conditions for when phase separation of UBQLNs is observed? Furthermore, considering the focus of this work on UBQLN2 droplets (phase separation), it will be important to examine several concentrations of both UBQLNs and a-syn rather than just one fixed set of concentrations.

4) The authors correctly point out that it remains controversial whether a-synuclein's ability to phase separate is linked to its complex set of biological functions (still being determined). An area that could be addressed within this work is the quantitative effect on a-synuclein phase separation, such as the determination of the threshold concentration (saturation concentration) upon which a-synuclein phase separates. A comparison of this in the absence and presence of UBQLN2 would be helpful.

5) While a-syn binding to UBQLN2 is affected by the STI1-2 variant, you can still see a-synuclein be recruited into STI1-2 droplets in Figure S1A. What's the explanation?

6) From looking at the WBs, you need at least one STI1 domain around for binding between a-syn and UBQLN2 to occur. This is made more complicated still since it seems that a-syn can bind to dSTI1-1 but can't bind to dSTI1-2 (main figure). Sample 9 doesn't have STI1-2 but it seems to bind just fine (S2B). I think more work needs to be done to validate that the STI1-2 domain is the 'primary binding site for a-syn'. Prior work has also suggested that STI1-2 contributes to UBQLN oligomerization and I am not sure how that pertains to a-syn interactions with UBQLN2.

7) The FRAP experiments in Figure 3C do not appear conclusive at first glance. The authors mention that a-syn aggregates inside UBQLN2 droplets over time, although the main difference here is a 5-10% change in fractional recovery percentage. The overall rate appears similar across 3h, 6, and 12 h (although this isn't quantified).

****Minor comments:****

1) Recombinant a-synuclein does not undergo LLPS - but these images are not presented here.

2) I do not understand Figure 1D. What are we imaging in the UBQLN2 column if there is no a-syn but immunostaining against a-syn?

3) Are the droplets of UBQLN2 stress granules in Figure 3?

4) On pg. 6-7, there is a mention that "SO286 was revealed to bind to UBQLNs via a pulldown assay" - is this data published elsewhere as I didn't see a reference.

2. Significance:

Significance (Required)

****General assessment:**** To this reviewer, it is not surprising that there is an effect between UBQLN2 and a-synuclein considering the stickiness of a-synuclein and its propensity to aggregate. More challenging is assessing the significance of this interaction. As mentioned above, the novelty of this study is the identification of a compound, specifically SO286, that binds to the ST11 domain of UBQLN2, interferes with UBQLN2 association with itself, and further affects the interaction between UBQLN2 and a-synuclein (reducing the propensity for a-synuclein to aggregate in UBQLN2 droplets in vitro and in cells). Strengths of the study include the combination of in vitro and cell-based studies examining UBQLN/a-synuclein recruitment, although the effects of introducing SO286 into cells is not extensively characterized. In its current form, limitations of the work include preliminary molecular-based insights on the interaction between UBQLN2 and a-synuclein, and similar issue with the SO286 compound.

****Advance:**** Prior work has already suggested interactions between ubqln2 and a-synuclein (see work from the Paulson group that has also been cited here). Other work has suggested that a-synuclein can be driven to aggregate via phase separation of itself or other

interacting components. A potential advance of this work could be identification of the molecular interactions between α -synuclein and ubqln2 but this is still preliminary (see comments above). A second potential advance is the identification of the SO286 compound in probing STI1-mediated interactions.

Audience: Basic research communities interested would be the neurodegeneration field broadly, specifically Parkinson's and ALS subfields. Additional interest could come from researchers in the phase separation and aggregation fields.

****Expertise:**** Biophysics and cell biology of proteins involved in ubiquitin-associated protein degradation pathways. I am currently unable to review the TEM-based studies.

3. How much time do you estimate the authors will need to complete the suggested revisions:

Estimated time to Complete Revisions (Required)

(Decision Recommendation)

Between 1 and 3 months

4. Review Commons values the work of reviewers and encourages them to get credit for their work. Select 'Yes' below to register your reviewing activity at Web of Science Reviewer Recognition Service (formerly Publons); note that the content of your review will not be visible on Web of Science.

Yes

Review #2

1. Evidence, reproducibility and clarity:

Evidence, reproducibility and clarity (Required)

****Summary:****

This manuscript presents compelling evidence that UBQLN2 promotes α -synuclein (α -syn) fibril formation by recruiting α -syn, in a UBQLN2 UBL-domain-dependent manner, into its phase-separated condensates, where aggregation is triggered during the transition from

liquid to gel/solid states. The authors demonstrate that this process occurs under physiological conditions and does not require α -syn to undergo LLPS independently. While UBQLN1 and UBQLN4 can incorporate α -syn into their condensates, they do not promote α -syn aggregation, indicating a UBQLN2-specific phenomenon. UBQLN2 and α -syn directly interact, with the interaction mediated by UBQLN2's STI1 domains-particularly STI1-2- though it remains unclear whether this binding event is the primary driver of α -syn aggregation within droplets. Aggregation of α -syn within UBQLN2 droplets is observed both in vitro and in cultured cells under stress conditions that trigger condensate formation. A small molecule, SO286, disrupts both UBQLN2 self-association and its interaction with α -syn, thereby preventing α -syn aggregation without impairing UBQLN2 droplet formation. These findings reveal a previously unrecognized mechanism by which UBQLN2 LLPS can drive α -syn aggregation and highlight an STI1-targeting compound as a potential therapeutic strategy for Parkinson's disease.

****Major Comments:****

1. Droplets formed by UBQLN2 under sodium arsenite stress are almost certainly stress granules (SGs), which are known to be condensates composed of many proteins. Since these other proteins often contain intrinsically disordered regions and are prone to aggregation themselves, it's difficult to conclude that thioflavin staining in this context is detecting α -synuclein fibrils specifically. It may instead be binding to another aggregation-prone component of the SG. Given that SGs are thought to be sites of protein aggregation during prolonged stress, additional specificity is needed. To directly demonstrate α -synuclein fibril formation in SGs, immuno-EM imaging would be a more convincing approach-particularly if the conclusions rely heavily on data from arsenite-induced condensates.

2. Similarly, the slowed FRAP recovery of α -synuclein in SGs could simply reflect the gel-like nature of these structures, rather than true fibril formation. A useful control would be to compare FRAP kinetics of α -synuclein to other SG proteins, such as G3BP1 (which does not form fibrils) or TIAR1. If those proteins show faster recovery while α -synuclein remains immobile, it would strengthen the case for fibril formation.

3. Puromycin-induced UBQLN2 droplets, which are not stress granules, might offer a cleaner system to study α -synuclein fibrillization. Repeating the FRAP experiments in these droplets could be informative-if α -synuclein also shows slowed recovery here, it would lend further support to the conclusions drawn from the arsenite-treated SG model.

4. The claim that only UBQLN2 triggers α -synuclein condensation under puromycin treatment is not fully supported, as it's unclear whether puromycin induces condensate formation by UBQLN1 or UBQLN4 in the first place. Without showing that UBQLN1 or UBQLN4 form puncta under these conditions, it's difficult to conclude that their inability to promote α -synuclein condensation is due to a functional difference rather than a lack of condensate formation. The authors should include controls demonstrating whether UBQLN1 and UBQLN4 also phase separate under puromycin treatment to support this conclusion.

5. Lack of quantification throughout the paper or western blots, FRAP decreases the rigour and therefore significance of the studies. It is critical to substantiate some claims with quantification. Important examples include Figure 2A, 2B. Figure 3E, Figure 5H and Figure S4C.

****Optional:****

SO286 binds the STI1 regions of UBQLN2, which are also implicated in α -synuclein binding. While the compound clearly inhibits fibril formation, it does not prevent initial α -synuclein incorporation into droplets, leaving it unclear whether the observed inhibition is due to disrupted recruitment, blocked oligomerization, or both. Without a direct comparison between STI1-1 and STI1-2 deletion mutants and their sensitivity to SO286, the mechanism remains ambiguous. The authors could choose to test SO286 effects in cells or in vitro using UBQLN2 constructs lacking either STI1-1 or STI1-2 to clarify whether the compound's action overlaps with the regions required for direct α -synuclein binding and/or UBQLN2 self-interaction. This would help define the structural basis of SO286's inhibitory effect.

****Minor Comment:****

Typographical correction: Page 7, line 9-correct "UNQLN4" to "UBQLN4."

2. Significance:

Significance (Required)

This paper presents a strong case that UBQLN2 promotes α -synuclein fibril formation through a stress-induced phase transition, and it introduces SO286 as a potential way to block that process. The manuscript is well-written, the figures are clear, and most of the conclusions are supported by the data. That said, the study would benefit from deeper analysis of the stress-induced condensates in cells, consideration of alternative

explanations, and more thorough quantification in key experiments.

- Even with those caveats, the central idea-that UBQLN2 plays an active role in α -syn aggregation rather than just being present-is an important one and opens up a new direction for therapeutic exploration. The data on α -synuclein fibril formation within UBQLN2 droplets and the mechanistic insights will be especially relevant to researchers working on protein aggregation in neurodegenerative disease, LLPS, and those focused on UBQLN2 or α -synuclein. The SO286 findings extend the paper's relevance to a broader neurodegeneration audience, particularly those interested in translational impact. Overall, this work moves the field forward in a meaningful way.

Comparable papers might include:

ALS-linked mutations impair UBQLN2 stress-induced biomolecular condensate assembly in cells.

Riley JF, Fioramonti PJ, Rusnock AK, Hehnlly H, Castañeda CA. *J Neurochem*. 2021 Oct;159(1):145-155. doi: 10.1111/jnc.15453. Epub 2021 Aug 20. PMID: 34129687 Free PMC article.

- alpha-Synuclein aggregation nucleates through liquid-liquid phase separation.

Ray S, Singh N, Kumar R, Patel K, Pandey S, Datta D, Mahato J, Panigrahi R, Navalkar A, Mehra S, Gadhe L, Chatterjee D, Sawner AS, Maiti S, Bhatia S, Gerez JA, Chowdhury A, Kumar A, Padinhateeri R, Riek R, Krishnamoorthy G, Maji SK. *Nat Chem*. 2020 Aug;12(8):705-716. doi: 10.1038/s41557-020-0465-9. Epub 2020 Jun 8. PMID: 32514159

- Ubiquitin Modulates Liquid-Liquid Phase Separation of UBQLN2 via Disruption of Multivalent Interactions.

Dao TP, Kolaitis RM, Kim HJ, O'Donovan K, Martyniak B, Colicino E, Hehnlly H, Taylor JP, Castañeda CA. *Mol Cell*. 2018 Mar 15;69(6):965-978.e6. doi: 10.1016/j.molcel.2018.02.004. Epub 2018 Mar 8. PMID: 29526694

- Ubiquilin-2 regulates pathological alpha-synuclein.

Sandoval-Pistorius SS, Gerson JE, Liggans N, Ryou JH, Oak K, Li X, Negron-Rios KY, Fischer S, Barsh H, Crowley EV, Skinner ME, Sharkey LM, Barmada SJ, Paulson HL. *Sci Rep*. 2023 Jan 6;13(1):293. doi: 10.1038/s41598-022-26899-0.

Keywords for my field of expertise: stress granules, UBQLN, neurodegenerative disease, LLPS, ALS/FTD

I do not have enough expertise to confidently evaluate the HS-AFM imaging or Surface plasmon resonance (SPR) analysis.

3. How much time do you estimate the authors will need to complete the suggested revisions:

Estimated time to Complete Revisions (Required)

(Decision Recommendation)

Between 1 and 3 months

4. Review Commons values the work of reviewers and encourages them to get credit for their work. Select 'Yes' below to register your reviewing activity at Web of Science Reviewer Recognition Service (formerly Publons); note that the content of your review will not be visible on Web of Science.

Yes

Review #3

1. Evidence, reproducibility and clarity:

Evidence, reproducibility and clarity (Required)

The study by Takei et al. has examined how phase separation of alpha-synuclein, the major aggregating protein in Parkinson's disease, is influenced by Ubiquilin-2 (UBQLN2), since previous studies had implicated UBQLN2 in various neurodegenerative diseases, including Parkinson's disease. UBQLN2 itself is a protein that can undergo phase separation and partition into stress granules in cells and functions in transferring ubiquitinated proteins to the proteasome or autophagy. The authors used recombinant a-syn and UBQLN2 to demonstrate that a-syn is incorporated into UBQLN2 droplets and that this leads to irregularly shaped, undynamic a-syn-UBQLN2 co-condensates, from which fibrillar a-syn species emerge. They then use SH-SY5Y cells stably expressing EGFP-tagged a-syn to demonstrate that AsNaO2 stress, which induces droplet-like structures of UBQLN2 in cells, leads to co-condensation of UBQLN2 and a-syn in the cytoplasmic puncta, which gradually lose their dynamics and become Thioflavin S-positive. Finally, they utilize a compound (SO286), which they had previously identified in a compound screen to

selectively bind UBQLN2 with high affinity at the STI-1 and STI-2 regions. The authors show that this compound is able to suppress the UBQLN2-a-syn interaction and thus suppresses the UBQLN2-induced liquid-to-solid transition of a-syn in vitro and in cells.

The study still lacks a few important controls and has some obvious gaps and shortcomings in its presentation that should be addressed before publication. The methods section lacks clarity and misses important information, and the discussion could better discuss their findings in light of previous findings on UBQLN2 and stress granules as well as the limitations of their approaches. I summarize my major and minor comments below.

****Major points:****

1. A key question to address is whether the observed effect is specific to a-synuclein. How about other disease-linked phase separating proteins, e.g. TDP-43 or FUS (since UBQLN2 has been linked to ALS/FTD) or Tau? At least one of them should be used as a control protein, to get an idea whether this is an a-syn-specific effect or not.

2. A related question is: How about other (non-UBQLN) condensate systems formed in presence of 3% PEG - does a-syn partition into them as well? The authors demonstrate that a-syn also partitions into UBQLN1 and UBQLN4 condensates, which raises the question whether it will partition into any condensate under those experimental conditions. Also here, checking for specificity would be of great importance and highly informative.

3. The interaction assays described/shown in Fig. 2 and Fig.5A need clarification, as the methods are not sufficiently described. I could not figure out whether this was a pure in vitro interaction assay (with recombinant proteins and which buffer was used), or whether it involved a cell lysate/pulldown from cell lysate, in which case the interaction could also be indirect. The results text on p. 5 implies it is a pure in vitro interaction assay with recombinant proteins, but there is no corresponding methods part, and the so-called "in vitro pulldown assays" described in the methods on p. 11 involves transfected HEK293 cells, generation of a lysate and pulldown with flag beads, which is not a true "in vitro" interaction assay and one cannot claim that the observed interaction is really direct. Hence, the missing methods should be added, or the manuscript text should be corrected to accurately describe the interaction assays shown in Fig. 2A,B and Fig. 5A, as I could not figure out how they were really done. What was the reason why two different interaction assays (Pulldown, Fig. 2A, and Far Western, Fig. 2B) were used? It would be more convincing if the same assay was used to check interaction of the different UBQLNs

(UBQLN1,2,4) and the different UBQLN2-mutants (WT, deltaSTI-1, deltaSTI-2), instead of using two completely different assays. The rationale for using these assays and what exactly was done should be given, otherwise it is hard to follow this part of the manuscript.

4. For the AsNaO₂- or puromycin-induced UBQLN2 condensates in cells (Fig. 3), it would be important to clarify whether these structures are stress granules (SGs), e.g. by co-staining for established SG marker proteins. This would be expected, as previous studies have reported that UBQLN2 can be detected in SGs upon stress (e.g. Dao et al., 2018). If the authors find that the structures are indeed SGs, this would be highly interesting, because it would for the first time demonstrate that a-syn can also accumulate in SGs (or SG-like structures) after stress. If the structures do NOT overlap with known SG markers, it would be informative to know this as well and to investigate the nature of the biomolecular condensates (in future studies). Checking for the presence of SGs in their systems also seems important because of the knockout/knockdown experiments shown in Fig. 3, as previously UBQLN2 knockdown was shown to enhance SG numbers, whereas UBQLN2 overexpression was shown to suppress SGs (Alexander et al., PNAS 2018, PMID: 30442662). The discrepancies/ overlap to the findings by Alexander et al. should be discussed in the manuscript, so far I did not see this study mentioned.

5. Co-condensation of UBQLN2 and a-syn in cells was seen with EGFP-tagged a-synuclein stably expressed in SH-SY5Y cells. Can the authors also demonstrate the same co-condensation for endogenous a-synuclein, i.e. with antibody staining (in the absence of potential overexpression and EGFP-tagging), or in a more physiological model system of a-syn aggregation? Does UBQLN2 (but not UBQLN1 and 4 co-localize with a-syn aggregates in PD patients? These questions may go beyond the scope of this manuscript, but depending on the journal they aim for, the editors may want to see such data, and at minimum these points (limitations of current approach) should be addressed in the discussion.

6. The authors show that only small a-syn condensates can be observed in UBQLN2 KO cells or upon siRNA-mediated UBQLN2 silencing (Fig. 3D). Did they check what happens upon UBQLN2 overexpression, does this promote the formation / enlargement of a-syn condensates?

7. The fractionation into sarkosyl-soluble/insoluble presented in Fig. 3E and 5H is not convincing without a proper quantification, taking the ratio of phosphorylated S129p-Syn to total a-synuclein into account.

8. The data with compound SO286 are quite striking, however it would be important to show a negative control (e.g. compound of similar chemical properties from their compound screen), which does not exert these effects at the same concentrations (especially since the used concentrations are quite high, e.g. 2- 20 μ M in Fig. 4E).

****Minor points:****

1. Methods part (p. 11) "Expression and purification of recombinant proteins" is missing a-syn expression / purification protocol, which construct was used etc. For all proteins, the OD260/280 ration should be given, to give an idea about the degree of nucleic acid contamination.

2. Methods part (p.11) "Imaging of LLPS" should be rephrased as "Formation and imaging of condensates" and more details should be given, i.e. temperature, observation time, was the fluorescence microscope a wide-field or confocal microscope etc.

3. Fig. 5C: remove horizontal lines behind the bars and describe quantification in methods. Why was the "area covered by proteins" quantified here? This does not seem to be the best parameter when analyzing morphological changes... a parameter like roundness seems better suited.

4. Fig. 5C not mentioned in results text. Why does the legend say 0.25 mM SO286, whereas Fig. 5B implies that 20 μ M of the compound were used? Is Fig. 5C not a quantification of Fig. 5B, but a totally different experiment? These points need clarification.

2. Significance:

Significance (Required)

The study presents some surprising findings (potential influence of UBQLN2 on a-synuclein aggregation; potential recruitment of a-syn into SGs or SG-like structures upon stress, druggability of the interaction), and utilizes a large variety of methods (both in vitro and in cells). Hence, if the authors can clarify some of the remaining major concerns (see above), it could potentially be of great significance. Currently there is great interest in finding modulators of phase transitions of disease-linked aggregating proteins. Hence, if properly revised and the major concerns raised above can be adequately addressed, the study should be of great interest to the neurodegeneration and phase separation community.

3. How much time do you estimate the authors will need to complete the suggested revisions:

Estimated time to Complete Revisions (Required)

(Decision Recommendation)

Between 3 and 6 months

Yes

Full Revision

Manuscript number: RC-2025-02962

Corresponding author(s): Masaya, Imoto and Nobutaka, Hattori

1. General Statements [optional]

Thank you for reviewing our manuscript entitled “*Ubiquilin-2 liquid droplets catalyze α -synuclein fibril formation*” (RC-2025-02962) and for giving us the opportunity to submit a revised version. We have carefully considered all the reviewers’ comments and conducted the necessary experiments accordingly. Below, we provide point-by-point responses to each of the reviewers’ questions. All revised text is highlighted in blue in the revised manuscript. Page and line numbers refer to the MS Word version of the manuscript.

We believe that the revised manuscript fully addresses the concerns raised by the reviewers. Finally, all authors would like to express their sincere gratitude to the Editor and Reviewers for their insightful comments and constructive suggestions, which have greatly improved the quality of our manuscript.

Reviewer #1 (Evidence, reproducibility and clarity (Required)):

This manuscript hypothesizes that UBQLN2 uses its phase separation propensity to recruit and promote α -synuclein aggregation within UBQLN2 droplets. The authors use a combination of in vitro and cell culture-based experiments to examine the interactions between UBQLNs and α -synuclein. Using several deletion constructs of UBQLNs, the authors attempt to determine the domains responsible for the interaction between UBQLN2 and α -synuclein. They narrow down the UBQLN2 interaction region to the middle STI1 domains, although some of their data suggest complications in their interpretations. In any case, they suggest that UBQLN2 may promote α -synuclein aggregation. This latter point complements other phase separation studies as it has been proposed that several other aggregation-prone proteins can be driven towards aggregation via phase separation of the aggregation-prone protein itself or recruitment to condensates made by other proteins that phase separate. The authors show that α -synuclein is recruited to stress-induced UBQLN2 puncta in cells. The novelty of this study is the suggestion that a compound, specifically SO286, binds to the STI1 domain, interferes with UBQLN2 association with itself, and further affects the interaction between UBQLN2 and α -synuclein (reducing the propensity for α -synuclein to aggregate in UBQLN2 droplets in vitro and in cells). UBQLN2 droplets in cells appear smaller in the presence of SO286. While the study presents SO286 as a potential tool to disrupt UBQLN2 functions and disrupt UBQLN2- α -synuclein interactions, many questions

remain unanswered. As will be mentioned in more detail below, several controls are necessary to investigate how much α -synuclein affects UBQLN droplets both in vitro and in cells. Additionally, the molecular mechanisms underlying how α -synuclein interacts with UBQLN2 remain speculative. Some of the data presented within the manuscript raise questions as to which domains control interactions between UBQLN2 and α -synuclein – is it just STI1–2 or a combination of domains? Lastly, a big question still remains regarding the physiological connection between UBQLN2 and α -synuclein interactions. While the work highlights the potential for this interaction, extension into other cell models and/or primary cell types (e.g., neurons) would further raise the relevance and profile of this work. As it currently stands, the work is preliminary.

Major comments:

1) The data highlighting dUBL UBQLN2 phase separation without α -synuclein recruitment contrasts with data later in the manuscript where the authors suggest that α -synuclein interacts with UBQLN2 via the STI1 domains. These results are inconsistent and need to be explained. The lack of α -synuclein recruitment to dUBL droplets suggests that α -synuclein may need the UBL region to interact, unless I am misunderstanding?

Our response:

Thank you for raising this important point. Although partitioning of client proteins was thought to depend on solvation differences and interactions with scaffold proteins, a recent study by José A. Villegas and Emmanuel D. Levy demonstrated that the intrinsic stickiness of client proteins alone is sufficient to explain their differential partitioning within two phase-separated systems, without the need to consider the composition of the condensate (Protein Science 31: e4361, 2022).

However, our data suggest that a deletion mutant lacking the UBL domain of UBQLN2 formed liquid droplets that failed to incorporate α -syn at 24 h (Fig. EV1B). Therefore, we propose that in our system the UBL domain facilitates α -syn partitioning not by interacting with α -syn, but by altering the physicochemical environment of the condensates.

These sentences have been added to the revised manuscript (Page 10, Lines 30–37).

2) The experiments in Figure S1B focus on different UBQLNs and the effect of $\alpha\text{-syn}$ on their dissolution by hexanediol. However, the important controls of what happens without $\alpha\text{-syn}$ are not shown. How much of the observed effects are due to $\alpha\text{-syn}$ vs. the intrinsic ability of UBQLNs to respond to hexanediol over time? On a similar note, FRAP experiments should be repeated without added Syn to assess whether $\alpha\text{-syn}$ is changing UBQLN properties or if these are intrinsic properties of UBQLNs?

Our response:

Following the reviewer's suggestion, we examined the intrinsic liquid-like properties of each UBQLN droplet using 1,6-hexanediol and FRAP assays in the absence of $\alpha\text{-syn}$. As shown in revised version Fig. EV1E, both assays demonstrated that UBQLN2 initially exhibited a liquid-like state at 24 h, but underwent a liquid-to-gel/solid transition over time, which was not affected by the presence of $\alpha\text{-syn}$. Similarly, UBQLN1 remained in a liquid state at both 24 h and 96 h, while UBQLN4 exhibited a gel-like state at both time points, regardless of the presence or absence of $\alpha\text{-syn}$. These results indicate that $\alpha\text{-syn}$ does not alter the intrinsic phase properties of UBQLNs.

The data and the relevant sentences have been included in the revised manuscript (Page 4, Lines 27–28 and Fig. EV1E).

3) While the focus of this work is not on different UBQLNs, the different properties with α -syn could stem from differences on how α -syn influences the phase diagram of the different UBQLNs. Have the authors considered experiments where they add α -syn to UBQLN conditions where they do not phase separate to determine how/if α -syn influences the conditions for when phase separation of UBQLNs is observed? Furthermore, considering the focus of this work on UBQLN2 droplets (phase separation), it will be important to examine several concentrations of both UBQLNs and α -syn rather than just one fixed set of concentrations.

Our response:

We appreciate the reviewer's comment and agree that this control is important. We have previously performed this experiment, although the data were not included in the original submission. We have now added these results in Fig. EV1A. The data show that 3 μM UBQLN2 only forms very small liquid droplets, and the addition of α -syn (up to 10 μM) does not affect this phase separation. At concentrations above 6 μM , UBQLN2 forms large droplets, and this phase behavior is also not altered by α -syn. Based on these observations, we used 10 μM UBQLN2 and 10 μM α -syn in the subsequent experiments.

4) The authors correctly point out that it remains controversial whether α -synuclein's ability to phase separate is linked to its complex set of biological functions (still being determined). An area that could be addressed within this work is the quantitative effect on α -synuclein phase separation, such as the determination of the threshold concentration (saturation concentration) upon which α -synuclein phase separates. A comparison of this in the absence and presence of UBQLN2 would be helpful.

Our response:

We have previously investigated whether α -syn undergoes phase separation on its own. However, even at concentrations previously reported to induce α -syn phase separation (Nat Chem 12: 705, 2020), only small aggregates were observed, rather than liquid droplets (please see Supporting data below). One possible explanation for this discrepancy is the difference in buffer composition; our study used a buffer previously used in UBQLN2 phase separation experiments (Mol Cell 69: 965, 2018), which differs from that used in earlier α -syn studies.

In contrast, we observed that 1 μ M α -syn was incorporated into 6 μ M UBQLN2, and 0.1 μ M α -syn was incorporated into 10 μ M UBQLN2, which may promote phase separation (please see our response to Major Comment #3 by Reviewer #1). Therefore, the phase separation thresholds of α -syn under different UBQLN2 conditions are as follows: without UBQLN2, > 10 μ M; with 6 μ M UBQLN2, 1 μ M; with 10 μ M UBQLN2, 0.1 μ M.

5) While α -syn binding to UBQLN2 is affected by the STI1-2 variant, you can still see asynuclein be recruited into STI1-2 droplets in Figure S1A. What's the explanation?

Our response:

As shown in revised version Fig. EV1B, the Δ STI1-2/UBQLN2 failed to undergo liquid-liquid phase separation, which is consistent with a previous report (Mol Cell 69, 965-978.e966, 2018), and instead formed small precipitates. Although the STI1-2 region constitutes the primary binding site for α -syn, α -syn was nonetheless recruited to these precipitates formed by the Δ STI1-2/UBQLN2 variant. As demonstrated in revised version Fig. 2B, α -syn weakly bound to the STI1-1 domain of UBQLN2 in addition to STI1-2. Therefore, the recruitment of α -syn to the precipitates formed by the Δ STI1-2/UBQLN2 variant was likely mediated by its interaction with the STI1-1 region. We have added the relevant sentences in the revised manuscript (revised manuscript Fig. 2B; Page 5, Lines 20-22).

6) From looking at the WBs, you need at least one STI1 domain around for binding between α -syn and UBQLN2 to occur. This is made more complicated still since it seems that α -syn can bind to dSTI1-1 but can't bind to dSTI1-2 (main figure). Sample 9 doesn't have STI1-2 but it seems to bind just fine (S2B). I think more work needs to be done to validate that the STI1-2 domain is the 'primary binding site for α -syn'. Prior work has also suggested that STI1-2 contributes to UBQLN oligomerization and I am not sure how that pertains to α -syn interactions with UBQLN2.

Our response:

Indeed α -syn binds to Δ STI1-1/UBQLN2 and shows weak binding to Δ STI1-2/UBQLN2, as shown in revised version Fig. 2B (please see our response to Major Comment #5 by Reviewer #1). These results suggest that in full-length α -syn, STI1-2 serves as the primary binding site.

Fig. EV2

7) The FRAP experiments in Figure 3C do not appear conclusive at first glance. The authors mention that α -syn aggregates inside UBQLN2 droplets over time, although the main difference here is a 5–10% change in fractional recovery percentage. The overall rate appears similar across 3h, 6, and 12 h (although this isn't quantified).

Our response:

In response to the reviewer's comment, we have conducted additional experiments. Specifically, we confirmed that gold colloids labeled the surface of the aggregate structure corresponding to the α -syn(WT)-EGFP puncta, as shown in revised version Fig. 3B. These results provide strong evidence that α -syn undergoes aggregation/fibrillization over time in UBQLN2 droplets upon AsNaO_2 treatment. Furthermore, to demonstrate the time-dependent aggregation of α -syn, we examined the phosphorylation levels of α -syn, a well-established hallmark of α -syn aggregation, in the sarkosyl-insoluble fraction from cells treated with AsNaO_2 for 3, 6, and 12 h. We observed a time-dependent increase in α -syn phosphorylation levels in the sarkosyl-insoluble fraction, as shown in the revised version Fig. 3C and Appendix Fig. S3A. These results provide strong evidence that α -syn undergoes aggregation/fibrillization over time in UBQLN2 droplets upon AsNaO_2 treatment.

Accordingly, the FRAP data originally presented in Fig. 3C has been replaced with immunoelectron microscopy data (revised version Fig. 3B) and the detection of phospho- α -syn in sarkosyl-insoluble fraction (revised version Fig. 3C, and Appendix Fig. S3A).

Additionally, following the suggestion raised by Reviewer #2, ThS staining data in the original manuscript have also been removed.

The relevant text has been updated accordingly (Page 6, Lines 13–27).

Minor comments:

1) Recombinant α -synuclein does not undergo LLPS – but these images are not presented here.

Our response:

In response to the reviewer's suggestion, we have added images showing that recombinant α -syn did not undergo LLPS for up to 20 days, as shown in revised version Appendix Fig. S1A.

2) I do not understand Figure 1D. What are we imaging in the UBQLN2 column if there is no α -syn but immunostaining against α -syn?

Our response:

To confirm that the gold colloid signals detected during α -syn immunostaining were not due to nonspecific binding, a UBQLN2-only sample was included as a negative control. No gold colloid signal was observed in this control, indicating that the signal detected in the sample containing both UBQLN2 and α -syn is specific and not attributable to nonspecific staining. The relevant sentences have been included in the revised text (Page 5, Lines 1–3).

3) Are the droplets of UBQLN2 stress granules in Figure 3?

Our response:

Thank you for raising this important point. When α -syn(WT)-EGFP/SH-SY5Y cells were treated with sodium arsenite (AsNaO_2 , 50 μ M), stress granules (SGs) were formed after 3 h, as indicated by the presence of puncta positive for the SG markers G3BP (shown in revised version Fig. EV3A). Most of these SGs were co-localized with UBQLN2, consistent with previous reports (Mol cell 69: 965, 2018 and PNAS 115: E11485, 2018). Although the G3BP-positive puncta disappeared 6 h after AsNaO_2 treatment, droplets of UBQLN2 containing α -syn remained and continued to grow over time. These results suggest that UBQLN2 droplets are resistant to SG disassembly and undergo a liquid-to-gel phase transition, thereby promoting α -syn aggregation. The data and the relevant sentences have been included in the revised text (Page 6, Lines 8–13 and Fig. EV3A).

4) On pg. 6–7, there is a mention that “SO286 was revealed to bind to UBQLNs via a pulldown assay” – is this data published elsewhere as I didn't see a reference.

Our response:

SO286 (Fig. 4A) was originally identified through a neuroprotective compound screen of our in-house chemical library. We found that SO286, but not SO82, an inactive analog of SO286, directly bound to UBQLN1, UBQLN2, and UBQLN4 using biotinylated SO286 and SO82. These data have not been previously published; therefore, we have included them in revised version Fig. 4A and B. The relevant sentences have been included in the revised text (Page 8, Lines 1–4).

Fig. 4 A

Fig. 4 B

(Fig.4A) Chemical structures of SO286, its inactive analogue SO82, and their biotin-conjugated derivatives. (Fig.4B) Lysates of α -Syn(WT)-EGFP/SH-SY5Y cells were incubated with biotin-SO286 or biotin-SO82. Biotin-labeled compounds were pulled down using streptavidin beads and eluted with biotin. Bound UBQLNs were detected by immunoblotting with specific antibodies.

Reviewer #1 (Significance (Required)):

General assessment: To this reviewer, it is not surprising that there is an effect between UBQLN2 and α -synuclein considering the stickiness of α -synuclein and its propensity to aggregate. More challenging is assessing the significance of this interaction. As mentioned above, the novelty of this study is the identification of a compound, specifically SO286, that binds to the ST11 domain of UBQLN2, interferes with UBQLN2 association with itself, and further affects the interaction between UBQLN2 and α -synuclein (reducing the propensity for α -synuclein to aggregate in UBQLN2 droplets in vitro and in cells). Strengths of the study include the combination of in vitro and cell-based studies examining UBQLN/ α -synuclein recruitment, although the effects of introducing SO286 into cells is not extensively characterized. In its current form, limitations of the work include preliminary molecular-based insights on the interaction between UBQLN2 and α -synuclein, and similar issue with the SO286 compound.

Advance: Prior work has already suggested interactions between ubqln2 and α -synuclein (see work from the Paulson group that has also been cited here). Other work has suggested that α -synuclein can be driven to aggregate via phase separation of itself or other interacting components. A potential advance of this work could be identification of the molecular interactions between α -synuclein and ubqln2 but this is still preliminary (see comments above). A second potential advance is the identification of the SO286 compound in probing ST11-mediated interactions.

Audience: Basic research communities interested would be the neurodegeneration field broadly, specifically Parkinson's and ALS subfields. Additional interest could come from researchers in the phase separation and aggregation fields.

Expertise: Biophysics and cell biology of proteins involved in ubiquitin-associated protein degradation pathways. I am currently unable to review the TEM-based studies.

Reviewer #2 (Evidence, reproducibility and clarity (Required)):

Summary:

This manuscript presents compelling evidence that UBQLN2 promotes α -synuclein (α -syn) fibril formation by recruiting α -syn, in a UBQLN2 UBL-domain-dependent manner, into its phase-separated condensates, where aggregation is triggered during the transition from liquid to gel/solid states. The authors demonstrate that this process occurs under physiological conditions and does not require α -syn to undergo LLPS independently. While UBQLN1 and UBQLN4 can incorporate α -syn into their condensates, they do not promote α -syn aggregation, indicating a UBQLN2-specific phenomenon. UBQLN2 and α -syn directly interact, with the interaction mediated by UBQLN2's STI1 domains—particularly STI1-2—though it remains unclear whether this binding event is the primary driver of α -syn aggregation within droplets. Aggregation of α -syn within UBQLN2 droplets is observed both in vitro and in cultured cells under stress conditions that trigger condensate formation. A small molecule, SO286, disrupts both UBQLN2 self-association and its interaction with α -syn, thereby preventing α -syn aggregation without impairing UBQLN2 droplet formation. These findings reveal a previously unrecognized mechanism by which UBQLN2 LLPS can drive α -syn aggregation and highlight an STI1-targeting compound as a potential therapeutic strategy for Parkinson's disease.

Major Comments:

1. Droplets formed by UBQLN2 under sodium arsenite stress are almost certainly stress granules (SGs), which are known to be condensates composed of many proteins. Since these other proteins often contain intrinsically disordered regions and are prone to aggregation themselves, it's difficult to conclude that thioflavin staining in this context is detecting α -synuclein fibrils specifically. It may instead be binding to another aggregation-prone component of the SG. Given that SGs are thought to be sites of protein aggregation during prolonged stress, additional specificity is needed. To directly demonstrate α -synuclein fibril formation in SGs, immuno-EM imaging would be a more convincing approach—particularly if the conclusions rely heavily on data from arsenite-induced condensates.

Our response:

Thank you for raising this important point. As described in our response to Reviewer #1, Minor comment #3, when α -syn(WT)-EGFP/SH-SY5Y cells were treated with sodium arsenite (AsNaO_2 , 50 μM), stress granules (SGs) formed after 3 h, as indicated by the presence of puncta positive for the SG markers G3BP (shown in revised version Fig. EV3A). Most of these SGs were co-localized with UBQLN2 consistent with previous reports (Mol cell 69: 965, 2018 and PNAS 115:E11485, 2018). However, the G3BP1-positive puncta disappeared 6 h after AsNaO_2 treatment, indicating that SGs had disassembled by this time point. In contrast, the UBQLN2-containing α -syn droplets persisted and continued to grow over time. These findings suggest that UBQLN2 droplets are resistant to SG disassembly and undergo a liquid-to-gel phase transition, thereby promoting α -syn aggregation.

However, as the reviewer rightly pointed out, we cannot exclude the possibility that UBQLN2 condensates at 12 h after AsNaO_2 treatment may contain aggregation-prone components other than α -syn. Therefore, ThS staining of UBQLN2 condensates alone is not sufficient to conclude that α -syn fibrils have formed within these condensates. In accordance with the reviewer's suggestion, we performed immunoelectron microscopy to directly demonstrate α -syn fibril formation within UBQLN2 condensates, as shown in revised version Fig. 3B. We confirmed that gold colloids labeled the surface of the aggregate structure corresponding to α -syn(WT)-EGFP puncta. Accordingly, ThS staining data originally presented in Fig. 3B has been removed and new immunoelectron microscopy data have been added (revised version Fig. 3B). The relevant text has been updated accordingly (Page 6, Lines 13–18).

2. Similarly, the slowed FRAP recovery of α -synuclein in SGs could simply reflect the gel-like nature of these structures, rather than true fibril formation. A useful control would be to compare FRAP kinetics of α -synuclein to other SG proteins, such as G3BP1 (which does not form fibrils) or TIAR1. If those proteins show faster recovery while α -synuclein remains immobile, it would strengthen the case for fibril formation.

Our response:

Thank you for the insightful comments. As described in our response to Major Comment #1 of Reviewer #2, AsNaO₂-induced SGs were observed at 3 h and co-localized with UBQLN2 droplets, but they disassembled by 6 h. Therefore, we believe that FRAP analysis using well-established SG components, as suggested by the reviewer, would not be suitable for evaluating α -syn fibril formation, as these SG markers are no longer present at the relevant time points.

3. Puromycin-induced UBQLN2 droplets, which are not stress granules, might offer a cleaner system to study α -synuclein fibrillization. Repeating the FRAP experiments in these droplets could be informative—if α -synuclein also shows slowed recovery here, it would lend further support to the conclusions drawn from the arsenite-treated SG model.

Our response:

We found that puromycin at this concentration induced the formation of large α -syn condensates with a low fluorescence recovery rate, as shown in revised version Fig. EV3E. Moreover, as noted by the reviewer, we also observed that puromycin at 3 μ g/mL did not induce SG formation for up to 8 h (revised version Fig. EV3F), indicating that puromycin promotes the formation of α -syn condensates independently of SG formation.

These data and the relevant sentences have been included in the revised text (Page 7, Lines 18–24, Page 10, Lines 19–20, and Fig. EV3E and F).

Fig. EV3 E

α -Syn (WT)-EGFP/SH-SY5Y cells were treated with 3 μ g/ml puromycin for 8 h and subsequently immunostained with anti-UBQLN2 (a), anti-UBQLN1 (b), or anti-UBQLN4 and DAPI. (c). UBQLN2 droplets and large α -syn-EGFP condensates are indicated by white arrows. (d) FRAP analysis was performed on α -syn droplets following treatment with 3 μ g/ml puromycin for 8 h. Plots show the average FRAP recovery curves from at least 8 separate droplets. Values are the mean \pm SD. Quantification of fractional recovery at 70 s after photobleaching from at least 8 separate droplets. Values are the means \pm SDs. Non-puncta regions were photobleached as a control in the FRAP analysis. ****p < 0.001 (Mann–Whitney test).

Fig. EV3 F

α -Syn(WT)-EGFP/SH-SY5Y cells were treated with 3 μ g/ml puromycin for 8 h or 50 μ M AsNaO₂ for 2h, and subsequently immunostained with anti-G3BP and DAPI.

4. The claim that only UBQLN2 triggers α -synuclein condensation under puromycin treatment is not fully supported, as it's unclear whether puromycin induces condensate formation by UBQLN1 or UBQLN4 in the first place. Without showing that UBQLN1 or UBQLN4 form puncta under these conditions, it's difficult to conclude that their inability to

promote α -synuclein condensation is due to a functional difference rather than a lack of condensate formation. The authors should include controls demonstrating whether UBQLN1 and UBQLN4 also phase separate under puromycin treatment to support this conclusion.

Our response:

We found that puromycin induces phase separation of UBQLN1 and UBQLN4 similarly to UBQLN2. As shown in revised Fig. EV3G, only UBQLN2 knockdown affected the formation of large α -syn puncta. These results indicate that UBQLN2 is the responsible factor for α -syn puncta formation, while UBQLN1 and UBQLN4 are likely incorporated through their interaction with UBQLN2.

5. Lack of quantification throughout the paper or western blots, FRAP decreases the rigour and therefore significance of the studies. It is critical to substantiate some claims with quantification. Important examples include Figure 2A, 2B. Figure 3E, Figure 5H and Figure S4C.

Our response:

In accordance with the reviewer's suggestion, the quantitative data for Figs. 2A, 2B, 3E, and 5H in the original manuscript have been incorporated into Appendix Fig. S2, Fig. 2B(d), Appendix Fig. S3D, and Appendix Fig. S5C, respectively, in the revised manuscript.

The quantitative data for Fig. S4C were shown in Fig. 4G in the revised manuscript.

Figure 2A

Figure 2B

Appendix Fig.S3D

Appendix Fig.S5C

Optional:

1. SO286 binds the STI1 regions of UBQLN2, which are also implicated in α -synuclein binding. While the compound clearly inhibits fibril formation, it does not prevent initial α -synuclein incorporation into droplets, leaving it unclear whether the observed inhibition is due to disrupted recruitment, blocked oligomerization, or both. Without a direct comparison between STI1-1 and STI1-2 deletion mutants and their sensitivity to SO286, the mechanism remains ambiguous. The authors could choose to test SO286 effects in cells or in vitro using UBQLN2 constructs lacking either STI1-1 or STI1-2 to clarify whether the compound's action overlaps with the regions required for direct α -synuclein binding and/or UBQLN2 self-interaction. This would help define the structural basis of SO286's inhibitory effect.

Our response:

As shown below, UBQLN2(Δ STI1-1) retains the ability to form droplets and incorporate α -syn, whereas UBQLN2(Δ STI1-2) fails to form droplets. These observations indicate that the STI1-2 region is essential for droplet formation.

Although SO286 binds to both the STI1-1 and STI1-2 regions with similar K_d values, the STI1-2 region appears to be the major binding site for α -syn. Therefore, deleting the STI1-2 region significantly affects the protein's ability to form droplets and interact with α -syn, making it difficult to interpret the functional consequences of SO286 binding using these deletion mutants. In our view, such an approach would not provide a clear mechanistic insight into the SO286's inhibitory action and may lead to confounding results. For this reason, we have not pursued further deletion-based experiments in the revised manuscript, though we believe this remains an important topic for future structural studies.

Minor Comment:

Typographical correction: Page 7, line 9—correct "UNQLN4" to "UBQLN4."

Our response:

Thank you for noticing this error. We have corrected the typographical error as suggested.

Reviewer #2 (Significance (Required)):

This paper presents a strong case that UBQLN2 promotes α -synuclein fibril formation through a stress-induced phase transition, and it introduces SO286 as a potential way to block that process. The manuscript is well-written, the figures are clear, and most of the conclusions are supported by the data. That said, the study would benefit from deeper analysis of the stress-induced condensates in cells, consideration of alternative explanations, and more thorough quantification in key experiments.

Even with those caveats, the central idea—that UBQLN2 plays an active role in α -syn aggregation rather than just being present—is an important one and opens up a new direction

for therapeutic exploration. The data on α -synuclein fibril formation within UBQLN2 droplets and the mechanistic insights will be especially relevant to researchers working on protein aggregation in neurodegenerative disease, LLPS, and those focused on UBQLN2 or α -synuclein. The SO286 findings extend the paper's relevance to a broader neurodegeneration audience, particularly those interested in translational impact. Overall, this work moves the field forward in a meaningful way.

Comparable papers might include:

ALS-linked mutations impair UBQLN2 stress-induced biomolecular condensate assembly in cells.

Riley JF, Fioramonti PJ, Rusnock AK, Hehnly H, Castañeda CA. J Neurochem. 2021 Oct;159(1):145-155. doi: 10.1111/jnc.15453. Epub 2021 Aug 20. PMID: 34129687 Free PMC article.

alpha-Synuclein aggregation nucleates through liquid-liquid phase separation.

Ray S, Singh N, Kumar R, Patel K, Pandey S, Datta D, Mahato J, Panigrahi R, Navalkar A, Mehra S, Gadhe L, Chatterjee D, Sawner AS, Maiti S, Bhatia S, Gerez JA, Chowdhury A, Kumar A, Padinhateeri R, Riek R, Krishnamoorthy G, Maji SK. Nat Chem. 2020 Aug;12(8):705-716. doi: 10.1038/s41557-020-0465-9. Epub 2020 Jun 8. PMID: 32514159

Ubiquitin Modulates Liquid-Liquid Phase Separation of UBQLN2 via Disruption of Multivalent Interactions.

Dao TP, Kolaitis RM, Kim HJ, O'Donovan K, Martyniak B, Colicino E, Hehnly H, Taylor JP, Castañeda CA. Mol Cell. 2018 Mar 15;69(6):965-978.e6. doi: 10.1016/j.molcel.2018.02.004. Epub 2018 Mar 8. PMID: 29526694

Ubiquilin-2 regulates pathological alpha-synuclein.

Sandoval-Pistorius SS, Gerson JE, Liggans N, Ryou JH, Oak K, Li X, Negron-Rios KY, Fischer S, Barsh H, Crowley EV, Skinner ME, Sharkey LM, Barmada SJ, Paulson HL. Sci Rep. 2023 Jan 6;13(1):293. doi: 10.1038/s41598-022-26899-0.

Keywords for my field of expertise: stress granules, UBQLN, neurodegenerative disease, LLPS, ALS/FTD

I do not have enough expertise to confidently evaluate the HS-AFM imaging or Surface plasmon resonance (SPR) analysis.

Reviewer #3 (Evidence, reproducibility and clarity (Required)):

The study by Takei et al. has examined how phase separation of alpha-synuclein, the major aggregating protein in Parkinson's disease, is influenced by Ubiquilin-2 (UBQLN2), since previous studies had implicated UBQLN2 in various neurodegenerative diseases, including Parkinson's disease. UBQLN2 itself is a protein that can undergo phase separation and partition into stress granules in cells and functions in transferring ubiquitinated proteins to the proteasome or autophagy. The authors used recombinant a-syn and UBQLN2 to demonstrate that a-syn is incorporated into UBQLN2 droplets and that this leads to irregularly shaped, undynamic a-syn-UBQLN2 co-condensates, from which fibrillar a-syn species emerge. They then use SH-SY5Y cells stably expressing EGFP-tagged a-syn to demonstrate that AsNaO2 stress, which induces droplet-like structures of UBQLN2 in cells, leads to co-condensation of UBQLN2 and a-syn in the cytoplasmic puncta, which gradually lose their dynamics and become Thioflavin S-positive. Finally, they utilize a compound (SO286), which they had previously identified in a compound screen to selectively bind UBQLN2 with high affinity at the STI-1 and STI-2 regions. The authors show that this compound is able to suppress the UBQLN2-a-syn interaction and thus suppresses the UBQLN2-induced liquid-to-solid transition of a-syn in vitro and in cells.

The study still lacks a few important controls and has some obvious gaps and shortcomings in its presentation that should be addressed before publication. The methods section lacks clarity and misses important information, and the discussion could better discuss their findings in light of previous findings on UBQLN2 and stress granules as well as the limitations of their approaches. I summarize my major and minor comments below.

Major points:

1- A key question to address is whether the observed effect is specific to a-synuclein. How about other disease-linked phase separating proteins, e.g. TDP-43 or FUS (since UBQLN2 has been linked to ALS/FTD) or Tau? At least one of them should be used as a control protein, to get an idea whether this is an a-syn-specific effect or not.

Our response:

We thank the reviewer for the insightful comments. As shown in the supporting data below, UBQLN2 droplets did not incorporate BSA (5% DyLight633-labeled), indicating that the observed incorporation of α -syn by UBQLN2 is not a nonspecific event. However, following the reviewer's suggestion, we examined whether tau is incorporated into UBQLN2 droplets *in vitro*. As shown in Fig. EV5D, tau protein at 8 μ M (5% DyLight633-labeled) did not undergo LLPS under the conditions in which UBQLN2 formed droplets (3% PEG); however, it was immediately incorporated into the UBQLN2 droplets. Furthermore, tau aggregated and solidified within UBQLN2 droplets after 96 h, as assessed by FRAP analysis, as shown below (B). This tau aggregation in UBQLN2 droplets was inhibited by SO286. Although detailed and extensive studies are currently underway, our results suggest that UBQLN2 LLPS may play a common and important role in disease development by promoting the fibrillization of disease-associated,

aggregation-prone proteins. The data and the relevant sentences have been included in the revised text (Fig. EV5D and Page 9, Lines 22–26 and Page 12, Lines 3–10).

Supporting data

Fig. EV5D

(A) Fluorescence microscopy images of individual solutions containing 10 μ M UBQLN2 construct (1% DyLight488-labeled) and 8 μ M tau construct (5% DyLight633-labeled) in the presence of 3% PEG, as well as images of mixed solutions of UBQLN2 and tau under the same conditions. (B) FRAP analysis of UBQLN2 (top) and tau (bottom) droplets at 24 h or 96 h, in the presence or absence of SO286. The plots show average FRAP recovery curves from 10 individual droplets (*left*). Quantification of fractional recovery at 70 s after photobleaching from 10 separate droplets. **** $p < 0.0001$ (Dunnett's test, versus 96 h/Control). Values are the means \pm SDs. (*right*)

2– A related question is: How about other (non-UBQLN) condensate systems formed in presence of 3% PEG – does α -syn partition into them as well? The authors demonstrate that α -syn also partitions into UBQLN1 and UBQLN4 condensates, which raises the question

whether it will partition into any condensate under those experimental conditions. Also here, checking for specificity would be of great importance and highly informative.

Our response:

We observed that α -syn partitions into UBQLN1 and UBQLN4 droplets, as well as UBQLN2 droplets, but not into UBL-deleted UBQLN2 droplets (revised version Fig. EV1B). These results suggest that α -syn does not partition into any condensates under our experimental conditions. Additionally, in response to the reviewer's comment, we conducted preliminary experiments to examine whether α -syn partitions into non-UBQLN droplets. For this purpose, we selected tau protein as a candidate for forming non-UBQLN droplets. Indeed, tau at 8 μ M underwent LLPS to form droplets in the presence of 10% PEG, as described in the reference (Nature Communications, 11:2809, 2020). Under these conditions, α -syn did not form droplets on its own, but it did partition into the tau droplets. Accumulating evidence has demonstrated the overlap of α -syn and tau pathologies in patients with PD (Brain, 140:2982, 2017; J Neurol Neurosurg Psychiatry, 90:1234, 2019), and that tau interacts with α -syn and accelerates its aggregation (Brain, 145:3454, 2022). In this study, we found that α -syn was incorporated into tau droplets (Fig. EV1C), which may help explain why tau pathology is also commonly observed in the brain of patients with PD. The data and the relevant sentences have been included in the revised text (Fig. EV1C and Page 3, Lines 35–38 and Page 12, Lines 3–10).

Fluorescence microscopy images of individual solutions containing 8 μ M Tau construct (5% DyLight633-labeled) and 10 μ M α -syn construct (3% DyLight488-labeled) in the presence of 10% PEG, as well as images of mixed solutions of Tau and α -syn under the same conditions.

3– The interaction assays described/shown in Fig. 2 and Fig.5A need clarification, as the methods are not sufficiently described. I could not figure out whether this was a pure in vitro interaction assay (with recombinant proteins and which buffer was used), or whether it involved a cell lysate/pulldown from cell lysate, in which case the interaction could also be indirect. The results text on p. 5 implies it is a pure in vitro interaction assay with recombinant proteins, but there is no corresponding methods part, and the so-called “in vitro pulldown assays” described in the methods on p. 11 involves transfected HEK293 cells, generation of a lysate and pulldown with flag beads, which is not a true “in vitro” interaction

assay and one cannot claim that the observed interaction is really direct. Hence, the missing methods should be added, or the manuscript text should be corrected to accurately describe the interaction assays shown in Fig. 2A,B and Fig. 5A, as I could not figure out how they were really done. What was the reason why two different interaction assays (Pulldown, Fig. 2A, and Far Western, Fig. 2B) were used? It would be more convincing if the same assay was used to check interaction of the different UBQLNs (UBQLN1,2,4) and the different UBQLN2-mutants (WT, deltaSTI-1, deltaSTI-2), instead of using two completely different assays. The rationale for using these assays and what exactly was done should be given, otherwise it is hard to follow this part of the manuscript.

Our response:

We thank the reviewer for the comment and apologize for not providing a sufficiently detailed description of the methods used in these experiments.

Following the reviewer's suggestion, "In vitro pulldown assays" in the **Materials and Methods** section has been rephrased as "Interaction assays." Additionally, the protocols of these experiments have been revised as follows (the changes are indicated by underscores).

Interaction assays

To detect the interaction between UBQLN2 and α -syn, FLAG-tagged α -syn was purified from HEK293T cells, while HA-tagged UBQLN constructs were expressed and purified from *E. coli*. HEK293T cells were transfected 24 h after seeding using Lipofectamine 3000 with α -syn-FLAG. Forty-eight hours post-transfection, the cells were lysed in BA100 buffer (20 mM Tris-HCl [pH 7.5], 100 mM NaCl, 0.1% Triton-X 100, and proteinase inhibitor). The cell lysates were then incubated with anti-FLAG beads (Wako) for 3 h at 4 °C with constant rotation. The α -syn-FLAG/anti-FLAG beads were washed twice with BA100 buffer and three times with LLPS buffer before being incubated with either UBQLN1, UBQLN2, or UBQLN4 recombinant proteins for 2 h at 4 °C with constant rotation. The beads were then washed three times with LLPS buffer and eluted with FLAG peptide (Wako). Next, the samples were loaded onto a 10%–20% Bis-tris gel and the proteins were separated via sodium dodecyl sulfate polyacrylamide gel electrophoresis (SDS-PAGE). Proteins were then transferred onto a polyvinylidene fluoride (PVDF) membrane and probed with specific antibodies, followed by detection using West Dura Extended Duration Substrate (Thermo Fisher Scientific).

To detect the self-interaction of UBQLN2, FLAG-tagged UBQLN2 was purified from UBQLN1/2/4 triple-knockout T-REx 293 cells to eliminate interference from endogenous UBQLN proteins and to enable specific analysis of interactions with HA-tagged UBQLN2 purified from *E. coli*. The cells were transfected 24 h after seeding using Lipofectamine 3000 with FLAG-UBQLN2. Forty-eight hours post-transfection, the cells were lysed in BA100 buffer. The cell lysates were then incubated with anti-FLAG beads for 3 h at 4 °C with constant rotation. Next, the FLAG-UBQLN2/anti-FLAG beads were washed twice with BA100 buffer and three times with LLPS buffer before being incubated with HA-UBQLN2 recombinant protein for 2 h

at 4 °C with constant rotation. The beads were then washed three times with LLPS buffer and eluted with FLAG peptide. The samples then underwent SDS-PAGE and probing with antibodies using the same protocol as for the detection of interactions between UBQLN2 and α -syn-FLAG.

Furthermore, we appreciate the reviewer's question regarding the use of two different interaction assays. We initially used pulldown assays to evaluate protein-protein interactions. However, we observed that full-length UBQLN2 exhibited a tendency to bind nonspecifically to pulldown beads. While we were able to optimize the conditions to minimize this nonspecific binding for full-length UBQLN2 (as shown in Figs. 2 and 5), the same optimization was not sufficient when using UBQLN2 deletion mutants. To address this limitation, we conducted additional tests and found that far-western blotting provided a more reliable and specific method for detecting interactions involving the deletion mutants. Therefore, both assays were used based on their suitability for the specific UBQLN2 constructs analyzed.

4- For the AsNaO₂- or puromycin-induced UBQLN2 condensates in cells (Fig. 3), it would be important to clarify whether these structures are stress granules (SGs), e.g. by co-staining for established SG marker proteins. This would be expected, as previous studies have reported that UBQLN2 can be detected in SGs upon stress (e.g. Dao et al., 2018). If the authors find that the structures are indeed SGs, this would be highly interesting, because it would for the first time demonstrate that α -syn can also accumulate in SGs (or SG-like structures) after stress. If the structures do NOT overlap with known SG markers, it would be informative to know this as well and to investigate the nature of the biomolecular condensates (in future studies). Checking for the presence of SGs in their systems also seems important because of the knockout/knockdown experiments shown in Fig. 3, as previously UBQLN2 knockdown was shown to enhance SG numbers, whereas UBQLN2 overexpression was shown to suppress SGs (Alexander et al., PNAS 2018, PMID: 30442662). The discrepancies/ overlap to the findings by Alexander et al. should be discussed in the manuscript, so far I did not see this study mentioned.

Our response:

Thank you for raising this important point. As described in our response to Minor comment #3 of Reviewer #1, when SH-SY5Y cells stably expressing α -syn(WT)-EGFP were treated with sodium arsenite (AsNaO₂, 50 μ M), stress granules (SGs) formed after 3 h, as indicated by the presence of puncta positive for the SG markers G3BP. Most of these SGs were co-localized with UBQLN2 consistent with previous reports (PNAS 115: E11485, 2018 & Mol cell 69:965, 2018). However, the G3BP-positive puncta disappeared 6 h after AsNaO₂ treatment, indicating that SGs had disassembled by this time point. In contrast, the UBQLN2-containing α -syn droplets persisted and continued to grow over time, as shown below (Fig. EV3A). These findings suggest that UBQLN2 droplets are resistant to SG disassembly and undergo a liquid-to-gel phase transition, thereby promoting α -syn aggregation.

The data and the relevant sentences have been included in the revised text (Page 6, Lines 8–13 and Fig. EV3A).

However, although a previous report (PNAS 115: E11485, 2018) demonstrated that UBQLN2 knockout enhances SG formation under heat stress, we found that UBQLN2 knockout (as shown below, Appendix Fig. S3C) or UBQLN2 knockdown (as shown below, Fig. EV3D) did not affect both AsNaO₂-induced SG formation and disassembly, indicating that SG formation triggered by AsNaO₂ occurs independently of UBQLN2 LLPS in our cell system.

Nevertheless, it may be attributed to differences in the type of stress applied or the cell type used. Following the reviewer's suggestion, these observations and related discussion have been included in the revised text (Page 7, Lines 9–10, Page 10, Lines 15–25, Fig. EV3D and Appendix Fig. S3C).

(Fig. EV3A) α -Syn(WT)-EGFP/SH-SY5Y cells were stimulated with 50 μ M AsNaO₂ for 3, 6, and 12 h and immunostained with anti-UBQLN2, anti-G3BP and Hoechst.

(Appendix Fig. S3C) UBQLN2-KO α -syn(WT)-EGFP/SH-SY5Y cells were treated with 50 μ M AsNaO₂ for the indicated time periods, and immunostained with anti-G3BP and anti-eIF4G1 antibodies.

Fig. EV3D

(Fig EV3D) Fluorescence microscopy of α -syn(WT)-EGFP/SH-SY5Y cells with UBQLN2 knockdown. (Left) Cells were transfected with si-Control or si-UBQLN2, treated with 50 μ M AsNaO₂ for the indicated time points post 48 h transfection, and immunostained with anti-G3BP and DAPI. (Right) Quantification of the number of G3BP-positive condensates per cell. The values are normalized to the mean of the Ctrl si group at 1 h. N.S., nonsignificant. Statistical comparisons between control si and UBQLN2 si were performed at each time point using the Mann–Whitney test (n = 3–5).

5- Co-condensation of UBQLN2 and α -syn in cells was seen with EGFP-tagged α -synuclein stably expressed in SH-SY5Y cells. Can the authors also demonstrate the same co-condensation for endogenous α -synuclein, i.e. with antibody staining (in the absence of potential overexpression and EGFP-tagging), or in a more physiological model system of α -syn aggregation? Does UBQLN2 (but not UBQLN1 and 4 co-localize with α -syn aggregates in PD patients? These questions may go beyond the scope of this manuscript, but depending on the journal they aim for, the editors may want to see such data, and at minimum these points (limitations of current approach) should be addressed in the discussion.

Our response:

In this study, we conducted our analysis using overexpressed α -syn, as the expression levels of endogenous α -syn are low and difficult to detect. We are currently investigating this under more physiological conditions using PARK4 iPSC-derived neurons, in which α -syn is duplicated, as well as primary neuronal cells.

However, following the reviewer's suggestion, we conducted immunohistochemical studies on brain sections from patients with sporadic PD. We found that UBQLN2 immunoreactivity was observed in Lewy bodies (LBs) in the substantia nigra of four sporadic PD patients, suggesting that UBQLN2 may be pathologically involved in α -syn aggregation, as shown below. UBQLN1, and UBQLN4, albeit very rarely, were detected in some α -syn-positive inclusions, though less frequently than UBQLN2, possibly owing to the ability of UBQLN2 to form heterodimers with UBQLN1 and UBQLN4 (Biochem. J., 399: 397, 2006; EMBO Rep., 14: 373, 2013).

The data and the relevant sentences have been included in the revised text (Page 2, Lines 10–12, Page 7, Lines 29–35 and Fig. 3F and Fig. EV3H).

6- The authors show that only small α -syn condensates can be observed in UBQLN2 KO cells or upon siRNA-mediated UBQLN2 silencing (Fig. 3D). Did they check what happens upon UBQLN2 overexpression, does this promote the formation / enlargement of α -syn condensates?

Our response:

We have previously investigated the effect of UBQLN2 overexpression on the formation of α -syn condensates. When FLAG-UBQLN2 was overexpressed in α -syn-EGFP/SH-SY5Y cells, UBQLN2 formed puncta, as detected by anti-FLAG staining, even in the absence of AsNaO₂ treatment. Notably, judging from the FRAP analysis, α -syn also formed puncta that co-localized with those of UBQLN2 (revised version Fig. EV3B), suggesting that overexpressed UBQLN2 sequesters α -syn and promotes α -syn condensate formation.

These data and the relevant sentences have been included in the revised text (Page 6, Lines 31–37 and Fig. EV3B).

Fig.EV3B

7– The fractionation into sarkosyl-soluble/insoluble presented in Fig. 3E and 5H is not convincing without a proper quantification, taking the ratio of phosphorylated S129p-Syn to total α -synuclein into account.

Our response:

Following the reviewer’s suggestion, the quantitative data for Figs. 3E and 5H have been added to the corresponding figures in the revised manuscript (Appendix Figs. S3D, S5C).

Appendix Fig.S3D

Appendix Fig.S5C

8- The data with compound SO286 are quite striking, however it would be important to show a negative control (e.g. compound of similar chemical properties from their compound screen), which does not exert these effects at the same concentrations (especially since the used concentrations are quite high, e.g. 2– 20 μ M in Fig. 4E).

Our response:

As described in our response to Reviewer #1, Minor Comment #4, we have previously identified SO82 as an inactive analog of SO286 that does not bind to UBQLN2, UBQLN1, or UBQLN4 (shown in Fig. 4A and B). SO286, but not SO82, inhibited both UBQLN2–UBQLN2 and UBQLN2– α Syn interactions (revised version Figs. 4H and 5A). Using SO286 and SO82, we examined their effects on UBQLN2-catalyzed α -syn aggregation both *in vitro* and in a cultured cell system.

As shown in revised version Fig. EV5B, SO286, but not SO82, suppressed the time-dependent spherical collapse of UBQLN2 droplets *in vitro* (revised version Figs. EV5B, 5C). Furthermore, in the cultured cell system, SO286 inhibited the formation of large UBQLN2/ α -syn condensates, whereas SO82 did not (revised version Fig. EV5E).

These data and the corresponding descriptions have been included in the revised manuscript (Page 8, Lines 1–4, 20–21, and 32, and Page 9, Lines 14–18 and 33–34, and Figs. 4A, 4B, 4H, 5A, EV5B, EV5C, and EV5E).

Minor points:

1- *Methods part (p. 11) "Expression and purification of recombinant proteins" is missing α -syn expression / purification protocol, which construct was used etc. For all proteins, the OD_{260/280} ration should be given, to give an idea about the degree of nucleic acid contamination.*

Our response:

We would like to clarify that the expression and purification protocols for recombinant α -syn are indeed included in the Methods section (Page 13, Line 30 to Page14, Line 8). We apologize if this was unclear or difficult to locate.

It appears that it would be more appropriate to separate the purification protocols for the UBQLN and α -syn constructs. Therefore, the protocol in the "Expression and purification of recombinant proteins" section has been revised as follows (the changes are indicated by underscores). Additionally, following the reviewer's suggestion, the OD_{260/280} ration of each protein has been included (Page 14, Lines 7–8)

"UBQLN1, UBQLN2, UBQLN4, HA-UBQLN2, and α -syn constructs were amplified, subcloned, and inserted into the pGEX-6p-1 vector (Cytiva).

Glutathione S-transferase (GST)-tagged UBQLN constructs were expressed in the *Escherichia coli* strain (TaKaRa) in Luria-Bertani broth (Nacalai Tesque) at 37 °C overnight. Bacteria were induced with isopropyl β -D-1-thiogalactopyranoside and harvested after 4 h at 37 °C. The bacterial pellets were frozen, lysed in phosphate-buffered saline (PBS; TaKaRa) containing 0.1% Triton X-100 and protease inhibitor cocktail (Merck Millipore), and cleared by centrifugation at

13,000 × g for 15 min at 4 °C. GST-tagged proteins were purified using Glutathione Sepharose 4 Fast Flow (Cytiva). To cleave the GST tags, the beads were incubated with PreScission Protease (Cytiva) for 16 h at 4 °C. Flow-through fractions containing unlabeled recombinant proteins were collected, concentrated, and the buffer was exchanged with LLPS buffer (200 mM NaCl, 20 mM sodium phosphate, and 0.5 mM ethylenediaminetetraacetic acid; pH 6.8) using an Amicon Ultra Centrifugal Filter (Merck Millipore).

GST-tagged α -syn was expressed in *Escherichia coli* and purified using the same method as for UBQLN constructs, which involves affinity chromatography with Glutathione Sepharose 4 Fast Flow followed by cleavage of the GST tag using PreScission Protease.

The 260/280 absorbance ratios of the prepared UBQLN1, UBQLN2, UBQLN4, and α -syn proteins are 0.90, 0.85, 0.93, and 0.85, respectively.”

2– Methods part (p.11) “Imaging of LLPS” should be rephrased as “Formation and imaging of condensates” and more details should be given, i.e. temperature, observation time, was the fluorescence microscope a wide-field or confocal microscope etc.

Our response:

Following the reviewer’s suggestion, “Imaging of LLPS” has been rephrased as “Formation and imaging of condensates.” Moreover, the protocol has been revised as follows (the changes are indicated by underscores).

“Formation and imaging of condensates

The UBQLN1, UBQLN2, and UBQLN4 constructs were prepared to contain 10 μ M protein (spiked with DyLight488-labeled UBQLN1, UBQLN2, or UBQLN4 at a 1:100 molar ratio) in LLPS buffer containing 3% PEG (for UBQLN2 and 4) or 5% PEG (for UBQLN1). These samples were mixed with 10 μ M α -syn (spiked with DyLight633-labeled α -syn, 1:100 molar ratio), and SO286 was added to a 96-well glass bottom plate (Mattek) and incubated with 3% BSA (Sigma-Aldrich) to reduce the rapid coating of the protein droplets onto the glass surface. Condensates were imaged at 1, 24, 48, 72, and 96 h after phase separation using a BZ-X810 fluorescence microscope (Keyence) at room temperature.”

3– Fig. 5C: remove horizontal lines behind the bars and describe quantification in methods. Why was the “area covered by proteins” quantified here? This does not seem to be the best parameter when analyzing morphological changes... a parameter like roundness seems better suited.

Our response:

Fig. 5C has been revised as follows.

Fig. 5C (revised version) presents quantitative data from high-speed AFM, as shown in Fig. EV5A (revised version). Because spinodal decomposition-type phase separation was observed in

the high-speed AFM analysis, quantification was performed based on the area rather than the roundness.

4- Fig. 5C not mentioned in results text. Why does the legend say 0.25 mM SO286, whereas Fig. 5B implies that 20 μ M of the compound were used? Is Fig. 5C not a quantification of Fig. 5B, but a totally different experiment? These points need clarification.

Our response:

Thank you for pointing this out. Figure 5C is cited in the Results section of the original manuscript (Page 7): Intriguingly, SO286 treatment specifically inhibited the spinodal decomposition of the UBQLN2/ α -syn mixture, but not that formed by α -syn and UBQLN1 or UBQLN4 (Fig. 5C; revised version Fig. EV5A). Figure 5B shows fluorescence images obtained with 20 μ M SO286, whereas Fig. 5C reports the quantification of high-speed AFM images (revised version Fig. EV5A) with 0.25 mM SO286. To avoid confusion, we revised Fig. 5C's legend as follows:

Each UBQLN (1 μ M) was premixed with SO286 (0.25 mM) before being mixed with α -syn (1 μ M). After high-speed AFM imaging, the resulting images (revised version of Fig. EV5A) were used to quantify the proportion of surface area occupied by the proteins (α -syn together with UBQLN1, UBQLN2, or UBQLN4).

Reviewer #3 (Significance (Required)):

The study presents some surprising findings (potential influence of UBQLN2 on α -synuclein aggregation; potential recruitment of α -syn into SGs or SG-like structures upon stress,

Full Revision

druggability of the interaction), and utilizes a large variety of methods (both in vitro and in cells). Hence, if the authors can clarify some of the remaining major concerns (see above), it could potentially be of great significance. Currently there is great interest in finding modulators of phase transitions of disease-linked aggregating proteins. Hence, if properly revised and the major concerns raised above can be adequately addressed, the study should be of great interest to the neurodegeneration and phase separation community.

1. Description of the revisions that have already been incorporated in the transferred manuscript

Please insert a point-by-point reply describing the revisions that were already carried out and included in the transferred manuscript. If no revisions have been carried out yet, please leave this section empty.

2. Description of analyses that authors prefer not to carry out

Please include a point-by-point response explaining why some of the requested data or additional analyses might not be necessary or cannot be provided within the scope of a revision. This can be due to time or resource limitations or in case of disagreement about the necessity of such additional data given the scope of the study. Please leave empty if not applicable.

Prof. Masaya Imoto
Juntendo University Graduate School of Medicine
Division for Development of Autophagy Modulating Drugs
2-1-1 Hongo Bunkyo
Tokyo, Tokyo 1138421
Japan

8th Aug 2025

Re: EMBOJ-2025-121908-T
Ubiquilin-2 liquid droplets catalyze α -synuclein fibril formation

Dear Dr. Imoto,

Thank you for transferring your revised manuscript on UBQLN2 droplets and alpha-synuclein fibrils from Review Commons to The EMBO Journal. Given the general interest and the constructive encouragement by the original referees, I decided to send the study back to them and treat it essentially like a regular revision. I have now received feedback from all three reviewers, and given their overall positive comments (see below), would like to consider the study further for EMBO Journal publication, pending adequate addressing of a few remaining concerns raised by referees 1 and 3.

When preparing a revised manuscript, please try to adhere to the guidelines listed below and in our Guide to Authors as closely as possible, as this should greatly facilitate our assessment at the time of resubmission - in particular regarding the completion of our author checklist and a dedicated reagents and tools table (both linked below), the inclusion of separate, individual image-only files for main and EV figures, and the grouping of Appendix figures and tables into a single Appendix PDF starting with a brief header and table of contents page. Please also carefully double-check the bibliography, as several references appear to be incomplete (lacking volume/page numbers) or duplicated at this stage; as well as the order and labeling of the various manuscript sections. Finally, you shall also receive a formal request for preparation and provision of figure source data.

Please do not hesitate to contact me should you have any further questions regarding this final revision round or the resubmission requirements.

Thank you again for the opportunity to consider this study for The EMBO Journal. I look forward to receiving your revision.

Yours sincerely,

Hartmut Vodermaier

- a point-by-point response to the referees' comments, with a detailed description of the changes made (as a word file).
- a word file of the manuscript text.
- individual production quality figure files (one file per figure)
- a complete author checklist, which you can download from our author guidelines (<https://www.embopress.org/page/journal/14602075/authorguide>).

- Expanded View files (replacing Supplementary Information)

- a Reagents and Tools Table as part of the Methods section, which can be downloaded from our author guidelines

(<https://www.embopress.org/page/journal/14602075/authorguide#structuredmethods>)

Revision to The EMBO Journal should be submitted online within 90 days, unless an extension has been requested and approved by the editor; please click on the link below to submit the revision online before 6th Nov 2025:

Link Not Available

Referee #1:

This revision has addressed many comments raised by the reviewers. There are several key insights here, the main being that UBQLN2 droplets/condensates (and perhaps other proteins too) provide the right environment for certain proteins to aggregate. This builds on prior observations about the constrained, percolated environment inside a condensate. Intriguingly, the authors identify ST11 domains of UBQLN2 as interactors of α -synuclein. UBQLN2 can colocalize with α -synuclein (and tau too). Key to this work, they identify SO286 as a compound that can disrupt some of the liquid-to-solid transitions of UBQLN2 condensates in vitro and in cells; this disruption also affects α -synuclein aggregation. There are still a couple of questions that I hope the authors can address:

(1) The 24h images in Fig 4F(b) for the Control and SO82 (20 μ M) are identical.

(2) I am trying to better understand how SO286 works. It is intriguing that the compound perturbs the liquid-to-solid transition but not the overall phase separation property of UBQLN2. Perhaps the reason for not dissolving the droplets is that SO286 needs to be stoichiometric with ST11 domains. Given the high concentration of UBQLN2 inside the droplets, you would need substantially more of SO286 to disrupt phase separation. It would be useful to know how much SO286 partitions into droplets to better understand how much SO286 is necessary to interact with UBQLN2 in the droplet phase?

(3) Figure 5G: the panel showing SO286 effects is missing if I recall correctly, compared to the original version of the paper. I believe the original version had both control and SO286 images. Am I missing something?

(4) There are two recent reports that may be important for later followup work regarding structural elucidation of the ST11 interface as mentioned in the discussion on pg. 10:

<https://www.biorxiv.org/content/10.1101/2025.03.14.643327v2>

<https://www.biorxiv.org/content/10.1101/2024.07.10.602902v2>

Minor comments:

Line 10 on page 3 - ubiquitin effects on disassembling UBQLN2 droplets should also cite Dao et al 2018.

Fig 2A - on pg. 5, the description in text should also include mention of UBQLN1/4?

Fig 3B - number of cells - could this be increased just to improve quantification/comparisons? According to the legend, at least 9 images per condition set were collected, so could there be more than 10 cells used to quantify large puncta?

Referee #2:

I have reviewed the authors' response to the points raised in my initial review and found that they have met or exceeded my recommendations for additional clarifying experiments and quantifications, particularly in addressing my concerns about the characterization of UBQLN2 condensates. Their new experiments, including EM imaging, convincingly demonstrate the persistence of UBQLN2/ α -synuclein droplets following stress granule disassembly, supporting the specificity of the α -synuclein interaction with UBQLN2 condensates. The inclusion of FRAP analysis of α -synuclein within puromycin-induced UBQLN2 condensates further strengthens their conclusions.

In response to Reviewer #3, the authors also extended their analysis to include tau. Their initial experiments show that tau exhibits behavior similar to α -synuclein, and that SO286 has a comparable effect. This is especially significant given previous findings that UBQLN2 regulates tau accumulation in disease, and it raises the possibility that UBQLN2 phase separation may be

a shared mechanism contributing to multiple proteinopathies involving UBQLN2-interacting proteins or substrates. Overall, the authors have thoroughly addressed the concerns of all three reviewers. The additional experiments enhance the strength and clarity of the conclusions and, in my opinion, raise the impact of the study to a level appropriate for publication in The EMBO Journal.

Referee #3:

The authors have thoroughly revised their manuscript and addressed all major concerns raised by the reviewers. Only a few minor concerns remain to be addressed:

1- The quantifications shown in Appendix figures S2, S3 and S5 lack error bars and statistics. This should be fixed, or it should be explicitly said that the quantification comes from 1 replicate only. However, if the quantifications indeed comes from multiple replicates, it should be mentioned how many replicates / experiments were used for the quantification and error bars/statistics should be added.

2- I also did not spot a section on "Western blot quantification" in the methods section - this should be added or better highlighted (in case it was buried somewhere and I just overlooked it...). Here it should be detailed how the quantifications were made, incl. how many replicates were quantified.

3- Fig. 5G: The figure panel shows ThioT staining after different time points of AsNaO₂-treatment in presence of SO286. Here the comparison to non-SO286 (or SO82 control)-treated cells is missing, so that one can judge what difference the compound makes for ThioT positivity. This essential comparison should be shown, otherwise the figure panel /data is meaningless and should be removed.

4- The new data on human post-mortem brain (Fig. 3F) (accumulation of UBQLN proteins in Lewy bodies of sporadic PD patients) looks very interesting! However here, a negative control staining, e.g. with an IgG control antibody + secondary antibody, or antibody specific to another Ub-binding protein or neurodegeneration-associated protein + a green secondary antibody, is necessary to show that the co-stainings for UBQLN1, 2 and 4 are really specific signals, and not any antibody staining (or intrinsic fluorescence) would lead to the green signal. Obviously such stainings would have to be carried out in parallel and imaged at equal settings / equal image processing.

5- Fig. 5C had a typo in the y-axis.

Additional suggestions /recommendations for improving the manuscript:

6- Some figures have very different font sizes in different panels; some bar graphs are colored, others black-and-white, i.e. the figure style is not yet very uniform. Especially the very small font size in some panels should be avoided to keep these panels legible, and figures would profit from a bit better alignment in style/design across different figures and panels.

7- In addition, I would suggest to move some figure panels from EV figures to main figures, since they seem very relevant to me and I would highlight them more by moving them to "center stage". Specifically, this concerns the SG data shown in EV3A - I think it is a very relevant and novel finding that alpha-synuclein initially moves into G3BP1-positive SGs, which then disappear but synuclein condensates/aggregates remain, so I would highlight this finding more. Same for the data on Tau/alpha-synuclein co-partitioning - this is not just a control, but a very interesting, potentially disease-relevant finding, hence I think it's worth moving the data currently shown in EV1C and EV5D into the main figures. In contrast, the interaction data shown in Fig. 2, for example, seem less relevant to me and could be moved to an EV figure.

Rev_Com_number: RC-2025-02962

New_manu_number: EMBOJ-2025-121908-T

Corr_author: Imoto

Title: Ubiquilin-2 liquid droplets catalyze α -synuclein fibril formation

We sincerely thank the editor and reviewers for their careful evaluation of our manuscript and their constructive feedback. We are encouraged by the positive comments and appreciation of our work. In response to the reviewers' suggestions, we have conducted the additional experiments and revised the manuscript accordingly. Our detailed, point-by-point responses to the editor's and reviewers' comments are provided below, and the corresponding changes have been highlighted in the revised manuscript for ease of reference.

Editor

Please also carefully double-check the bibliography, as several references appear to be incomplete (lacking volume/page numbers) or duplicated at this stage; as well as the order and labeling of the various manuscript sections. Finally, you shall also receive a formal request for preparation and provision of figure source data.

Our response:

We have carefully audited the bibliography and made all necessary corrections, including adding missing volume/issue/page numbers, removing duplicate entries, and standardizing formatting to the journal style. We also reviewed and corrected the order and labeling of all manuscript sections to conform with the journal's guidelines.

Microscopy source data files exceeding 300 MB (Figures 3, 6, and EV3) have been uploaded to the BioImage Archive at the following URL.

<https://www.ebi.ac.uk/biostudies/bioimages/studies/S-BIAD2253?key=13a69bf7-7dea-42a2-ba11-085d45ac0a2f>

All other Source Data files have been uploaded via the submission system.

Referee #1:

(1) The 24h images in Fig 4F(b) for the Control and SO82 (20 μ M) are identical.

Our response:

We sincerely thank the reviewer for pointing out that the same image was inadvertently used in Figure 4F(b). This was entirely our oversight during figure preparation, and we apologize for the confusion caused. We have replaced the incorrect image in for Control at 24 h with the correct one derived from the original dataset, and the revised figure is now included in the updated manuscript.

In addition to the issues raised by the reviewers, we independently identified an additional error involving the reuse of the same α -Syn image for the 1,6-HD 24 h condition in UBQLN4+ α -Syn, which was mistakenly duplicated from the α -Syn image for the none 24 h condition in UBQLN4+ α -Syn in Figure EV1D. Accordingly, we have replaced the incorrect image for the

1,6-HD 24-hour condition with the correct one from the original dataset. The revised figure has been included in the updated manuscript.

These corrections do not affect the results or conclusions of the study.

(2) I am trying to better understand how SO286 works. It is intriguing that the compound perturbs the liquid-to-solid transition but not the overall phase separation property of UBQLN2. Perhaps the reason for not dissolving the droplets is that SO286 needs to be stoichiometric with STII domains. Given the high concentration of UBQLN2 inside the droplets, you would need substantially more of SO286 to disrupt phase separation. It would be useful to know how much SO286 partitions into droplets to better understand how much SO286 is necessary to interact with UBQLN2 in the droplet phase?

Our response:

In response to the reviewer's comment, we tested whether higher concentrations of SO286 could disrupt the phase separation of UBQLN2. However, even at 100 μ M, UBQLN2 still underwent phase separation to form liquid droplets, as shown below. Preparation of SO286 at concentrations higher than 100 μ M was not feasible due to its limited solubility in DMSO. Therefore, we were unable to determine the concentration of SO286 required to affect the phase separation of UBQLN2.

(3) Figure 5G: the panel showing SO286 effects is missing if I recall correctly, compared to the original version of the paper. I believe the original version had both control and SO286 images. Am I missing something?

Our response:

We sincerely thank the reviewer for carefully checking our figures and pointing out this mistake. Indeed, the panel showing the effect of SO286 in Figure 5G in revised version was inadvertently omitted during the preparation of the revised manuscript. We apologize for this oversight. We have now included the correct panel in the revised Figure 6B in re-revised version, and confirm that the results and conclusions remain unchanged.

(4) *There are two recent reports that may be important for later followup work regarding structural elucidation of the STI1 interface as mentioned in the discussion on pg. 10:*

<https://www.biorxiv.org/content/10.1101/2025.03.14.643327v2>

<https://www.biorxiv.org/content/10.1101/2024.07.10.602902v2>

Our response:

We are grateful to the reviewer for highlighting these recent reports. We agree that they will be highly valuable for future follow-up work on the structural elucidation of the STI1 interface, and we will take them into careful consideration in our subsequent studies.

Minor comments:

Line 10 on page 3 - ubiquitin effects on disassembling UBQLN2 droplets should also cite Dao et al 2018.

Our response:

We thank the reviewer for this helpful suggestion. We have now cited Dao et al. (2018) in the revised manuscript (page 3, line 10).

Fig 2A - on pg. 5, the description in text should also include mention of UBQLN1/4?

Our response:

Following the reviewer's suggestion, we have added a sentence regarding UBQLN1/4 (page 5, line 16).

Fig 3B - number of cells - could this be increased just to improve quantification/comparisons? According to the legend, at least 9 images per condition set were collected, so could there be more than 10 cells used to quantify large puncta?

Our response:

Thank you very much for your valuable comment. As shown in the images of Fig. 3E(a) in re-revised version, there are approximately 10~15 cells (nuclei) in each image. Thus, for a total of 10 images, around 100 cells (nuclei) were quantified. For quantification, we calculated the number of puncta per cell (nucleus) by dividing the number of large puncta in the field by the number of nuclei. Because the frequency of arsenic-induced large puncta formation is approximately 10%, a certain degree of variability, as represented by the error bars, is expected from a biological standpoint. Nevertheless, the number of cells (nuclei) analyzed is sufficient to ensure statistical power, and the analysis consistently revealed statistically significant differences between conditions, supporting the robustness and biological relevance of our findings.

Referee #3:

1- The quantifications shown in Appendix figures S2, S3 and S5 lack error bars and statistics. This should be fixed, or it should be explicitly said that the quantification comes from 1 replicate only. However, if the quantifications indeed comes from multiple replicates, it should be mentioned how many replicates / experiments were used for the quantification and error bars/statistics should be added.

Our response:

The quantifications shown in Appendix Figures S2, S3, and S5 were derived from the representative blots shown in Figure EV2A (n=2), Figure 3D (n=2), Figure 3F (n=2), and Figures 6C and EV6B (n=2), respectively. The number of replicates has been added to each Figure legend.

2- I also did not spot a section on "Western blot quantification" in the methods section - this should be added or better highlighted (in case it was buried somewhere and I just overlooked it...). Here it should be detailed how the quantifications were made, incl. how many replicates were quantified.

Our response:

As suggested, we have added the following section to the Methods to describe the quantification of Western blotting:

Quantification of Western blotting

The band intensities of Western blot images were quantified using Fiji (ImageJ, NIH). For each blot, the intensity of target protein bands was measured by defining regions of interest (ROIs) around the bands, and background signals were subtracted. The resulting values were normalized to the corresponding loading control. The number of biological replicates analyzed and the corresponding statistical analyses are detailed in the Figure legends.

3- Fig. 5G: The figure panel shows ThioT staining after different time points of AsNaO2-treatment in presence of SO286. Here the comparison to non-SO286 (or SO82 control)-treated cells is missing, so that one can judge what difference the compound makes for ThioT positivity. This essential comparison should be shown, otherwise the figure panel /data is meaningless and should be removed.

Our response:

We sincerely thank the reviewer for carefully checking our figures and pointing out this mistake. Indeed, the panel showing the effect of SO286 in Figure 5G in revised version was

inadvertently omitted during the preparation of the revised manuscript. We apologize for this oversight. We have now included the correct panel in the revised Figure 6B in re-revised version, and confirm that the results and conclusions remain unchanged.

4- The new data on human post-mortem brain (Fig. 3F) (accumulation of UBQLN proteins in Lewy bodies of sporadic PD patients) looks very interesting! However here, a negative control staining, e.g. with an IgG control antibody + secondary antibody, or antibody specific to another Ub-binding protein or neurodegeneration-associated protein + a green secondary antibody, is necessary to show that the co-stainings for UBQLN1, 2 and 4 are really specific signals, and not any antibody staining (or intrinsic fluorescence) would lead to the green signal. Obviously such stainings would have to be carried out in parallel and imaged at equal settings / equal image processing.

Our response:

Thank you for this important suggestion. As negative controls, we performed staining with the corresponding control IgG together with the secondary antibody. For rabbit IgG, a very weak background signal was detected; however, the signals obtained with UBQLN1 and UBQLN2 antibodies were more than 10-fold stronger and clearly distinguishable from the background. In contrast, mouse IgG controls showed no detectable signals. These results indicate that the observed UBQLN1, UBQLN2, and UBQLN4 signals in Lewy bodies are specific. We have now clarified this point in the revised text and included representative IgG control images in Fig. 3G.

Figure 3G. Sections from the midbrain of pathologically diagnosed PD cases were stained with each anti-UBQLN antibodies or anti-p- α -syn antibody. As negative controls, the corresponding control IgG (rabbit or mouse, at the same concentrations) was used together with the anti-p- α -syn antibody.

5- Fig. 5C had a typo in the y-axis.

Our response:

The typo in the y-axis of Figure 5C has been corrected as suggested.

Additional suggestions /recommendations for improving the manuscript:

6- *Some figures have very different font sizes in different panels; some bar graphs are colored, others black-and-white, i.e. the figure style is not yet very uniform. Especially the very small font size in some panels should be avoided to keep these panels legible, and figures would profit from a bit better alignment in style/design across different figures and panels.*

Our response:

According to the reviewer's suggestion, we have revised the figures to ensure uniformity in font sizes, color schemes, and overall style across all panels.

7- *In addition, I would suggest to move some figure panels from EV figures to main figures,*

since they seem very relevant to me and I would highlight them more by moving them to "center stage". Specifically, this concerns the SG data shown in EV3A - I think it is a very relevant and novel finding that alpha-synuclein initially moves into G3BP1-positive SGs, which then disappear but synuclein condensates/aggregates remain, so I would highlight this finding more. Same for the data on Tau/alpha-synuclein co-partitioning - this is not just a control, but a very interesting, potentially disease-relevant finding, hence I think it's worth moving the data currently shown in EV1C and EV5D into the main figures. In contrast, the interaction data shown in Fig. 2, for example, seem less relevant to me and could be moved to an EV figure.

Our response:

According to the reviewer's suggestion, Figures EV1C, EV3A, and EV5D have been moved to Figures 1B, 3B, and 5F, respectively, in the re-revised version. In addition, Figure 2A has been moved to Figure EV2A.

Prof. Masaya Imoto
Juntendo University Graduate School of Medicine
Division for Development of Autophagy Modulating Drugs
2-1-1 Hongo Bunkyo
Tokyo, Tokyo 1138421
Japan

11th Sep 2025

Re: EMBOJ-2025-121908R
Ubiquilin-2 liquid droplets catalyze α -synuclein fibril formation

Dear Prof. Imoto,

Thank you for submitting your re-revised manuscript to The EMBO Journal. I have now carefully checked your responses to the referee issues that had remained in the previous round of review, and found all points satisfactorily addressed. We shall therefore be happy to accept and publish this work, but there are several important editorial points that would still need to be corrected:

- Please make sure to upload all main and EV figure files as IMAGE-ONLY (without legend text - legends should only be at the end of the main manuscript).
- On the other hand, legend text for Appendix figures and tables should only be present in the APPENDIX PDF, so please delete them from the main text file.
- In the Appendix PDF, Appendix Table S1 is currently highly pixelated - please make sure to replace it with a non-image/text-only version of the table to ensure proper resolution and readability.
- On the abstract page of the manuscript, please include 4-5 general keyword terms to enhance searchability.
- Please rename the Conflict of Interest section into "Disclosure and Competing Interests Statement", in accordance with our updated Guide to Authors (<https://www.embopress.org/competing-interests>)
- As we are switching from a free-text author contribution statement towards a more formal statement based on Contributor Role Taxonomy (CRediT) terms, please remove the present Author Contribution section and instead specify each author's contribution(s) directly in the Author Information page of our submission system during upload of the final manuscript. See <https://casrai.org/credit/> for more information.
- Please note that the "Data and materials availability" section has to be renamed simply to DATA AVAILABILITY. In this section, make sure to not only specify the accession code for the deposited dataset, but to also include a functional hyperlink to the respective database.
- Please correct the in-text reference for Appendix Table S1 (currently only called "Table S1").
- While the Source Data for the main figures has been correctly uploaded as individual archives for each figure, please note that all Source Data archives for Appendix and Expanded View figures should be combined in one top-level archive file before uploading.
- Please provide suggestions for a short 'blurb' text prefacing and summing up the conceptual aspect of the study in two sentences (max. 250 characters), followed by 3-5 one-sentence 'bullet points' with brief factual statements of key results of the paper; they will form the basis of an editor-written 'Synopsis' accompanying the online version of the article. Please also upload a synopsis image, which can be used as a "visual title" for the synopsis section of your paper. The image should be in PNG or JPG format, and please make sure that it remains in the modest DIMENSIONS of EXACTLY 550 PIXELS WIDE and 300-600 PIXELS HIGH.
- Finally, during routine pre-acceptance checks, our data editors have raised the following queries regarding figures, data, and legends, which I would ask you to address (ideally using the Track Changes option):
 - ** Please note that the EXACT p-VALUES should be provided in the legends of figures 1C, 3E, G, H; 4A, C, D, F; 5B, EV1 D, EV3 A, B D, F; EV5 C, EV 6 A,
 - ** Please note that the WHITE ARROWS are not defined in the legend of figures 1B, 3A, B; . This needs to be rectified.

I am therefore returning the manuscript to you for a final round of minor revision, to allow you to make these modifications and upload the revised files. Once we will have received them, we should be ready to swiftly proceed with formal acceptance and production of the manuscript.

Yours sincerely,

Hartmut Vodermaier

Revision to The EMBO Journal should be submitted online within 90 days, unless an extension has been requested and approved by the editor; please click on the link below to submit the revision online before 10th Dec 2025:

Link Not Available

Rev_Com_number: RC-2025-02962

New_manu_number: EMBOJ-2025-121908R

Corr_author: Imoto

Title: Ubiquilin-2 liquid droplets catalyze α -synuclein fibril formation